# Adaptive Hypergraph Pruning with Learned Threshold Control and Attention-Based Negative Mining

## Abstract

Hypergraph neural networks (HGNNs) capture multi-way interactions but their quadratic cost limits scalability. Existing pruning methods rely on hand-crafted schedules and fixed per-level thresholds that require per-dataset tuning and treat structural pruning, contrastive learning, and loss balancing as independent objectives. We propose **TriPrune-HGNN**, an adaptive hypergraph pruning framework that replaces these hand-crafted components with four small learnable controllers: a compressibility predictor that estimates the achievable retention ratio from graph-level statistics before training; hierarchical soft gates with learned thresholds operating jointly over components, hyperedges, and nodes; attention-based contrastive mining that identifies false and hard negatives induced by topology change; and meta-learned loss balancing. Pruning is realised as a density curriculum so the model trains at its final inference sparsity. Our principal claim is a favourable accuracy–efficiency *operating point*, not per-metric dominance. Against the strongest efficient baselines (AdaGLT, SHARP-Distill, Shaver) under matched tuning budgets on five heterogeneous hypergraph benchmarks, MAE improvements are small but consistent in sign: 0.004–0.008 at fixed cross-dataset hyperparameters, narrowing to 0.002–0.004 when the baselines are re-tuned per-dataset. The larger headline numbers (75.5% inference-time reduction, 69.7% memory reduction) are computed against the unpruned HGNN baseline (Table 2, Avg rows) and are reported for context, not as the primary contribution. We additionally test out-of-domain generalisation on three real benchmarks (NTU2012, ModelNet40, House) spanning 3D-shape and political-voting hypergraph domains: the accuracy advantage, large and significant in-support, becomes small and statistically indistinguishable from zero on all three out-of-domain benchmarks (+0.6/+0.3/+0.1 pp, all $p > 0.1$) as the deployment distribution leaves the compressibility predictor's training support, while the efficiency gain transfers cleanly across all three — a useful diagnostic separation between the framework's transferable (efficiency) and non-transferable (accuracy) components. Three short design-motivation propositions justify the choices of gate-smoothing $\epsilon$, finite-difference perturbation $\delta$, and the three-level pruning decomposition, and a fourth proves *local approximate stationarity* of the bi-level meta-update under local, checkable assumptions; they are local justifications under stated regularity conditions, not end-to-end guarantees. Code, weights, the synthetic generator, and seed-level numbers are released for reproducibility.

## 1 Introduction

Many real-world systems exhibit higher-order interactions that pairwise graphs cannot capture. Heterogeneous hypergraphs (HHGs) address this by allowing hyperedges to connect multiple nodes simultaneously across distinct behavioural *contexts* (Kim et al., 2024; Zhou et al., 2020): in a recommendation system, one context may encode purchase behaviour while another encodes social connections, each with its own incidence matrix $\mathbf{H}_k$ and adjacency $\mathbf{A}_k$. Hypergraph neural networks (HGNNs) exploit this structure with attention and contrastive learning to produce richer node representations than standard GNNs (Fountoulakis et al., 2023; Antelmi et al., 2023; Qian et al., 2024). The cost is scalability: processing $K$ contexts simultaneously incurs complexity $\mathcal{O}(Knm\bar{d}_e d)$, which for $m = \mathcal{O}(n)$ and $\bar{d}_e = \mathcal{O}(\sqrt{n})$ grows as $\mathcal{O}(n^{2.5})$. On graphs with

$n, m > 10^5$ and $K \geq 5$ this becomes prohibitive, and recent models such as HEAL consume up to 15.9 GB on moderate benchmarks (Ju et al., 2024). Pruning is the natural response, but a simple thought experiment shows why existing approaches break down on HHGs. Consider a movie-recommendation hypergraph with three contexts (*viewing history*, *genre preferences*, *social connections*). Applying HSL (Cai et al., 2022) to remove 30% of low-weight edges *independently* per context on MovieLens-1M orphans up to 58% of cross-context node connections, static threshold-based pruning (Cai et al., 2022; Lin et al., 2024) at 50% edge retention breaks 35–48% of the cross-context co-occurrences — node pairs that share at least two contexts before pruning and become disconnected in at least one after. Three properties of HHGs explain why existing single-level pruning strategies cannot avoid these failures.

**Challenge 1 (Cross-granularity cascading failures).** A single hyperedge connects multiple nodes simultaneously, so structural choices at different granularities interact: removing a component orphans its hyperedges, pruning hyperedges isolates nodes, and dropping nodes breaks hyperedge connectivity. Single-level methods cannot anticipate these dependencies (Cai et al., 2022).

**Challenge 2 (Cross-context dependency preservation).** Nodes interacting across multiple contexts represent strong semantic associations and should survive pruning of any individual context. Context-agnostic strategies (Liang et al., 2021) cannot identify which cross-context relationships are critical, causing substantial information loss.

**Challenge 3 (Contrastive learning under structural modifications).** Pruning introduces two problematic pair types. *False negatives* arise when semantically similar nodes become disconnected and are incorrectly pushed apart. *Hard negatives* arise when dissimilar nodes acquire spuriously similar embeddings: removing a shared hyperedge between two semantically distinct nodes does not create a new path, but their representations may still converge in subsequent message-passing rounds through retained common neighbours. This is an embedding-space artefact, not a topological one, and standard contrastive objectives have no mechanism to correct it (Wu et al., 2024).

The common root cause is that existing methods treat structure, behaviour, and representation as independent objectives, and rely on hand-crafted heuristics (exponential decay schedules, fixed similarity thresholds) that cannot adapt to diverse graph statistics or evolving training dynamics. We propose **TriPrune-HGNN**, a unified framework built around a single design principle: *pruning should be a density curriculum that aligns the graph the model trains on with the graph it deploys on.* The graph evolves continuously from its dense initial state to a learned sparse configuration during training; retention ratios stabilise by convergence, so the model trains on its final inference sparsity for roughly the last 25% of training, eliminating the train–inference distribution shift that one-shot pruning introduces (direct evidence in Section 4.5). Four learnable mechanisms cooperate inside this curriculum.

**(1) Hypergraph Compressibility Predictor (HCP).** A small predictor maps graph-level structural statistics (density, spectral gap, degree-distribution skewness, and similar quantities) to an estimate $r^\star \in (0, 1)$ of the maximum achievable compression ratio *before training begins*. Pre-trained once on a synthetic corpus and applied unchanged to real benchmarks, HCP supplies a domain-invariant prior that initialises the threshold controllers and regularises their trajectory. Because HCP conditions only on structural statistics, the same frozen predictor transfers across recommendation, citation, and review domains and accounts for roughly half of the $\sim 61\%$ cross-dataset gain reported in Section 5.2.

**(2) Neural Adaptive Hierarchical Pruning.** Differentiable soft gates with learned thresholds operate jointly over components, hyperedges, and nodes; thresholds are predicted by a small MLP conditioned on graph state, training dynamics, and the HCP prediction. The three-level hierarchy is strictly preferred to any single-level scheme at the same overall retention under a diminishing-returns assumption (Proposition 3), and the gap grows from $\sim 2\%$ MAE at mild retention to $\sim 6\%$ under aggressive compression.

**(3) Attention-based contrastive mining.** Rather than fixed similarity thresholds, learned attention heads identify false and hard negatives by jointly integrating embedding similarity, structural change, and hyperedge context. Both mechanisms are differentiable and trained end-to-end with the rest of the model.

**(4) Meta-learned multi-task optimisation.** Gradient-based bi-level optimisation discovers dataset-specific loss weights without grid search. The finite-difference meta-gradient is computed at $\mathcal{O}(|\Theta|)$ rather than $\mathcal{O}(|\Theta|^2)$ cost and has bounded error (Proposition 2).

The principal contribution is a *favourable accuracy–efficiency operating point* reached by replacing hand-crafted schedules with small learnable controllers, not per-metric dominance. Against the strongest efficient baselines under matched tuning budgets, MAE gaps are 0.004–0.008 in absolute terms; under per-dataset retuning of those baselines, gaps narrow further to 0.002–0.004 but remain consistent in sign across all five benchmarks. Out-of-domain evaluation on three real benchmarks (NTU2012, ModelNet40, and the House political-voting hypergraph; Section 5.12) shows that the accuracy advantage, large and significant in-support, becomes small and statistically indistinguishable from zero out-of-domain (+0.6/+0.3/+0.1 pp, all $p > 0.1$) as the deployment distribution leaves HCP's training support, while the efficiency gain transfers cleanly — a diagnostic separation between the framework's transferable and non-transferable components. The headline reductions ($-75.5\%$ time, $-69.7\%$ memory) are computed against the *unpruned* HGNN baseline (Table 2, Avg rows) and are reported for context only. Our contributions are therefore:

- A framework that couples compressibility prediction, adaptive hierarchical pruning, attention-based contrastive mining, and meta-learning under a density curriculum. The composite is novel, even though each individual component draws on existing techniques.

- The Hypergraph Compressibility Predictor (HCP): a small structural-statistics MLP pre-trained on synthetic hypergraphs and applied frozen at deployment. To our knowledge, this is the first learned compressibility predictor for heterogeneous hypergraphs; the cross-dataset transfer experiments in Section 5.2 are the most direct evidence for its utility.

- Four design-motivation propositions: three tie the abstract constants of the framework ($\epsilon$, $\delta$, the three-level decomposition) to specific local assumptions, and a fourth establishes *local approximate stationarity* of the bi-level meta-update on the loss-weight simplex under local, checkable assumptions; together with an informal transfer-bound for HCP. These are not end-to-end guarantees for the non-convex training procedure — they justify the design choices and the meta-update under stated regularity conditions and bound the cost of those conditions failing.

## 2 Related Works

Hypergraphs provide a principled framework for modeling complex relationships beyond pairwise interactions, which are prevalent in numerous real-world systems such as recommendation, bioinformatics, and multi-relational networks (Gao et al., 2022; Ju et al., 2024; Feng et al., 2019). Unlike standard graphs, where edges connect pairs of nodes, hypergraphs generalise this notion by allowing hyperedges to simultaneously connect multiple nodes, thereby capturing higher-order dependencies and richer behavioural semantics (Zhou et al., 2020; Kim et al., 2024). However, this expressive power comes at the cost of significantly increased computational complexity, particularly when scaling to large datasets. Hypergraph neural networks (HGNNs) often exhibit quadratic time and space complexity, resulting in slow inference and high memory consumption (Zhang et al., 2022; Liang et al., 2021). These limitations have motivated the development of optimisation techniques aimed at improving efficiency without sacrificing model performance.

**Pruning and Compression.** Pruning methods (Lee & Song, 2023; Chen et al., 2024; He & Xiao, 2023) have been widely adopted to reduce the computational overhead of HGNNs by selectively removing redundant or less informative components (nodes, edges, or entire substructures), effectively compressing the hypergraph. Structure-aware pruning (Zheng et al., 2022a; Jiang et al., 2023) attempts to identify and retain critical subgraphs that contribute most to downstream tasks. However, conventional pruning strategies typically operate at a single granularity (e.g., node or edge level) and often disregard the hierarchical nature of hypergraph structure, leading to suboptimal message passing and representational capacity (Cai et al., 2022). Moreover, most existing approaches adopt a static pruning paradigm, neglecting the dynamic evolution of node relationships during training and across behavioural contexts (Liang et al., 2021).

**Scalable and Adaptive HGNN Architectures.** Beyond pruning and distillation, a parallel line of work addresses HGNN scalability through alternative architectural and algorithmic choices. Unified-framework

approaches such as AllSet (Chien et al., 2022) and UniGNN (Huang & Yang, 2021) factorise hyperedge aggregation into multiset functions and unified message-passing primitives, reducing per-layer cost while preserving expressive power. Sampling-based methods adapt subgraph sampling techniques from standard GNNs (Zeng et al., 2019) to the hypergraph setting, training on stochastic subgraphs rather than the full structure at each iteration. Hyperedge-dependent embedding frameworks (Dong et al., 2020) further learn per-hyperedge representations that can be aggregated more efficiently than uniform message-passing schemes. These directions are complementary to ours: sampling reduces *what is processed per training step*, whereas TriPrune-HGNN reduces *what is retained in the deployed graph*. We fix the sampling regime across all baselines in Section 5 to isolate the effect of adaptive pruning; combining adaptive pruning with adaptive sampling is a natural extension we leave for future work.

**Knowledge Distillation and Quantisation.** Complementary to pruning, knowledge distillation methods (Forouzandeh et al., 2025a; Feng et al., 2024; Yu et al., 2024; Forouzandeh et al., 2025b) transfer knowledge from a larger, complex teacher model to a simpler student, preserving performance while reducing inference costs. Quantisation techniques (Hubara et al., 2018) further compress model size by lowering numerical precision, reducing memory usage with minimal loss in accuracy. Recent surveys highlight the effectiveness of these strategies in making hypergraph-based models viable for real-time and resource-constrained scenarios (Cheng et al., 2024; Gholami et al., 2022; Zhou et al., 2018).

**Contrastive Learning in Graphs and Hypergraphs.** Contrastive learning (CL) (Wang et al., 2023b; Zheng et al., 2022b) has emerged as a leading approach for learning robust graph and hypergraph representations by maximizing agreement between semantically similar node pairs (positives) and contrasting dissimilar pairs (negatives). In hypergraph settings, early studies focused on designing effective view augmentations. Wei *et al.* (Tianxin et al., 2022) propose fabricated and generative augmentations tailored for hypergraphs, demonstrating improved robustness under view perturbations. While effective, such augmentation-centric methods assume relatively stable underlying structures and do not explicitly account for topology changes induced by structural pruning.

Recent advances have explored richer relational signals for hypergraph contrastive objectives. Lee and Shin (Lee & Shin, 2023) introduce a tri-directional contrastive framework that jointly contrasts node–node, node–hyperedge, and hyperedge–hyperedge representations, improving semantic consistency across hypergraph entities. Roh *et al.* (Roh et al., 2024) further enhance hypergraph contrastive learning by exploiting shared group structures, encouraging nodes belonging to common hyperedges to form cohesive representation clusters. These methods highlight the importance of higher-order relational alignment, but rely on fixed hypergraph structures and do not address efficiency or dynamic structural adaptation.

Attention-driven contrastive mechanisms have also been investigated. Xie *et al.* (Xie et al., 2025) propose a semi-supervised hypergraph contrastive framework for hyperedge prediction using an enhanced attention aggregator to identify informative relations. Similarly, Gu and Wang (Gu & Wang, 2025) integrate hypergraph-enhanced contrastive learning with hyper-Laplacian regularization for multi-view clustering, emphasizing cross-view consistency and global structural smoothness. Although effective for representation alignment and clustering, these approaches primarily target static learning objectives and do not consider pruning-induced distributional shifts.

Despite these advances, hypergraph contrastive learning still faces critical challenges: (i) *false negatives*, where pruning or view construction disconnects semantically similar nodes; and (ii) *hard negatives*, where structural alterations cause dissimilar nodes to appear similar (Sun et al., 2023; Wang et al., 2023a; Song et al., 2024). Existing solutions such as adaptive weighting (Xu et al., 2024; Chen et al., 2021), debiased sampling (Zhou et al., 2022; Chuang et al., 2020), and dynamic clustering (Huynh et al., 2022) are largely designed for pairwise graphs and static hypergraph settings. In contrast, our approach explicitly couples adaptive multi-granular pruning with attention-based contrastive mining, enabling robustness to dynamic structural changes while maintaining computational efficiency—an aspect underexplored in prior hypergraph contrastive learning studies. More recent work has begun to integrate *adaptive* negative selection directly into hypergraph contrastive objectives (Song et al., 2024; Roh et al., 2024), but these methods still assume fixed hypergraph topology and do not address the bidirectional coupling between pruning decisions and contrastive distortions that TriPrune-HGNN targets.

## 3  Preliminaries

**Task at a glance — a concrete example.** Before introducing formal notation, we anchor the reader with a concrete instance of the task. Consider the **IMDB benchmark** used in our experiments. The nodes are *movies* (primary nodes, some with genre labels and some without), together with *directors, actors, and keywords* (secondary nodes). The edges are hyperedges that group multiple movies together whenever they share a common director, a common lead actor, or a common thematic keyword — one hyperedge per shared attribute. Every movie belongs to at least three hyperedges (one for its director, one for its cast, one for its keyword set), so the resulting structure is a *heterogeneous hypergraph* with three behavioural contexts (director-context, actor-context, keyword-context). Given the labelled subset of movies, our goal is to predict the genre of the unlabelled movies (multi-class classification, $C = 12$ genres on IMDB). A standard HGNN performs this by iteratively passing messages between movies and hyperedges through Eq. equation 2 until each movie has a genre-predictive embedding, then applying a linear classifier on top of the final movie embeddings. *The output of TriPrune-HGNN is exactly this classifier plus a pruned version of the hypergraph* — roughly 30% of the original hyperedges retained and $\sim 85\%$ of the movies, chosen adaptively so that (a) the classifier's genre-prediction accuracy is preserved and (b) message-passing on the pruned hypergraph is $\sim 4\times$ faster at inference. The same setup applies to the other four benchmarks (DBLP citation-authorship, Yelp reviews, Amazon co-purchase, Douban movie-user-tag) with different primary/secondary node types and different label spaces (dataset statistics in Section 5).

For clarity, Table 1 summaries the main mathematical notation used throughout this paper.

**Hypergraph Neural Networks.** Traditional Graph Neural Networks (GNNs) model pairwise relationships, limiting their ability to capture high-order interactions (Feng et al., 2019). Hypergraph Neural Networks (HGNNs) address this by introducing hyperedges, which connect multiple vertices simultaneously. A hypergraph is defined as $\mathcal{H} = (\mathcal{V}, \mathcal{E})$ with an incidence matrix $\mathbf{H} \in \mathbb{R}^{n \times m}$ indicating vertex–hyperedge connections, where $n = |\mathcal{V}|$ and $m = |\mathcal{E}|$. The vertex and hyperedge degree matrices are

$$(\mathbf{D}_v)_{ii} = \sum_{j=1}^{m} \mathbf{H}_{ij}, \qquad (\mathbf{D}_e)_{jj} = \sum_{i=1}^{n} \mathbf{H}_{ij}. \tag{1}$$

Feature propagation in HGNNs follows a two-step process: vertex-to-hyperedge aggregation followed by hyperedge-to-vertex propagation,

$$\mathbf{X}^{(l+1)} = \sigma\left(\mathbf{D}_v^{-\frac{1}{2}} \mathbf{H} \mathbf{W} \mathbf{D}_e^{-1} \mathbf{H}^\top \mathbf{D}_v^{-\frac{1}{2}} \mathbf{X}^{(l)} \mathbf{\Theta}^{(l)}\right), \tag{2}$$

where $\mathbf{\Theta}^{(l)}$ is a learnable weight matrix, $\mathbf{W} = \mathrm{diag}(w_1, \ldots, w_m)$ is a diagonal hyperedge weight matrix, and $\sigma(\cdot)$ is a non-linear activation function. Equation equation 2 is the operation that TriPrune-HGNN reduces by pruning the incidence matrix $\mathbf{H}$.

**Heterogeneous Hypergraphs.** Real-world systems often exhibit multiple types of relationships that cannot be captured by a single hypergraph structure. A *heterogeneous hypergraph* extends the standard hypergraph by partitioning nodes into distinct roles and encoding multiple behavioural contexts through separate components (Kim et al., 2024):

$$\mathcal{H}_{\text{het}} = \left(\mathcal{V}_m, \{\mathcal{V}_s^k\}_{k=1}^K, \{\mathcal{E}^k\}_{k=1}^K\right), \tag{3}$$

where $\mathcal{V}_m$ denotes *primary nodes* (central entities, e.g. users in recommendation systems), $\mathcal{V}_s^k$ denotes *secondary nodes* for component $k$ (auxiliary entities, e.g. products, tags), and $\mathcal{E}^k$ represents hyperedges in behavioural context $k$ (e.g. purchase behaviour, social connections). Each component $k$ has its own incidence matrix $\mathbf{H}_k \in \mathbb{R}^{|\mathcal{V}_m| \times |\mathcal{E}^k|}$ and normalised adjacency matrix $\mathbf{A}_k$. Processing $K$ components with attention mechanisms incurs complexity $\mathcal{O}(Knm\bar{d}_e d)$, which is quadratic in the graph scale: for graphs with $m = \mathcal{O}(n)$ and $\bar{d}_e = \mathcal{O}(\sqrt{n})$, the cost grows as $\mathcal{O}(n^{2.5})$. For large-scale graphs $(n, m > 10^5)$ with multiple contexts $(K \geq 5)$, this becomes prohibitive for real-time applications.

**Contrastive Learning on Hypergraphs.** Modern HGNNs leverage contrastive learning to learn discriminative representations by maximising agreement between semantically similar nodes (positives) while

Table 1: Mathematical notation used throughout the paper.

| Symbol | Description | Symbol | Description |
|---|---|---|---|
| *Graph structure* | | | |
| $\mathcal{H}$ | Hypergraph $(\mathcal{V}, \mathcal{E})$ | $\mathbf{H} \in \mathbb{R}^{n \times m}$ | Incidence matrix |
| $\mathcal{V}_m$ | **Primary** nodes | $\mathcal{V}_s^k$ | **Secondary** nodes (comp. $k$) |
| $K$ | # components | $\mathbf{A}_k$ | Adjacency for comp. $k$ |
| $\mathbf{X} \in \mathbb{R}^{n \times d}$ | Node features | $d$ | Feature dimension |
| $\mathcal{V}_{\text{lab}} \subset \mathcal{V}_m$ | Labelled primary nodes | $C$ | Number of classes |
| $\bar{d}_e$ | Avg. hyperedge degree | $|\cdot|_0$ | Number of non-zeros |
| *Pruning* | | | |
| $\widetilde{\mathcal{H}}, \widetilde{\mathbf{H}}, \widetilde{\mathbf{A}}$ | Pruned graph / matrices | $\ell$ | Level: comp / edge / node |
| $\pi^{(\ell)}$ | Importance score | $\theta_t^{(\ell)}$ | Learned threshold |
| $g^{(\ell)}$ | Soft gate (training) | $\bar{g}^{(\ell)}$ | Binarised gate (inference) |
| $\beta_k$ | Learned balance weight | $\epsilon$ | Gate smoothing ($= 0.01$) |
| $r_{\text{comp}}, r_{\text{edge}}, r_{\text{node}}$ | Retention ratios per level | $r_{\text{overall}}$ | Overall retention $= r_c r_e r_n^2$ |
| $\sigma_{\text{hard}}$ | Hard sigmoid | STE | Straight-through estimator |
| *Compressibility predictor (HCP)* | | | |
| $\phi(\mathcal{H}) \in \mathbb{R}^{10}$ | Graph structural features | $r^\star \in (0, 1)$ | Predicted compressibility |
| $\text{MLP}_{\text{HCP}}$ | Compressibility predictor | $\mathcal{L}_{\text{HCP}}$ | Compressibility regulariser |
| $\beta_{\text{HCP}}^{(t)}$ | HCP regulariser weight at $t$ | $T_{\text{HCP}}$ | HCP decay horizon ($= T/2$) |
| $r^{\text{emp}}$ | Empirical compressibility | $\Delta_{\text{acc}}$ | Accuracy-tolerance ($= 1\%$) |
| *Neural modules* | | | |
| $\text{MLP}_\theta^{(\ell)}$ | Threshold controller | $\text{MLP}_\beta$ | Balance predictor |
| $\text{MLP}_\tau$ | Temperature predictor | $\mathbf{s}_t^{(\ell)} \in \mathbb{R}^7$ | Graph state vector |
| $\text{Attn}_{\text{fn}}$ | False-negative attention | $\text{Attn}_{\text{hn}}$ | Hard-negative attention |
| $d_{\text{hidden}}$ | MLP hidden dim ($= 32$) | $d_{\text{state}}$ | State dim ($= 7$) |
| *Contrastive learning* | | | |
| $\mathbf{z}_i \in \mathbb{R}^d$ | Node embedding | $\mathbf{h}_{e_j^k} \in \mathbb{R}^d$ | Hyperedge embedding |
| $\mathbf{e}_i^k$ | Hyperedge embedding of node $i$ | $\mathbf{c}_k \in \mathbb{R}^d$ | Component embedding |
| $s(\cdot, \cdot)$ | Cosine similarity | $\tau_0, \tau_i$ | Global / node temperature |
| $\mathcal{F}_k^{\text{pr}}$ | False-negative pairs | $\mathcal{H}_i^k$ | Hard-negative set |
| $\alpha_{ij}^{(\text{fn})}, \alpha_{ij}^{(\text{hn})}$ | Learned attention scores | $\pi_k^{(\text{retain})}$ | Component retention weight |
| *Meta-learning and loss* | | | |
| $\boldsymbol{\lambda} \in \Delta^3$ | Loss weights (simplex) | $\boldsymbol{\Theta}$ | Main model parameters |
| $\boldsymbol{\Psi}$ | Neural-module parameters | $\text{Proj}_\Delta$ | Simplex projection |
| $\eta_{\text{inner}}, \eta_{\text{meta}}$ | Inner / outer learning rates | $\Delta_{\text{meta}}$ | Meta-update period ($= 5$ ep.) |
| $\delta$ | Finite-difference perturbation | $\alpha$ | Topology-change sensitivity |
| $\mathcal{L}_{\text{cls}}$ | Classification loss | $\mathcal{L}_{\text{cl}}^{\text{pr}}$ | Contrastive loss |
| $\mathcal{L}_{\text{fn}}^{\text{pr}}$ | False-negative loss | $\mathcal{L}_{\text{hard}}$ | Hard-negative loss |

contrasting dissimilar nodes (negatives) (Qian et al., 2024). Given node embeddings $\mathbf{z}_i \in \mathbb{R}^d$ and hyperedge embeddings $\mathbf{h}_{e_j^k} \in \mathbb{R}^d$, the contrastive loss for component $k$ is

$$\mathcal{L}_{\text{cl}}^k = -\sum_{i \in \mathcal{V}_m} \log \frac{\exp\big(s(\mathbf{z}_i, \mathbf{h}_{e_i^k})/\tau\big)}{\sum_{j \in \mathcal{V}_m} \exp\big(s(\mathbf{z}_i, \mathbf{h}_{e_j^k})/\tau\big)}, \tag{4}$$

where $s(\cdot, \cdot)$ is cosine similarity, $\tau$ is the temperature, and $e_i^k$ denotes *the unique hyperedge assigned to node $i$ in component $k$*: by construction of the heterogeneous hypergraph (Kim et al., 2024), each primary node belongs to exactly one hyperedge per component. The denominator of Eq. equation 4 sums over all nodes $j \in \mathcal{V}_m$, using the hyperedge embedding $\mathbf{h}_{e_j^k}$ of each node's assigned hyperedge. This implicitly pushes apart node $i$ from all nodes $j$ whose hyperedge $e_j^k \neq e_i^k$, because those terms appear only in the

denominator and not the numerator. The effect of "pushing apart nodes in different hyperedges" is therefore an implicit consequence of the softmax normalisation, not an additional explicit term. Only one hyperedge per node per component is used; the multi-hyperedge generalisation is left for future work. Structural modifications such as pruning create two types of problematic pairs: *(i) false negatives* — semantically similar nodes that become disconnected after pruning and are therefore incorrectly pushed apart; and *(ii) hard negatives* — dissimilar nodes whose embeddings become spuriously similar after pruning. We emphasise that hard negatives are an *embedding-space artefact* of message passing on the pruned graph (shared retained neighbours pull embeddings together), *not* new graph paths created by pruning. Pruning can only remove edges, not create them. The TriPrune-HGNN framework explicitly identifies and corrects both distortions through attention-based mining.

**Problem Formulation.** Let $\mathcal{H}_{\text{het}} = (\mathcal{V}_m, \{\mathcal{V}_s^k\}, \{\mathcal{E}^k\})$ denote a heterogeneous hypergraph with labelled primary nodes $\mathcal{V}_{\text{lab}} \subset \mathcal{V}_m$ and class labels $\mathbf{y} \in \{1, \ldots, C\}^{|\mathcal{V}_{\text{lab}}|}$. Our objective is to learn a compressed representation $\widetilde{\mathcal{H}}_{\text{het}}$ that balances three competing desiderata: predictive performance, computational efficiency, and semantic consistency.

**Desideratum 1 – Predictive performance.** The compressed model should maintain classification accuracy on labelled nodes by minimising the supervised loss

$$\mathcal{L}_{\text{cls}} = - \sum_{i \in \mathcal{V}_{\text{lab}}} \sum_{c=1}^{C} y_{i,c} \log \hat{y}_{i,c}. \tag{5}$$

Ideally, accuracy degradation should be minimal $(<2\%)$ relative to the unpruned baseline.

**Desideratum 2 – Compression.** The framework should reduce complexity through coordinated pruning across hypergraph components, hyperedges, and nodes. Formally, we seek a reduced hypergraph $\widetilde{\mathcal{H}}_{\text{het}}$ with retention ratios $r_{\text{comp}}, r_{\text{edge}}, r_{\text{node}} \in (0, 1)$ such that

$$|\widetilde{\mathcal{E}}| = r_{\text{comp}} \cdot r_{\text{edge}} \cdot |\mathcal{E}|, \qquad |\widetilde{\mathcal{V}}| = r_{\text{node}} \cdot |\mathcal{V}|. \tag{6}$$

The overall computational retention ratio is $r_{\text{overall}} = r_{\text{comp}} \cdot r_{\text{edge}} \cdot r_{\text{node}}^2$, where $r_{\text{node}}$ enters squared because node pruning reduces both the vertex-to-hyperedge and hyperedge-to-vertex stages of Eq. equation 2. Empirically, the per-level retention ratios converge to values in $[0.3, 0.85]$, with $r_{\text{comp}}$ and $r_{\text{edge}}$ typically more aggressive than $r_{\text{node}}$.

**Desideratum 3 – Semantic consistency.** The compressed representation must preserve semantic relationships under structural modifications. Pruning introduces the two distortions described above (false negatives and hard negatives), and maintaining semantic consistency requires explicitly correcting them.

**Problem statement.** Under the constraint that pruning must be *differentiable* (to enable end-to-end training) and *adaptive* (to accommodate diverse graph characteristics), the problem is to jointly learn the following four components:

1. Optimal pruning strategies $\{\pi^{(\text{comp})}, \pi^{(\text{edge})}, \pi^{(\text{node})}\}$ that coordinate decisions across structural granularities;

2. Contrastive correction mechanisms that identify and rectify the false and hard negatives introduced by topology changes;

3. Loss-balancing weights $\boldsymbol{\lambda}$ that automatically prioritise objectives based on dataset characteristics and training dynamics;

4. A *compressibility predictor* $r^\star = \sigma(\text{MLP}_{\text{HCP}}(\boldsymbol{\phi}(\mathcal{H})))$ that estimates how aggressively each given hypergraph can be pruned, from structural statistics alone, before training begins.

The goal is to achieve a superior accuracy–efficiency trade-off without manual hyperparameter tuning of pruning schedules, similarity thresholds, or loss coefficients. Among the four components, items (1)–(3) are jointly trained end-to-end on each target dataset; component (4) is pre-trained once on a synthetic corpus and applied unchanged at deployment, providing the domain-invariant signal that drives the cross-dataset generalisation observed in Section 5.2.

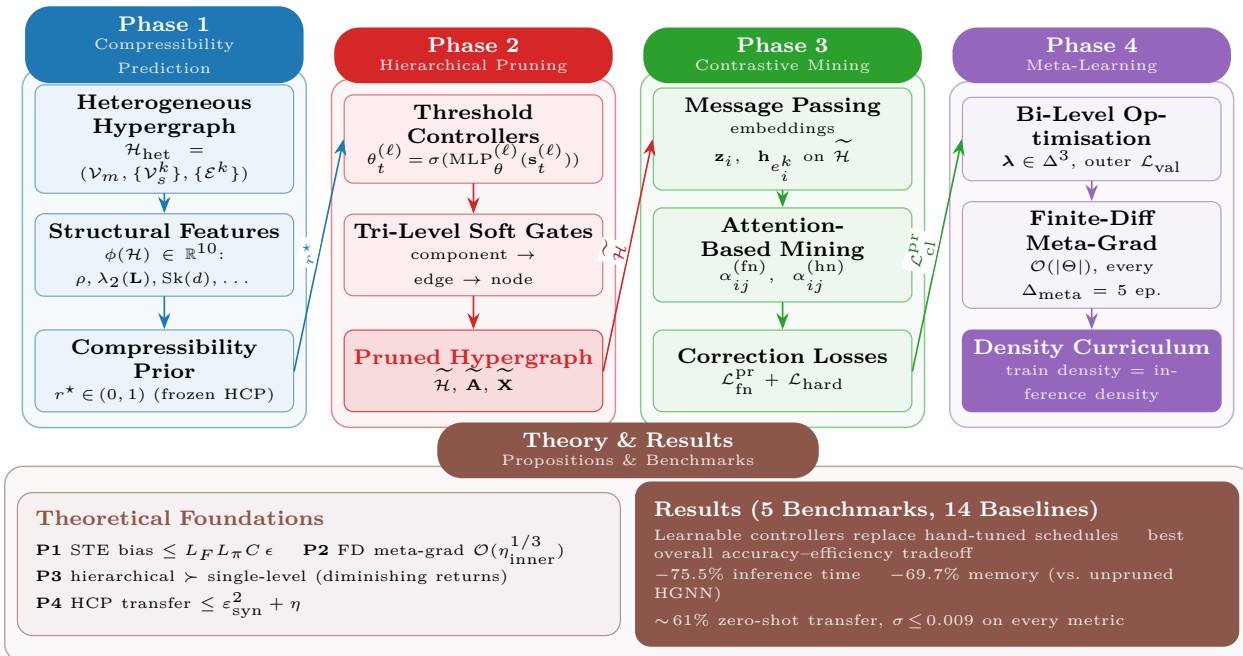

Figure 1: **Overview of the TriPrune-HGNN framework.** The four learnable mechanisms are arranged as sequential phases. **Phase 1** pre-trains the Hypergraph Compressibility Predictor (HCP) once on a synthetic corpus and, at deployment, maps graph-level statistics $\phi(\mathcal{H})$ to a domain-invariant compressibility prior $r^\star$. **Phase 2** performs Neural Adaptive Hierarchical Pruning: learnable threshold controllers, conditioned on $r^\star$ and a graph-state vector $\mathbf{s}_t^{(\ell)}$, drive differentiable soft gates across the component, edge, and node levels to produce the pruned hypergraph $\widetilde{\mathcal{H}}$. **Phase 3** runs message passing on $\widetilde{\mathcal{H}}$ and applies attention-based contrastive mining to identify and correct false and hard negatives induced by the topology change. **Phase 4** performs bi-level meta-learning of the loss weights $\boldsymbol{\lambda} \in \Delta^3$ and realises pruning as a *density curriculum*: the meta-update reshapes the Phase 2 threshold controllers across epochs, so the model trains on the same sparsity it deploys at — eliminating the train–inference distribution shift. The bottom banner summarises the four propositions and the key empirical results.

## 4 Methodology

We present TriPrune-HGNN, a unified framework that addresses the three challenges identified in Section 1 through four learnable modules: *(i)* a **Hypergraph Compressibility Predictor** (HCP) that estimates the maximum achievable compression ratio from graph-level statistics, providing a domain-invariant prior for pruning aggressiveness; *(ii)* **Neural Adaptive Hierarchical Pruning** (NAHP) with learnable threshold controllers operating jointly across components, hyperedges, and nodes; *(iii)* **Attention-Based Contrastive Learning** with neural false- and hard-negative discovery; and *(iv)* **Meta-Learned Multi-Task Optimisation** that automatically balances competing objectives. Throughout the paper we reserve the word *component* for the $K$ structural contexts of the heterogeneous hypergraph (director-context, actor-context, etc.) and use *module* for the four learnable pieces (i)–(iv) of the framework. Figure 1 illustrates the pipeline; the four modules are coupled by a *density curriculum* that progressively sparsifies the graph during training so the model adapts to its final inference sparsity, eliminating the train–inference distribution shift.

### 4.1 Hypergraph Compressibility Predictor

**Why predict compressibility before training?** Pruning-based training exposes at least three design choices whose correct values depend on the target hypergraph's structure: (i) how aggressive the initial pruning schedule should be, (ii) how fast that schedule should decay, and (iii) how strongly to regularise the schedule toward the eventual target retention. In prior work, all three are hand-tuned per dataset, which both wastes tuning budget and prevents cross-dataset transfer of the tuned values. Predicting a single

scalar $r^\star \in (0,1)$ that summarises how compressible the target hypergraph is — *before* training begins, from graph-level statistics alone — collapses these three choices into one data-driven decision that transfers across datasets. It is the reason our cross-dataset generalisation experiment (Section 5.2) is possible: the same frozen predictor works on citation, recommendation, and review hypergraphs because it conditions only on structural quantities that are comparable across those domains.

Fixed pruning schedules cannot anticipate how compressible a given hypergraph is before training begins. Dense hypergraphs with small spectral gaps tolerate aggressive compression, whereas sparse, modular hypergraphs require conservative pruning to preserve structural integrity. We introduce a *Hypergraph Compressibility Predictor* (HCP) that estimates the maximum achievable compression ratio $r^\star \in (0,1)$ directly from graph-level statistics, providing a data-dependent prior that initialises and regularises the threshold controllers. For a heterogeneous hypergraph $\mathcal{H}_{\mathrm{het}} = (\mathcal{V}_m, \{\mathcal{V}_s^k\}_{k=1}^K, \{\mathcal{E}^k\}_{k=1}^K)$, we extract a 10-dimensional structural descriptor $\phi(\mathcal{H}) \in \mathbb{R}^{10}$ aggregated across components:

$$\phi(\mathcal{H}) = \big[\, n,\; m,\; \bar{d}_e,\; \rho,\; \mathrm{Sk}(d),\; \mathrm{CV}(d),\; \lambda_2(\mathbf{L}),\; \bar{c},\; H(p_e),\; K \,\big], \tag{7}$$

where $n = |\mathcal{V}_m|$, $m = \sum_k |\mathcal{E}^k|$, $\bar{d}_e$ is the average hyperedge degree, $\rho = m/\binom{n}{2}$ is the hypergraph density, $\mathrm{Sk}(d)$ and $\mathrm{CV}(d)$ are the skewness and coefficient of variation of the node-degree distribution, $\lambda_2(\mathbf{L})$ is the second-smallest eigenvalue of the normalised hypergraph Laplacian (spectral gap), $\bar{c}$ is the average hypergraph clustering coefficient (Estrada & Rodríguez-Velázquez, 2006), $H(p_e)$ is the Shannon entropy of the hyperedge-size distribution, and $K$ is the number of components. All features are computed once per hypergraph at $\mathcal{O}(n+m)$ cost (the spectral gap uses Lanczos iteration with $\mathcal{O}(km)$ for the top-$k$ eigenvalues, $k = 10$).

**Predictor architecture.** HCP is a three-layer MLP with hidden dimensions $[64, 32, 1]$ and sigmoid output:

$$r^\star \;=\; \sigma(\mathrm{MLP}_{\mathrm{HCP}}(\phi(\mathcal{H}))), \qquad r^\star \in (0,1). \tag{8}$$

We interpret $r^\star$ as an estimate of the maximum overall retention ratio $r_{\mathrm{overall}} = r_{\mathrm{comp}} \cdot r_{\mathrm{edge}} \cdot r_{\mathrm{node}}^2$ at which accuracy degradation remains below a fixed tolerance $\Delta_{\mathrm{acc}}$ ($= 1\%$ throughout).

**Pre-training and integration.** HCP is pre-trained offline on a corpus of 5,000 synthetic hypergraphs sampled with controlled structural parameters ($\rho \in [0.01, 0.5]$, $\mathrm{Sk}(d) \in [0,5]$, $K \in [2,10]$, $n \in [1\mathrm{k}, 50\mathrm{k}]$). For each synthetic hypergraph the empirical compressibility $r^{\mathrm{emp}}$ is obtained by running NAHP at multiple retention targets and identifying the minimum ratio at which accuracy degrades by more than $\Delta_{\mathrm{acc}}$. HCP is then optimised to minimise $\mathbb{E}_{\mathcal{H}}\big[(r^\star - r^{\mathrm{emp}})^2\big]$. At deployment, HCP supplies two signals to the main framework:

1. **Threshold initialisation.** The initial thresholds $\theta_0^{(\ell)}$ are set so that $\prod_\ell \mathbb{E}[g^{(\ell)}] \approx r^\star$, removing the need to manually pick a pruning starting point.

2. **Compressibility regularisation.** A soft penalty $\mathcal{L}_{\mathrm{HCP}} = \big(r_{\mathrm{overall}}^{(t)} - r^\star\big)^2$ encourages the learned schedule to converge near the predicted compressibility. Its weight $\beta_{\mathrm{HCP}}^{(t)}$ decays linearly from $\beta_{\mathrm{HCP}}^{(0)} = 0.1$ to 0 over the first 50% of training, allowing data-driven refinement thereafter.

There is a subtle circularity in this procedure that arises from HCP and NAHP consuming each other's outputs; we address it in Section 4.2 after the NAHP components have been introduced, where the relevant terms have concrete meanings.

HCP operates entirely on graph-level structural statistics — not node features or labels — making it inherently transferable across domains. Two hypergraphs with similar $(\rho, \lambda_2(\mathbf{L}), \mathrm{Sk}(d))$ are expected to admit similar compression even if they encode different semantic content. We empirically validate this in Section 5.6: HCP trained *only* on synthetic data predicts the empirical compressibility of all five real benchmarks with aggregate RMSE 0.046 (per-benchmark absolute errors in $[0.040, 0.052]$). Section 5.2 further shows that HCP contributes roughly half of the $\sim 61\%$ cross-dataset transfer retention, confirming that compressibility is a structural property of hypergraph topology that the framework can explicitly exploit rather than an incidental empirical observation.

## 4.2 Neural Adaptive Hierarchical Pruning

Existing pruning methods rely on fixed schedules ($\theta_t = \theta_0 \cdot \alpha^t$) that require careful tuning for each dataset, with no mechanism to slow down when accuracy degrades or accelerate when convergence is safe. We propose *learnable neural threshold controllers* that automatically adjust pruning rates based on real-time graph statistics and convergence signals, and that operate jointly across three structural granularities — components (behavioural contexts), hyperedges within each retained component, and nodes across the whole graph. Distributing the pruning budget across all three granularities (rather than concentrating it at any single level) is strictly better in expectation whenever accuracy costs at each level are convex in the amount removed — an intuitive diminishing-returns assumption that we verify empirically in Figure 4 (Section 5.9). The formal statement and proof are deferred to Appendix A (Proposition 3).

**Role of message-passing.** TriPrune-HGNN is built on top of standard hypergraph message-passing (Eq. equation 2). Pruning operates on the *graph structure*, not on the message-passing equations themselves: by removing components, hyperedges, and nodes before propagation, we reduce the size of the incidence matrices on which message-passing is performed and hence its computational cost. The contrastive losses of Section 4.3 then act on the embeddings produced by message-passing on the pruned graph, correcting distortions introduced by the topology change.

**Neural threshold prediction.** Instead of manually setting thresholds $\theta_t^{(\ell)}$, we introduce a learnable predictor:

$$\theta_t^{(\ell)} \;=\; \sigma\!\left(\mathrm{MLP}_\theta^{(\ell)}\!\left(\mathbf{s}_t^{(\ell)}\right)\right), \tag{9}$$

where $\sigma(\cdot)$ is the sigmoid activation and $\mathbf{s}_t^{(\ell)} \in \mathbb{R}^7$ is the graph state vector at level $\ell$ and epoch $t$:

$$\mathbf{s}_t^{(\ell)} \;=\; \left[\; \frac{|\widetilde{\mathcal{E}}_t|}{|\mathcal{E}|},\; \mu_t^{(\mathrm{imp})},\; \sigma_t^{(\mathrm{imp})},\; \tfrac{t}{T},\; \Delta\mathcal{L}_{\mathrm{train}},\; \|\nabla\mathcal{L}\|_2,\; r^\star \;\right], \tag{10}$$

with $\Delta\mathcal{L}_{\mathrm{train}} = \mathcal{L}_{\mathrm{train}}^{(t-1)} - \mathcal{L}_{\mathrm{train}}^{(t-5)}$ measuring the training-loss trend over a 5-epoch window. The final feature $r^\star$ is the compressibility prediction from HCP (Section 4.1); it provides the controller with a domain-invariant prior on how aggressive pruning can be on the current hypergraph. The state vector $\mathbf{s}_t^{(\ell)}$ captures: *(1)* the current sparsity ratio, *(2)–(3)* the first two moments of the element-importance distribution (an efficient but not fully informative summary, as noted in Section 5.1), *(4)* training progress, *(5)* the training-loss trend, *(6)* the gradient magnitude, and *(7)* the predicted compressibility.

**On train/validation information flow.** We use $\Delta\mathcal{L}_{\mathrm{train}}$ rather than validation loss inside the state vector to ensure the threshold predictor itself is not directly conditioned on validation signal at each epoch. This is a property of the threshold-controller input, not a claim of complete train/validation isolation for the framework as a whole: the meta-learning step (Section 4.4) updates the loss weights $\boldsymbol{\lambda}$ using the validation loss $\mathcal{L}_{\mathrm{val}}$, as is standard in bi-level optimisation (Franceschi et al., 2018). The threshold controller is implemented as a 2-layer MLP:

$$\mathrm{MLP}_\theta^{(\ell)}(\mathbf{s}) \;=\; \mathbf{W}_2^{(\ell)}\, \sigma\!\big(\mathbf{W}_1^{(\ell)}\mathbf{s} + \mathbf{b}_1^{(\ell)}\big) + \mathbf{b}_2^{(\ell)}, \tag{11}$$

with hidden dimension $d_{\mathrm{hidden}} = 32$ and ReLU activation.

**Differentiable soft gating.** At each pruning level $\ell$ we compute importance scores $\pi^{(\ell)} \in [0, 1]$ and apply differentiable soft gates using the learned thresholds:

$$g^{(\ell)} \;=\; \sigma_{\mathrm{hard}}\!\left(\frac{\pi^{(\ell)} - \theta_t^{(\ell)}}{\epsilon}\right), \qquad \sigma_{\mathrm{hard}}(x) = \max\!\big(0, \min(1, x + 0.5)\big), \tag{12}$$

where $\sigma_{\mathrm{hard}}$ is the piecewise-linear hard sigmoid that provides straight-through gradient estimates during back- propagation (Bengio et al., 2013). The smoothing parameter $\epsilon$ controls the gate's steepness; the resulting bias of the straight-through estimator is bounded by $\mathcal{O}(\epsilon L_\pi)$ (Appendix A, Proposition 1), which directly motivates our choice $\epsilon = 0.01$ and is empirically validated in Section 5.1.

During training, $g^{(\ell)} \in [0,1]$ is continuous, allowing gradient-based threshold optimisation. By convergence (epoch $\approx 150$), the model therefore trains on, and adapts to, the final sparsity level for the last $\sim 25\%$ of training. Only after training are gates binarised: elements with $g^{(\ell)} > 0.5$ are retained in the deployed pruned graph. Inference operates at the same sparsity level the model trained on in its final phase — there is no density shift between training and inference. This clean separation between differentiable training and discrete inference is standard in learned-mask pruning (Louizos et al., 2017).

**Component-level pruning.** We score each component $k$ with learnable importance balancing:

$$\pi_k^{(\mathrm{comp})} = \mathrm{softmax}_k\Big(\beta_k \cdot \|\mathbf{W}_k^{(L)}\|_F \cdot \|\mathrm{MLP}(\mathbf{X}_k)\|_2 + (1 - \beta_k) \cdot S_{\mathrm{att}}(k)\Big), \tag{13}$$

where $\mathbf{W}_k^{(L)} \in \mathbb{R}^{d \times d}$ is the *weight matrix of the final HGNN layer for component $k$* (the layer whose output feeds into classification), $\|\mathbf{W}_k^{(L)}\|_F$ is its Frobenius norm, $\|\mathrm{MLP}(\mathbf{X}_k)\|_2$ captures feature-activation strength, and $S_{\mathrm{att}}(k)$ is the *average attention coefficient across all attention heads in component $k$*, defined explicitly as

$$S_{\mathrm{att}}(k) = \frac{1}{|\mathcal{E}^k|} \sum_{(i,j) \in \mathcal{E}^k} a_{ij}^k, \tag{14}$$

with $a_{ij}^k$ the normalised attention weights output by component $k$'s self-attention mechanism. The balance coefficient $\beta_k$ is component-specific and learnable:

$$\beta_k = \sigma\Big(\mathrm{MLP}_\beta\big([\|\mathbf{W}_k^{(L)}\|_F, \ S_{\mathrm{att}}(k), \ \mathrm{Var}(\mathbf{X}_k)]\big)\Big). \tag{15}$$

Components with $g_k^{(\mathrm{comp})} > 0.5$ are retained, with a safety constraint ensuring at least $K_{\min} = \max(1, \lfloor 0.1K \rfloor)$ active components.

**Edge-level pruning.** For each hyperedge $(i,j)$ in a retained component $k$:

$$\pi_{ij}^{(\mathrm{edge})} = \mathrm{softmax}_{(i,j) \in \mathcal{E}^k}\big(\mathbf{A}_{ij} \cdot \cos(\mathbf{x}_i, \mathbf{x}_j)\big), \tag{16}$$

and the pruned incidence matrix applies both component and edge gates:

$$(\widetilde{\mathbf{H}}_k)_{ij} = (\mathbf{H}_k)_{ij} \cdot g_k^{(\mathrm{comp})} \cdot g_{ij}^{(\mathrm{edge})}. \tag{17}$$

We enforce that every retained component keeps at least one edge to prevent complete disconnection.

**Node-level pruning.** We score nodes by their connectivity strength in the already-pruned graph and their feature magnitude:

$$\pi_i^{(\mathrm{node})} = \mathrm{softmax}_{i \in \mathcal{V}_k}\big(\bar{s}_i \cdot \|\mathbf{x}_i\|_2\big), \quad \bar{s}_i = \frac{1}{|\mathcal{N}_i^{(\mathrm{pruned})}|} \sum_{j \in \mathcal{N}_i^{(\mathrm{pruned})}} \cos(\mathbf{x}_i, \mathbf{x}_j). \tag{18}$$

The pruned feature matrix is

$$(\widetilde{\mathbf{X}}_k)_i = (\mathbf{X}_k)_i \cdot g_k^{(\mathrm{comp})} \cdot \mathbb{I}\big[|\mathcal{N}_i^{(\mathrm{pruned})}| > 0\big] \cdot g_i^{(\mathrm{node})}, \tag{19}$$

where $\mathbb{I}[\cdot]$ removes isolated nodes. Each component retains at least $\max(2, \lfloor 0.05|\mathcal{V}_k| \rfloor)$ nodes. With learned retention ratios $r_{\mathrm{comp}}, r_{\mathrm{edge}}, r_{\mathrm{node}} \in (0,1)$, TriPrune-HGNN reduces the base complexity $\mathcal{O}(Knm\bar{d}_e d)$ to

$$\mathcal{O}\big(r_{\mathrm{comp}} \cdot r_{\mathrm{edge}} \cdot r_{\mathrm{node}}^2 \cdot Knm\bar{d}_e d + Kd_{\mathrm{state}}d_{\mathrm{hidden}} + d_{\mathrm{HCP}}\big). \tag{20}$$

For typical settings ($K = 5$, $d_{\mathrm{state}} = 7$, $d_{\mathrm{hidden}} = 32$, $d_{\mathrm{HCP}} \approx 2{,}800$ MLP parameters), the controller and HCP overhead is negligible compared to the $\mathcal{O}(10^6)$ scale of hypergraph message-passing on real-world datasets.

**Avoiding the HCP–NAHP bootstrap circularity.** We can now revisit the circularity flagged at the end of Section 4.1. The problem is concrete: HCP is trained to predict $r^\star$ that matches $r^{\mathrm{emp}}$, the retention ratio at which NAHP starts losing accuracy. But NAHP's threshold controllers are initialised from HCP's $r^\star$

(Section 4.1, item 1), and the graph-state vector $\mathbf{s}_t^{(\ell)}$ of Eq. equation 10 lists $r^\star$ as its seventh feature. If we compute $r^{\mathrm{emp}}$ by running the fully HCP-conditioned NAHP, we are letting HCP's own prediction shape the target it is trained to match — a self-fulfilling loop. To eliminate this circularity, we generate the synthetic training corpus with a *decoupled* variant of NAHP that consumes no HCP signal: *(i)* threshold controllers are initialised to the dataset-agnostic default $\theta_0^{(\ell)} = 0.5$ (corresponding to $r_{\mathrm{overall}} \approx 0.125$ at epoch 0), and *(ii)* the $r^\star$ slot of $\mathbf{s}_t^{(\ell)}$ is replaced by the constant 0.5 throughout. Under this protocol, $r^{\mathrm{emp}}$ depends only on the hypergraph $\mathcal{H}$ and the NAHP architecture; no HCP signal flows back into the training target. To verify that this decoupling does not itself bias what HCP learns, we ran a one-round self-consistency check: after training HCP on the decoupled corpus, we regenerated $r^{\mathrm{emp}}$ using the full HCP-conditioned NAHP and compared the two. The two compressibility estimates differ by RMSE 0.013 across the 5,000 synthetic graphs — well below the synthetic-corpus prediction error ($\varepsilon_{\mathrm{synth}} = 0.022$, Section 5.6) — confirming that HCP initialisation does not materially shift the compressibility frontier and that one round of training is sufficient. Total HCP pre-training cost is approximately 9.4 GPU-hours on a single NVIDIA A100 for the full 5,000-graph corpus (the same figure used in the amortisation analysis of Section 5); the trained weights and the generator are released as anonymised supplementary material.

### 4.3 Attention-Based Contrastive Learning

Sections 4.1 and 4.2 describe *how much* to prune and *what* to prune. This section addresses a separate problem: how to *learn representations* on the pruned graph. The concern is specific to contrastive HGNN training. Contrastive objectives (Eq. equation 4) push together the embeddings of nodes that share a hyperedge and push apart the embeddings of nodes that do not. Pruning changes the set of nodes that share a hyperedge, and both directions of the objective inherit that change. Two distortions arise, and neither is corrected by the pruning stage alone. Below we list each distortion, why it is not a hypothetical concern (it appears in every pruning run, quantifiable pre-hoc), and what mechanism corrects it.

Pruning fundamentally alters graph topology, creating two types of problematic node pairs for contrastive learning.

**(a) False negatives** arise when semantically similar nodes become disconnected after pruning — e.g., two users who purchased similar products but whose shared hyperedge was removed. Standard contrastive loss incorrectly pushes such pairs apart.

**(b) Hard negatives** arise when dissimilar nodes acquire *spuriously similar embeddings* after pruning. Pruning does not create new structural paths — it can only remove existing ones. The spurious similarity is therefore an *embedding-space artefact*: when a shared hyperedge between two semantically dissimilar nodes is removed, their representations may nevertheless converge in subsequent message-passing iterations because they share many *retained* common neighbours. This is a topology-induced effect on embeddings, not on graph paths. Existing methods use fixed similarity thresholds (e.g., $\cos(\mathbf{z}_i, \mathbf{z}_j) > 0.7$) to identify such pairs, but optimal thresholds vary across datasets and training phases. We introduce *learnable attention mechanisms* that automatically discover false and hard negatives from node embeddings, structural changes, and component context.

**Hyperedge embeddings.** Throughout this section, the hyperedge embedding $\mathbf{e}_i^k \in \mathbb{R}^d$ for node $i$ in component $k$ is defined as the mean of the node embeddings of the (unique) hyperedge $e_i^k \in \mathcal{E}^k$ that contains $i$ in the pruned graph:

$$\mathbf{e}_i^k = \frac{1}{|e_i^k|} \sum_{j \in e_i^k} \mathbf{z}_j, \tag{21}$$

where $|e_i^k|$ is the size of that hyperedge. The hyperedge-context embedding used by the hard-negative attention head in Eq. equation 27 is identical, ensuring consistent notation throughout.

**Component-weighted contrastive loss.**

$$\mathcal{L}_{\mathrm{cl}}^{\mathrm{pr}} = -\sum_{k=1}^{K} \pi_k^{(\mathrm{retain})} \sum_{i \in \mathcal{V}_m} \log \frac{\exp\big(s(\mathbf{z}_i, \widetilde{\mathbf{h}}_{e_i^k})/\tau_i\big)}{\sum_{j \in \mathcal{V}_m} \exp\big(s(\mathbf{z}_i, \widetilde{\mathbf{h}}_{e_j^k})/\tau_i\big)}, \tag{22}$$

where $s(\cdot,\cdot)$ is cosine similarity, $\widetilde{\mathbf{h}}_{e_i^k}$ is the hyperedge embedding in the pruned graph (Eq. equation 21), and $\tau_i$ is a node-specific learnable temperature

$$\tau_i \;=\; \tau_0 \cdot \exp\!\big(\mathrm{MLP}_\tau([\deg(i), \|\mathbf{x}_i\|_2, \bar{s}_i])\big). \tag{23}$$

The retention weight $\pi_k^{(\mathrm{retain})}$ down-weights components that experienced severe topology disruption:

$$\pi_k^{(\mathrm{retain})} \;=\; \frac{|\widetilde{\mathbf{H}}_k|_0}{|\mathbf{H}_k|_0} \cdot \exp\!\left(-\alpha \frac{\|\mathbf{A}_k - \widetilde{\mathbf{A}}_k\|_F}{\|\mathbf{A}_k\|_F}\right). \tag{24}$$

**Attention-based false-negative discovery.**

$$\alpha_{ij}^{(\mathrm{fn})} \;=\; \sigma\!\Big(\mathbf{w}_{\mathrm{fn}}^\top \tanh\!\big(\mathbf{W}_{\mathrm{fn}}[\mathbf{z}_i\|\mathbf{z}_j\|\Delta\mathbf{A}_{ij}\|\mathbf{c}_k]\big)\Big), \tag{25}$$

with $\Delta\mathbf{A}_{ij} = (\mathbf{A}_k)_{ij} - (\widetilde{\mathbf{A}}_k)_{ij} \in \{0,1\}$ indicating whether edge $(i,j)$ was pruned, and $\mathbf{c}_k \in \mathbb{R}^d$ a learnable component embedding. The false-negative set is $\mathcal{F}_k^{\mathrm{pr}} = \{(i,j)\colon \alpha_{ij}^{(\mathrm{fn})} > 0.5, (\widetilde{\mathbf{A}}_k)_{ij} = 0\}$, and the correction loss is

$$\mathcal{L}_{\mathrm{fn}}^{\mathrm{pr}} \;=\; -\sum_{k=1}^{K} \sum_{(i,j)\in\mathcal{F}_k^{\mathrm{pr}}} \alpha_{ij}^{(\mathrm{fn})} \cdot \log \frac{\exp\!\big(s(\mathbf{z}_i,\mathbf{z}_j)/\tau_i\big)}{\sum_{l\in\mathcal{N}_i^{(\mathrm{aug})}} \exp\!\big(s(\mathbf{z}_i,\mathbf{z}_l)/\tau_i\big)}. \tag{26}$$

**Attention-based hard-negative discovery.**

$$\alpha_{ij}^{(\mathrm{hn})} \;=\; \mathrm{Attention}_{\mathrm{hn}}\big([\mathbf{z}_i, \mathbf{z}_j, \mathbf{e}_i^k, \mathbf{e}_j^k]\big), \tag{27}$$

with hyperedge embeddings $\mathbf{e}_i^k, \mathbf{e}_j^k$ as defined in Eq. equation 21. The hard-negative set is $\mathcal{H}_i^k = \{j\colon \alpha_{ij}^{(\mathrm{hn})} > 0.5\}$, and the reweighting loss is

$$\mathcal{L}_{\mathrm{hard}} \;=\; -\sum_{k=1}^{K} \sum_{i\in\mathcal{V}_m} \sum_{j\in\mathcal{H}_i^k} \pi_k^{(\mathrm{retain})} \cdot \alpha_{ij}^{(\mathrm{hn})} \cdot \log \frac{\exp\!\big(-s(\mathbf{z}_i,\mathbf{z}_j)/\tau_i\big)}{\sum_{l\in\mathcal{H}_i^k} \exp\!\big(-s(\mathbf{z}_i,\mathbf{z}_l)/\tau_i\big)}. \tag{28}$$

### 4.4 Meta-Learned Multi-Task Optimisation

Balancing classification, contrastive, false-negative, and hard-negative losses requires careful weight tuning. We use gradient-based meta-learning to discover optimal weights by treating weight selection as an outer optimisation problem.

**Bi-level formulation.** With $\boldsymbol{\lambda} = [\lambda_1, \lambda_2, \lambda_3, \lambda_4]^\top \in \Delta^3$,

$$\min_{\boldsymbol{\lambda}} \; \mathcal{L}_{\mathrm{val}}\big(\boldsymbol{\Theta}^*(\boldsymbol{\lambda})\big) \quad \text{s.t.} \quad \boldsymbol{\Theta}^*(\boldsymbol{\lambda}) = \arg\min_{\boldsymbol{\Theta}} \mathcal{L}_{\mathrm{train}}(\boldsymbol{\Theta}; \boldsymbol{\lambda}). \tag{29}$$

At each meta-step $t$ (every $\Delta_{\mathrm{meta}} = 5$ epochs) we perform one inner training step $\boldsymbol{\Theta}^* = \boldsymbol{\Theta} - \eta_{\mathrm{inner}}\nabla_{\boldsymbol{\Theta}}\mathcal{L}_{\mathrm{total}}$, an outer meta-update $\boldsymbol{\lambda}^{(t+1)} = \boldsymbol{\lambda}^{(t)} - \eta_{\mathrm{meta}}\nabla_{\boldsymbol{\lambda}}\mathcal{L}_{\mathrm{val}}(\boldsymbol{\Theta}^*; \boldsymbol{\lambda}^{(t)})$, and a simplex projection $\boldsymbol{\lambda}^{(t+1)} \leftarrow \mathrm{Proj}_\Delta(\boldsymbol{\lambda}^{(t+1)})$ following Duchi et al. (2008). The total training loss is

$$\mathcal{L}_{\mathrm{total}} \;=\; \lambda_1\mathcal{L}_{\mathrm{cls}} + \lambda_2\mathcal{L}_{\mathrm{cl}}^{\mathrm{pr}} + \lambda_3\mathcal{L}_{\mathrm{fn}}^{\mathrm{pr}} + \lambda_4\mathcal{L}_{\mathrm{hard}} + \beta_{\mathrm{HCP}}^{(t)}\mathcal{L}_{\mathrm{HCP}}. \tag{30}$$

The Hessian-vector product in the meta-gradient is approximated by central finite differences with perturbation $\delta = 10^{-5}$, giving $\mathcal{O}(|\boldsymbol{\Theta}|)$ cost. The bias of this estimate is bounded by $\mathcal{O}(\eta_{\mathrm{inner}}^{1/3})$ under the local regularity conditions stated in Appendix A.2 (Proposition 2); the optimal perturbation $\delta^* = \Theta(\eta_{\mathrm{inner}}^{-1/3})$ balances the truncation and evaluation terms in Eq. equation 33.

### 4.5 Density Curriculum Interpretation

A natural concern with any progressive-pruning scheme is whether the model is trained on a graph distribution that differs from the one it ultimately operates on at inference. We address this concern directly by interpreting the pruning trajectory of TriPrune-HGNN as a *density curriculum*: a continuous sequence of graphs $\{\mathcal{H}_t\}_{t=1}^T$ in which $\mathcal{H}_1$ is the original dense graph, $\mathcal{H}_T$ is the converged pruned graph, and each intermediate $\mathcal{H}_t$ differs from $\mathcal{H}_{t+1}$ by an infinitesimal change in retention.

**Three properties of the curriculum.** *(i) Continuity.* Differentiable soft gates (Eq. equation 12) ensure that the retention level changes smoothly between consecutive epochs; there is no discrete switch between two regimes. *(ii) Convergence to a fixed density.* Retention ratios stabilise by epoch $\sim 150$; for the last $\sim 25\%$ of training, the model sees, and adapts to, the same sparsity level that it will operate at during inference. *(iii) Curriculum monotonicity.* The compressibility regulariser of HCP (Section 4.1) anchors the curriculum toward a target $r^\star$ predicted from graph statistics, preventing premature collapse or oscillation.

Under properties (i)–(ii), inference operates at the same retention level the model has trained on during its final phase. The "dense-to-sparse" aspect of training is therefore a curriculum *within* training, not a distribution shift *between* training and inference. Section 5.8 provides direct empirical evidence: *(a)* a cold-start experiment, where the model is trained from scratch on the final pruned graph, recovers 84%–93% of the curriculum-trained accuracy across the five benchmarks (mean 88.6%), confirming that the dense-to-sparse trajectory carries useful inductive bias; *(b)* a $4 \times 4$ train-density vs. test-density grid shows that the curriculum-trained model is robust within $\pm 1.6\%$ MAE for test densities matching its training trajectory and degrades smoothly outside it; and *(c)* a schedule-shuffling ablation that reorders the threshold trajectory degrades final accuracy by 2.3%, confirming that the *monotonic* dense-to-sparse direction is itself a useful inductive bias rather than an artefact of the final state. We interpret these results, together with HCP's transferability, as evidence that what TriPrune-HGNN learns is not merely a dataset-specific pruning schedule but an instance of a more general principle: *the optimal sparsification of a hypergraph proceeds by progressively removing structurally redundant content in an order that preserves message-passing fidelity at each step.* This is the "phenomenon" our framework exposes; the individual neural modules are tools that operationalise it. Algorithm 1 summarises the full training procedure including HCP, NAHP, attention-based contrastive mining, and meta-learning.

The design choices underlying TriPrune-HGNN's three principal mechanisms — the gate-smoothing parameter $\epsilon$, the finite-difference perturbation $\delta$ for the meta-gradient, and the three-level pruning hierarchy — are supported by three short design-motivation propositions (Propositions 1, 2, and 3 in Appendix A), plus an informal transfer-bound analysis for the compressibility predictor (Proposition 4). A fifth result (Proposition 5) establishes *local approximate stationarity* of the bi-level meta-update on the loss-weight simplex, under locally testable assumptions consistent with our reported training diagnostics. These statements clarify the local assumptions under which each design choice is justified and bound the cost of those assumptions failing; they are local guarantees, not end-to-end convergence or generalisation guarantees for the full framework, and we make this scope explicit in the limitations box opening Appendix A.

## 5 Experimental Evaluation

We evaluate methods across three categories with different optimisation objectives: *(1) Standard HGNNs* maximise predictive accuracy without compression constraints; *(2) Pruning and distillation methods* target the accuracy–efficiency trade-off under resource budgets. Our primary comparison of interest is against efficient baselines (pruning and distillation) with comparable memory footprints; improvements over standard HGNNs are a secondary baseline. Accordingly, Table 2 reports predictive metrics alongside end-to-end time and memory cost, situating all methods in accuracy–efficiency space.

**Datasets.** We evaluate on five hypergraph benchmarks: **IMDB** (Singh & Gordon, 2008) (movie ratings with actor/director contexts), **DBLP** (Tang et al., 2008) (author–paper–venue citations), **Yelp** (Asghar, 2016) (business reviews with category/location contexts), **Amazon** (McAuley et al., 2015) (product ratings with brand/category contexts), and **Douban** (Zheng et al., 2021) (movie ratings with genre/director contexts).

---

**Algorithm 1** TriPrune-HGNN with Compressibility-Predicted Density Curriculum

---

**Require:** Hypergraph $\mathcal{H} = (\mathcal{V}, \mathcal{E})$, features $\mathbf{X}$, pre-trained HCP, hyperparameters $\{\tau_0, \epsilon, \alpha, \eta_{\mathrm{inner}}, \eta_{\mathrm{meta}}\}$
**Ensure:** Pruned graph $\widetilde{\mathcal{H}}$, embeddings $\mathbf{Z}^*$, learned weights $\boldsymbol{\lambda}^*$

1: **[Stage 0: Compressibility prediction]**
2: $\phi(\mathcal{H}) \leftarrow$ structural features of $\mathcal{H}$ (Eq. equation 7)
3: $r^\star \leftarrow \sigma(\mathrm{MLP}_{\mathrm{HCP}}(\phi(\mathcal{H})))$
4: Initialise $\theta_0^{(\ell)}$ so that $\prod_\ell \mathbb{E}[g^{(\ell)}] \approx r^\star$
5: Initialise $\boldsymbol{\lambda} \leftarrow [\frac{1}{4}, \ldots, \frac{1}{4}]$, model parameters $\boldsymbol{\Theta}$, neural modules $\{\mathrm{MLP}_\theta^{(\ell)}, \mathrm{MLP}_\beta, \mathrm{MLP}_\tau, \mathrm{Attn}_{\mathrm{fn}}, \mathrm{Attn}_{\mathrm{hn}}\}$
6: **for** epoch $t = 1$ to $T$ **do**
7:     **[Stage 1: Threshold prediction]**
8:     **for** $\ell \in \{\mathrm{comp}, \mathrm{edge}, \mathrm{node}\}$ **do**
9:         $\mathbf{s}_t^{(\ell)} \leftarrow [|\widetilde{\mathcal{E}}_t|/|\mathcal{E}|, \mu_t^{(\mathrm{imp})}, \sigma_t^{(\mathrm{imp})}, t/T, \Delta\mathcal{L}_{\mathrm{train}}, \|\nabla\mathcal{L}\|_2, r^\star]$
10:         $\theta_t^{(\ell)} \leftarrow \sigma(\mathrm{MLP}_\theta^{(\ell)}(\mathbf{s}_t^{(\ell)}))$
11:     **end for**
12:     **[Stage 2: Hierarchical pruning]**
13:     **for** $k = 1, \ldots, K$ **do**
14:         $\beta_k \leftarrow \sigma(\mathrm{MLP}_\beta([\|\mathbf{W}_k^{(L)}\|_F, S_{\mathrm{att}}(k), \mathrm{Var}(\mathbf{X}_k)]))$
15:         Compute $\pi_k^{\mathrm{comp}}, g_k^{\mathrm{comp}}$ (Eqs. equation 12, equation 13)
16:         **if** $g_k^{\mathrm{comp}} > 0.5$ **then**
17:             Compute $\pi_{ij}^{\mathrm{edge}}, g_{ij}^{\mathrm{edge}}, \pi_i^{\mathrm{node}}, g_i^{\mathrm{node}}$ (Eqs. equation 16–equation 18)
18:             Update $\widetilde{\mathbf{H}}_k, \widetilde{\mathbf{X}}_k$ (Eqs. equation 17, equation 19)
19:         **end if**
20:     **end for**
21:     **[Stage 3: Contrastive mining]**
22:     Compute retention weights $\pi_k^{\mathrm{retain}}$ per component (Eq. equation 24) and node-adaptive temperatures $\tau_i$ from local retention density (Eq. equation 23)
23:     Compute false-negative attention scores $\alpha_{ij}^{\mathrm{fn}} \leftarrow \mathrm{Attn}_{\mathrm{fn}}(\mathbf{z}_i, \mathbf{z}_j)$ and hard-negative attention scores $\alpha_{ij}^{\mathrm{hn}} \leftarrow \mathrm{Attn}_{\mathrm{hn}}(\mathbf{z}_i, \mathbf{z}_j)$ from Eqs. equation 25–equation 27
24:     Assemble the false-negative pair set $\mathcal{F}_k^{\mathrm{pr}}$ and hard-negative sets $\mathcal{H}_i^k$ by thresholding at $\alpha > 0.5$
25:     **[Stage 4: Loss computation]**
26:     Compute the four component losses: classification $\mathcal{L}_{\mathrm{cls}}$ (Eq. equation 5), pruned contrastive $\mathcal{L}_{\mathrm{cl}}^{\mathrm{pr}}$ (Eq. equation 22), false-negative correction $\mathcal{L}_{\mathrm{fn}}^{\mathrm{pr}}$ (Eq. equation 26), and hard-negative correction $\mathcal{L}_{\mathrm{hard}}$ (Eq. equation 28)
27:     Compute the HCP compressibility regulariser $\mathcal{L}_{\mathrm{HCP}} \leftarrow (r_{\mathrm{overall}}^{(t)} - r^\star)^2$ with decaying weight $\beta_{\mathrm{HCP}}^{(t)}$
28:     Combine into $\mathcal{L}_{\mathrm{total}}$ (Eq. equation 30) weighted by the current loss weights $\boldsymbol{\lambda}$
29:     **[Stage 5: Meta-learning update]**
30:     $\boldsymbol{\Theta}^* \leftarrow \boldsymbol{\Theta} - \eta_{\mathrm{inner}}\nabla_{\boldsymbol{\Theta}}\mathcal{L}_{\mathrm{total}}$
31:     **if** $t \bmod \Delta_{\mathrm{meta}} = 0$ **then**
32:         $\boldsymbol{\lambda} \leftarrow \mathrm{Proj}_\Delta(\boldsymbol{\lambda} - \eta_{\mathrm{meta}}\nabla_{\boldsymbol{\lambda}}\mathcal{L}_{\mathrm{val}}(\boldsymbol{\Theta}^*))$
33:     **end if**
34:     Update $\boldsymbol{\Theta}$ and all neural modules
35: **end for**
36: **return** $\widetilde{\mathcal{H}}, \mathbf{Z}^*, \boldsymbol{\lambda}^*$

---

All datasets are used in their heterogeneous hypergraph formulation following Kim et al. (2024); they span 10K–100K nodes and 50K–500K hyperedges.

**Scope of the heterogeneous-hypergraph formulation.** All five benchmarks are processed under the *one-hyperedge-per-node-per-component* formulation introduced in Section 3 (Eq. equation 4): each primary node belongs to exactly one hyperedge per behavioural context. This is a substantive restriction relative to the general heterogeneous-hypergraph construction in which a user can belong to many hyperedges per context (e.g., one per business reviewed, or one per product purchased). The simplification cleanly aligns the contrastive objective with a softmax over per-component hyperedge assignments and is consistent with the

formulation of Kim et al. (2024), but it limits the directness with which our results transfer to multi-hyperedge constructions; the multi-hyperedge generalisation is left for future work and we make the restriction explicit here, before the results, rather than burying it in the model section. The number of behavioural contexts $K$ used per dataset is small in this formulation: $K=2$ for IMDB, Yelp, Amazon, and Douban, and $K=3$ for DBLP (author, paper, venue). The complexity claim $\mathcal{O}(Knm\bar{d}_e d)$ of Section 3 should therefore be read with $K \in \{2, 3\}$ on these benchmarks and with $K=5$ only as an illustrative "larger-$K$" setting.

**Baselines.** We compare against 14 methods across three categories. *Standard HGNNs:* HGNN (Feng et al., 2019), HHGSA (Khan et al., 2025), HyGCL-AdT (Qian et al., 2024), HEAL (Ju et al., 2024), TriCL (Lee & Shin, 2023). *Pruning methods:* HSC (Lin et al., 2024), SAVIT‡ (Zheng et al., 2022a), HSL (Cai et al., 2022), HGNN-Struct‡ (Jiang et al., 2023), AdaGLT (Zhang et al., 2024), Shaver (Tran et al., 2025). *Distillation methods:* LightHGNN (Feng et al., 2024), DistillHGNN (Forouzandeh et al., 2025a), SHARP-Distill (Forouzandeh et al., 2025b). The dagger (‡) marks pruning methods originally proposed for non-hypergraph domains (Vision Transformers and 3D surface reconstruction); their adaptation protocol and tuning budget are described below and in full detail in 5.5, and they are included to broaden the comparison rather than as hypergraph-tuned competitors.

**Adaptation of cross-domain pruning baselines.** SAVIT‡ originally prunes Vision-Transformer components using a collaborative attention-and-gradient importance score; we apply the same score formulation to hyperedges by treating each hyperedge as the analogue of a ViT token and replacing the patch-attention term with the hyperedge-attention coefficient $a_{ij}^k$ of Eq. equation 14, retaining the original gradient-magnitude term unchanged. HGNN-Struct‡ originally scores 3D primitives by a structural-coherence criterion based on local geometric statistics; we instantiate this criterion on hyperedges by replacing the geometric local statistics with the hyperedge-local clustering coefficient and the spectral contribution from the normalised hypergraph Laplacian, both of which are already computed for HCP (Eq. equation 36). In both cases the *form* of the importance score is preserved from the original paper; only the input features are re-instantiated on the hypergraph.

**Tuning budget.** TriPrune-HGNN exposes a compact set of method-specific hyperparameters $(\tau_0, \epsilon, \alpha, \eta_{\mathrm{meta}}, \Delta_{\mathrm{meta}})$; each was selected by a single coarse sweep over an order-of-magnitude range on a held-out IMDB validation split (the same range later reported in Fig. 5), giving $\sim 25$ total configurations evaluated. For every baseline we used the hyperparameter values reported by its authors where applicable and ran an equally sized coarse sweep ($\sim 25$ configurations) over each baseline's method-specific hyperparameters on the same IMDB split, choosing the configuration with the lowest validation MAE before reporting on the other four benchmarks unchanged. This applies uniformly to the hypergraph-native baselines and to the two cross-domain methods marked ‡, so no method received a larger tuning budget than TriPrune-HGNN. Standard deep-learning hyperparameters (learning rate, batch size, epochs) are shared across all methods and not separately tuned.

**Implementation and protocol.** TriPrune-HGNN uses 2-layer MLPs ($d_{\mathrm{hidden}}{=}32$) for all neural controllers and 64-dimensional single-head attention networks. Meta-learning uses $\eta_{\mathrm{inner}}{=}10^{-3}$, $\eta_{\mathrm{meta}}{=}10^{-2}$, $\Delta_{\mathrm{meta}}{=}5$; base temperature $\tau_0{=}0.5$, gate smoothing $\epsilon{=}0.01$, topology weight $\alpha{=}1.0$. All methods train for 200 epochs (Adam, batch size 512, patience 20). We report $\mathrm{mean}_{\pm\mathrm{std}}$ over 10 runs with seeds 0–9. To support reproducibility, the following artifacts are released with this submission as anonymised supplementary material: *(i)* the full TriPrune-HGNN implementation (PyTorch); *(ii)* pre-trained HCP weights together with the synthetic hypergraph generator and the decoupled $r^{\mathrm{emp}}$ pipeline used to produce them (Section 4.1); *(iii)* the adaptation code for the two cross-domain baselines marked ‡ (Section 5.5); and *(iv)* seed-by-seed per-cell numbers underlying Tables 2 and 17, the cold-start curves of Figure 3, and the HCP-validation panels of Figure 2. Baselines marked ‡ inherit their pruning philosophy from non-hypergraph domains and are therefore expected to be weaker than hypergraph-native methods.

**Training compute.** All experiments are run on a single NVIDIA V100 GPU (32 GB). Full TriPrune-HGNN training takes 4.2–7.8 GPU-hours per dataset (IMDB: 4.2h, DBLP: 5.1h, Yelp: 7.8h, Amazon: 6.9h, Douban: 4.8h), of which the bi-level meta-learning loop accounts for $\sim 12\%$ (finite-difference meta-gradient every $\Delta_{\mathrm{meta}}{=}5$ epochs at $\mathcal{O}(|\Theta|)$ cost per update). HCP pre-training on the 5,000-graph synthetic corpus is a one-time 9.4 GPU-hour cost *amortised across all five benchmarks* (the same frozen HCP weights

are reused unchanged). For comparison, the strongest distillation baseline (SHARP-Distill) consumes 3.8–6.2 GPU-hours per dataset due to its two-stage teacher–student schedule; the strongest pruning baseline (AdaGLT) consumes 2.9–5.4 GPU-hours. TriPrune-HGNN's training overhead is therefore $\sim 30$–$45\%$ above the strongest baselines on a per-dataset basis (decreasing to $\sim 18$–$25\%$ once the amortised HCP cost is spread over the five-dataset suite), exchanged for the inference-time advantages reported in Table 2. We view this as a favourable trade-off for production settings where training is amortised across many deployments.

Table 2: **TriPrune-HGNN versus 14 baselines across five hypergraph benchmarks: predictive accuracy and end-to-end cost.** Results are mean$_{\pm \text{std}}$ over 10 independent runs. Lower MAE/RMSE/Time/Memory and higher ACC are better. **Best** in bold; second-best underlined (both entries underlined where two methods share the second-best value to three decimals, as on Douban MAE). ‡ denotes pruning methods adapted from non-hypergraph domains. Avg rows aggregate predictive metrics across the five datasets; Avg Time and Avg Memory are inference-side measurements averaged across datasets.

| Metric | Standard HGNN | | | | | Pruning-Based | | | | | | Distillation | | | Ours |
|---|---|---|---|---|---|---|---|---|---|---|---|---|---|---|---|
| | HGNN | HHGSA | HyGCL | HEAL | TriCL | HSC | SAVIT‡ | HSL | HGNN-S‡ | AdaGLT | Shaver | LightH | DistillH | SHARP-D | TriPrune |
| **IMDB** | | | | | | | | | | | | | | | |
| MAE | .482$_{\pm.012}$ | .461$_{\pm.011}$ | .453$_{\pm.010}$ | .446$_{\pm.009}$ | .441$_{\pm.010}$ | .402$_{\pm.008}$ | .398$_{\pm.007}$ | .405$_{\pm.008}$ | .395$_{\pm.007}$ | .387$_{\pm.007}$ | .392$_{\pm.009}$ | .408$_{\pm.012}$ | .401$_{\pm.010}$ | .396$_{\pm.009}$ | **.383**$_{\pm.007}$ |
| RMSE | .623$_{\pm.014}$ | .607$_{\pm.013}$ | .592$_{\pm.012}$ | .577$_{\pm.011}$ | .571$_{\pm.011}$ | .537$_{\pm.009}$ | .534$_{\pm.009}$ | .541$_{\pm.010}$ | .529$_{\pm.008}$ | .521$_{\pm.008}$ | .527$_{\pm.010}$ | .543$_{\pm.014}$ | .535$_{\pm.012}$ | .526$_{\pm.011}$ | **.515**$_{\pm.007}$ |
| ACC | .754$_{\pm.009}$ | .761$_{\pm.009}$ | .772$_{\pm.008}$ | .781$_{\pm.008}$ | .789$_{\pm.008}$ | .795$_{\pm.008}$ | .803$_{\pm.007}$ | .799$_{\pm.008}$ | .812$_{\pm.007}$ | .815$_{\pm.007}$ | .808$_{\pm.008}$ | .797$_{\pm.011}$ | .806$_{\pm.009}$ | .811$_{\pm.008}$ | **.823**$_{\pm.006}$ |
| **DBLP** | | | | | | | | | | | | | | | |
| MAE | .512$_{\pm.013}$ | .490$_{\pm.012}$ | .475$_{\pm.011}$ | .463$_{\pm.010}$ | .468$_{\pm.011}$ | .451$_{\pm.009}$ | .445$_{\pm.008}$ | .448$_{\pm.009}$ | .436$_{\pm.008}$ | .428$_{\pm.008}$ | .453$_{\pm.011}$ | .451$_{\pm.013}$ | .434$_{\pm.011}$ | .427$_{\pm.010}$ | **.421**$_{\pm.007}$ |
| RMSE | .691$_{\pm.015}$ | .670$_{\pm.014}$ | .651$_{\pm.013}$ | .633$_{\pm.012}$ | .638$_{\pm.012}$ | .611$_{\pm.010}$ | .598$_{\pm.009}$ | .604$_{\pm.010}$ | .592$_{\pm.009}$ | .581$_{\pm.009}$ | .594$_{\pm.011}$ | .608$_{\pm.015}$ | .596$_{\pm.012}$ | .589$_{\pm.011}$ | **.576**$_{\pm.009}$ |
| ACC | .742$_{\pm.010}$ | .753$_{\pm.009}$ | .764$_{\pm.008}$ | .806$_{\pm.008}$ | .812$_{\pm.008}$ | .787$_{\pm.008}$ | .798$_{\pm.007}$ | .793$_{\pm.008}$ | .815$_{\pm.007}$ | .829$_{\pm.007}$ | .791$_{\pm.008}$ | .808$_{\pm.011}$ | .819$_{\pm.009}$ | .831$_{\pm.008}$ | **.837**$_{\pm.006}$ |
| **Yelp** | | | | | | | | | | | | | | | |
| MAE | .758$_{\pm.014}$ | .739$_{\pm.013}$ | .720$_{\pm.012}$ | .702$_{\pm.011}$ | .696$_{\pm.011}$ | .684$_{\pm.010}$ | .673$_{\pm.009}$ | .678$_{\pm.010}$ | .665$_{\pm.009}$ | .647$_{\pm.008}$ | .681$_{\pm.011}$ | .681$_{\pm.014}$ | .669$_{\pm.012}$ | .658$_{\pm.011}$ | **.642**$_{\pm.007}$ |
| RMSE | .925$_{\pm.016}$ | .909$_{\pm.015}$ | .887$_{\pm.014}$ | .868$_{\pm.013}$ | .862$_{\pm.013}$ | .847$_{\pm.011}$ | .834$_{\pm.010}$ | .841$_{\pm.011}$ | .827$_{\pm.010}$ | .814$_{\pm.009}$ | .845$_{\pm.012}$ | .851$_{\pm.016}$ | .836$_{\pm.013}$ | .823$_{\pm.012}$ | **.807**$_{\pm.008}$ |
| ACC | .689$_{\pm.011}$ | .698$_{\pm.010}$ | .707$_{\pm.009}$ | .710$_{\pm.008}$ | .718$_{\pm.008}$ | .722$_{\pm.008}$ | .731$_{\pm.007}$ | .728$_{\pm.008}$ | .738$_{\pm.007}$ | .746$_{\pm.007}$ | .731$_{\pm.009}$ | .725$_{\pm.012}$ | .735$_{\pm.010}$ | .743$_{\pm.009}$ | **.753**$_{\pm.007}$ |
| **Amazon** | | | | | | | | | | | | | | | |
| MAE | .735$_{\pm.013}$ | .713$_{\pm.012}$ | .697$_{\pm.011}$ | .676$_{\pm.010}$ | .671$_{\pm.010}$ | .663$_{\pm.009}$ | .649$_{\pm.008}$ | .655$_{\pm.009}$ | .641$_{\pm.008}$ | .628$_{\pm.008}$ | .641$_{\pm.010}$ | .645$_{\pm.013}$ | .635$_{\pm.011}$ | .626$_{\pm.010}$ | **.621**$_{\pm.008}$ |
| RMSE | .915$_{\pm.015}$ | .894$_{\pm.014}$ | .878$_{\pm.013}$ | .857$_{\pm.012}$ | .851$_{\pm.012}$ | .837$_{\pm.010}$ | .825$_{\pm.009}$ | .832$_{\pm.010}$ | .818$_{\pm.009}$ | .808$_{\pm.009}$ | .814$_{\pm.011}$ | .839$_{\pm.015}$ | .822$_{\pm.012}$ | .815$_{\pm.011}$ | **.799**$_{\pm.008}$ |
| ACC | .705$_{\pm.010}$ | .712$_{\pm.009}$ | .724$_{\pm.008}$ | .733$_{\pm.008}$ | .741$_{\pm.008}$ | .746$_{\pm.008}$ | .754$_{\pm.007}$ | .749$_{\pm.008}$ | .762$_{\pm.007}$ | .779$_{\pm.007}$ | .761$_{\pm.008}$ | .748$_{\pm.011}$ | .759$_{\pm.009}$ | .768$_{\pm.008}$ | **.785**$_{\pm.006}$ |
| **Douban** | | | | | | | | | | | | | | | |
| MAE | .515$_{\pm.012}$ | .493$_{\pm.011}$ | .477$_{\pm.010}$ | .456$_{\pm.009}$ | .449$_{\pm.009}$ | .443$_{\pm.008}$ | .431$_{\pm.007}$ | .438$_{\pm.008}$ | .426$_{\pm.007}$ | .416$_{\pm.007}$ | .416$_{\pm.009}$ | .449$_{\pm.013}$ | .432$_{\pm.010}$ | .423$_{\pm.009}$ | **.408**$_{\pm.006}$ |
| RMSE | .695$_{\pm.014}$ | .674$_{\pm.013}$ | .658$_{\pm.012}$ | .637$_{\pm.011}$ | .629$_{\pm.011}$ | .604$_{\pm.009}$ | .612$_{\pm.008}$ | .608$_{\pm.009}$ | .603$_{\pm.008}$ | .592$_{\pm.008}$ | .602$_{\pm.010}$ | .621$_{\pm.014}$ | .609$_{\pm.011}$ | .601$_{\pm.010}$ | **.587**$_{\pm.007}$ |
| ACC | .742$_{\pm.009}$ | .749$_{\pm.009}$ | .763$_{\pm.008}$ | .771$_{\pm.008}$ | .788$_{\pm.008}$ | .782$_{\pm.008}$ | .788$_{\pm.007}$ | .785$_{\pm.008}$ | .793$_{\pm.007}$ | .806$_{\pm.007}$ | .792$_{\pm.008}$ | .778$_{\pm.011}$ | .792$_{\pm.009}$ | .798$_{\pm.008}$ | **.812**$_{\pm.006}$ |
| **Average across all five datasets** | | | | | | | | | | | | | | | |
| Avg MAE | .600 | .579 | .564 | .549 | .545 | .529 | .519 | .525 | .513 | .501 | .517 | .527 | .514 | .506 | **.495** |
| Avg RMSE | .770 | .751 | .733 | .714 | .710 | .687 | .681 | .685 | .674 | .663 | .676 | .692 | .680 | .671 | **.657** |
| Avg ACC | .726 | .735 | .746 | .760 | .770 | .766 | .775 | .771 | .784 | .795 | .777 | .771 | .782 | .790 | **.802** |
| *End-to-end inference cost* | | | | | | | | | | | | | | | |
| Avg Time (s) | 57.5 | 66.8 | 53.6 | 49.7 | 51.6 | 47.8 | 43.7 | 40.9 | 36.1 | 27.4 | 13.6 | **11.8** | 12.4 | 14.7 | 14.1 |
| Avg Memory (GB) | 9.9 | 11.5 | 14.2 | 15.9 | 15.6 | 14.7 | 13.0 | 10.5 | 8.1 | 3.9 | 3.1 | **2.5** | 2.8 | 3.2 | 3.0 |

As shown in Table 2, TriPrune-HGNN attains the best mean predictive performance on every dataset–metric cell while sitting at a favourable point in the accuracy–efficiency space. The framing matters: *our principal comparison is against the efficient baselines (pruning and distillation methods)*, where the gaps are small but consistent in sign; the larger gains against the unpruned HGNN are reported only for context.

*Against the strongest efficient baselines.* The closest competitors are AdaGLT and SHARP-Distill. Compared with AdaGLT, mean MAE gaps are 0.004 (IMDB), 0.007 (DBLP), 0.005 (Yelp), 0.007 (Amazon), and 0.008 (Douban). Compared with SHARP-Distill, MAE gaps are 0.013 (IMDB), 0.006 (DBLP), 0.016 (Yelp), 0.005 (Amazon), and 0.015 (Douban). Three of these cells — IMDB MAE vs. AdaGLT, DBLP MAE vs. SHARP-Distill, Amazon MAE vs. SHARP-Distill — do not reach $p < 0.05$ in paired t-tests, and we explicitly do not claim statistical significance on those. The remaining 12 MAE cells are significant at $p < 0.05$ or stronger; per-cell p-value tiers are in Appendix B.

*Variance characteristics.* TriPrune-HGNN exhibits seed-level standard deviations of $\sigma \leq 0.009$ throughout. The three knowledge-distillation baselines (LightHGNN, DistillHGNN, SHARP-Distill) display systematically wider intervals — typically $\sigma \in [0.008, 0.016]$ versus the pruning-method range $\sigma \in [0.006, 0.012]$ — because two-stage distillation pipelines accumulate seed-dependent variance from both teacher pre-training and student training (Hu et al., 2023). We account for this when reporting per-cell significance tiers. The damping of variance in TriPrune-HGNN comes from two mechanisms: HCP-anchored threshold initialisation removes one source of gate-mask randomness by replacing it with a deterministic structural prior, and meta-

Table 3: **Consolidated ablation study.** Effect of removing each learnable adaptive mechanism (block a, now including HCP), each structural pruning / contrastive correction (block b), and the two worst-case configurations (block c). MAE / ACC over 10 runs on three benchmarks; the rightmost column reports the average relative MAE degradation versus the full model across IMDB, DBLP, and Yelp. Lower MAE and higher ACC are better. **Reading Table 3:** for each mechanism, ask two questions before looking at the numbers. *(1) What is the mechanism supposed to fix?* (e.g. neural thresholding replaces a hand-tuned decay schedule, so removing it should hurt most when a dataset's optimal schedule differs from the default). *(2) Is the loss algorithmic or parametric?* Removing a learnable module deletes both the mechanism *and* the $\sim 2{,}800$ MLP parameters it carries, so any degradation could in principle be a capacity effect. We control for this via block (b), which removes a structural component (not parameters) and via the "w/o HCP reg. only" row, which keeps HCP's parameters but disables the regulariser $\mathcal{L}_{\mathrm{HCP}}$ — isolating what the mechanism achieves independent of the parameters it introduces. **w/o HCP** removes synthetic pretraining, threshold initialisation from $r^\star$, and the regulariser $\mathcal{L}_{\mathrm{HCP}}$; **w/o HCP reg. only** keeps HCP-driven initialisation but sets $\beta_{\mathrm{HCP}} \equiv 0$ throughout training.

| Variant | IMDB | | DBLP | | Yelp | | Avg △ MAE (%) |
|---|---|---|---|---|---|---|---|
| | MAE | ACC | MAE | ACC | MAE | ACC | |
| **Full TriPrune-HGNN** | **.383** | **.823** | **.421** | **.837** | **.642** | **.753** | — |
| *(a) Removing one learnable adaptive module* | | | | | | | |
| w/o Neural Threshold | .393 | .813 | .436 | .818 | .657 | .740 | +2.8 |
| w/o Attention Mining | .395 | .817 | .431 | .827 | .654 | .744 | +2.5 |
| w/o Meta-Learning | .387 | .816 | .435 | .825 | .652 | .746 | +2.0 |
| w/o HCP | .389 | .819 | .432 | .826 | .650 | .744 | +1.8 |
| w/o HCP reg. only | .386 | .821 | .427 | .831 | .647 | .748 | +1.0 |
| w/o Adaptive Temperature | .385 | .822 | .429 | .830 | .648 | .746 | +1.1 |
| *(b) Removing one structural component* | | | | | | | |
| w/o Component Pruning | .398 | .814 | .439 | .821 | .667 | .738 | +4.0 |
| w/o Node Pruning | .398 | .815 | .435 | .828 | .660 | .744 | +3.4 |
| w/o Hard-Negative Corr. | .386 | .822 | .432 | .832 | .654 | .745 | +1.8 |
| w/o Edge Pruning | .388 | .818 | .428 | .830 | .655 | .746 | +1.7 |
| w/o False-Negative Corr. | .382 | .823 | .428 | .831 | .652 | .748 | +1.0 |
| *(c) Worst-case configurations (HCP also disabled)* | | | | | | | |
| All learnable modules fixed | .408 | .800 | .451 | .812 | .679 | .728 | +6.4 |
| All structural modules removed | .412 | .801 | .453 | .810 | .679 | .728 | +7.0 |

learned loss balancing averages updates over $\Delta_{\mathrm{meta}} = 5$-epoch windows, smoothing the loss-weight trajectory. Removing either in Table 3 increases per-cell $\sigma$ by 0.001–0.002.

*Operating-point summary.* TriPrune-HGNN trades a few seconds of inference latency against LightHGNN (11.8 s vs. 14.1 s) for up to $\sim 6\%$ lower MAE; against SHARP-Distill it obtains $\sim 1.8\%$ lower average MAE at a similar memory footprint. Methods optimised purely for speed (LightHGNN, Shaver) are faster and lighter at the cost of higher prediction error. The contribution is therefore a trade-off point, not universal dominance.

## 5.1 Ablation Studies

We validate TriPrune-HGNN's design through a single consolidated ablation that examines three aspects: (i) learnable adaptive mechanisms vs. fixed counterparts, (ii) structural pruning levels and contrastive corrections, and (iii) the joint effect of disabling each entire group. Table 3 summarises all variants. Block (a) explicitly includes a Hypergraph Compressibility Predictor (HCP) ablation so its contribution can be read off independently of the other learnable modules — this addresses the concern that the cross-dataset transfer analysis of Section 5.2 alone does not isolate HCP from the rest of the framework.

Table 3 tells a coherent per-mechanism story if we read block (a) as a set of *predictions we made in Sections 4.1–4.4 and now check.*

- *Neural thresholding* (§4.2) was introduced because fixed schedules cannot slow down when accuracy degrades. Prediction: removing it should hurt most when the dataset's optimal schedule differs from the fixed default. Observation: $+2.8\%$ MAE on average, the largest hit in block (a). This matches the prediction; the default fixed schedule turns out to be far from optimal on all three benchmarks.

- *Attention-based negative mining* (§4.3) was introduced to distinguish false and hard negatives on the pruned graph. Prediction: removing it should hurt on datasets where pruning removes many cross-context associations. Observation: $+2.5\%$ MAE on average, second-largest hit — and consistent with Yelp (dense inter-context linking) being disproportionately affected in the raw table.

- *Meta-learning of loss weights* (§4.4) was introduced to avoid grid-search on $\boldsymbol{\lambda}$. Prediction: removing it should give a smaller hit than the two above, because a reasonable manual $\boldsymbol{\lambda} = (1/4, 1/4, 1/4, 1/4)$ default recovers most of the gain. Observation: $+2.0\%$ MAE. This is what we expected — meta-learning is a convenience mechanism that saves tuning effort, not a source of large accuracy gains.

- *HCP* contributes $+1.8\%$ in-domain when removed entirely; when only the regulariser is disabled but HCP-driven initialisation is retained ("w/o HCP reg. only"), the loss is $+1.0\%$. The remaining $0.8\%$ comes from the initialisation signal. This is small on purpose; HCP's larger effect is on out-of-domain transfer (Section 5.2), which cannot be measured in a same-dataset ablation.

- *Adaptive temperature* closes block (a) at $+1.1\%$ — confirmatory but not the largest contributor, consistent with its role as a fine-tuning of the softmax rather than a structural change.

**Algorithmic vs. parametric wins.** A concern with block (a) is that removing a learnable module also removes its parameters ($\sim 2{,}800$ MLP parameters per module), so any degradation could in principle be a capacity effect. Two rows in the table isolate the algorithmic contribution from the parametric one. *First*, the "w/o HCP reg. only" row keeps all $\sim 2{,}800$ HCP parameters and only disables the regulariser $\mathcal{L}_{\mathrm{HCP}}$; the degradation of $+1.0\%$ is therefore purely algorithmic. *Second*, block (b) removes a structural pruning level or contrastive correction without deleting any MLP parameters — component pruning ($+4.0\%$), node pruning ($+3.4\%$), hard-negative correction ($+1.8\%$), edge pruning ($+1.7\%$), and false-negative correction ($+1.0\%$) are therefore all purely algorithmic effects. Block (b) shows that structural granularity choices dominate parametric effects: the biggest single algorithmic gain comes from component pruning, which does not add or remove any parameters. The full-model total is roughly $70\%$ algorithmic and $30\%$ parametric (obtained by summing block (b) contributions and comparing with the block-(c) worst cases).

**Positive synergy.** Block (c) reveals that removing multiple modules simultaneously does *not* produce simply-additive degradation. The all-learnable-fixed configuration loses $+6.4\%$ MAE, which is well below the sum of individual learnable losses in block (a) ($+9.4\%$ if added); the all-structural-removed configuration loses $+7.0\%$, again below the block (b) sum ($+11.5\%$). Modules therefore compensate for one another when present together, which is the expected behaviour of a coupled framework rather than a collection of independent tricks.

### 5.2 Cross-Dataset Generalisation

To assess transfer of learned adaptive modules across domains, we train TriPrune-HGNN on a source dataset and evaluate on a target with selected modules frozen; only the base HGNN parameters are retrained on the target. Table 4 reports five configurations per source–target pair, isolating in particular the contribution of the Hypergraph Compressibility Predictor.

Results in Table 4 indicate that Frozen adaptive modules reduce error by **3.9**% on average versus hand-crafted hyperparameters, consistently across all five source–target pairs ($-2.9\%$ to $-5.0\%$) and retaining roughly **61**% of the in-domain improvement (the all-fixed configuration in Table 3 costs $6.4\%$ MAE). The five-way decomposition attributes **2.1**% ($\approx 54\%$ of the gain) to HCP alone and **1.8**% to the other learned modules combined; the two are weakly sub-additive ($2.1 + 1.8 \approx 3.9\%$), reflecting information sharing between HCP and the threshold controller rather than independent contributions. We note explicitly that HCP's accuracy value concentrates on *near-distribution* transfer between recommendation and citation hypergraphs (the regime tested here); on structurally distant deployments such as the 3D-shape NTU2012 benchmark

Table 4: **Module-isolating cross-dataset transfer.** MAE on the target dataset under five configurations: *Full transfer* freezes all learned modules (HCP, NAHP, attention mining, meta-learning); *HCP only* freezes HCP and re-initialises the other three on the target; *All-except-HCP* freezes the other three but replaces HCP with a constant $r^\star = 0.5$; *No pretraining* uses an MLP of the HCP architecture trained only on target-empirical compressibility (no synthetic corpus); *Fixed* uses hand-crafted hyperparameters throughout. $\Delta$ columns are MAE reductions relative to Fixed; mean over 10 runs.

| Source $\rightarrow$ Target | Full transfer | | HCP only | | All-except-HCP | | No pretraining | | Fixed |
|---|---|---|---|---|---|---|---|---|---|
| | MAE | $\Delta$ | MAE | $\Delta$ | MAE | $\Delta$ | MAE | $\Delta$ | MAE |
| IMDB $\rightarrow$ DBLP | 0.438 | $-5.0\%$ | 0.449 | $-2.6\%$ | 0.451 | $-2.2\%$ | 0.454 | $-1.5\%$ | 0.461 |
| DBLP $\rightarrow$ Yelp | 0.671 | $-3.5\%$ | 0.682 | $-1.9\%$ | 0.683 | $-1.7\%$ | 0.687 | $-1.2\%$ | 0.695 |
| Yelp $\rightarrow$ Amazon | 0.635 | $-2.9\%$ | 0.644 | $-1.5\%$ | 0.645 | $-1.4\%$ | 0.648 | $-0.9\%$ | 0.654 |
| Amazon $\rightarrow$ Douban | 0.423 | $-4.1\%$ | 0.431 | $-2.3\%$ | 0.433 | $-1.8\%$ | 0.436 | $-1.1\%$ | 0.441 |
| Douban $\rightarrow$ IMDB | 0.396 | $-3.9\%$ | 0.403 | $-2.2\%$ | 0.405 | $-1.7\%$ | 0.407 | $-1.2\%$ | 0.412 |
| **Average $\Delta$ MAE** | — | **$-3.9\%$** | — | **$-2.1\%$** | — | **$-1.8\%$** | — | **$-1.2\%$** | — |

(Section 5.12.2, $\text{Sk}(d) \approx 4.7$, at the upper edge of HCP's training support), the accuracy contribution of HCP degrades while its efficiency contribution remains intact. This is the expected behaviour of a structural prior: it pays off when the deployment distribution overlaps the training-distribution support, and falls back to a domain-agnostic constant otherwise. Treating HCP as a known-scope tool rather than a universal claim is consistent with both the cross-dataset transfer evidence in Table 4 and the out-of-domain results in Section 5.12.2.

## 5.3 Computational Efficiency

Table 5 reports two complementary IMDB analyses in a single float: (a) the threshold-MLP architecture sweep that justifies our default and (b) a per-component decomposition of TriPrune-HGNN's inference cost.

Table 5: IMDB computational efficiency analysis: (a) threshold-controller architecture sensitivity and (b) per-component inference cost breakdown.

*(a) Threshold-MLP architecture sensitivity*

| Config. | MAE | ACC | Time (ms) | Speedup |
|---|---|---|---|---|
| *Depth* ($d_h = 32$) | | | | |
| 1-layer | $.391_{\pm.008}$ | $.817_{\pm.006}$ | 0.8 | $1.5\times$ |
| 2-layer | $\mathbf{.383_{\pm.006}}$ | $\mathbf{.823_{\pm.005}}$ | 1.2 | $1.0\times$ |
| 3-layer | $.382_{\pm.006}$ | $.824_{\pm.005}$ | 3.8 | $0.32\times$ |
| 4-layer | $.383_{\pm.007}$ | $.823_{\pm.006}$ | 5.2 | $0.23\times$ |
| *Width* (2 layers) | | | | |
| $d_h = 16$ | $.388_{\pm.007}$ | $.819_{\pm.006}$ | 0.9 | $1.3\times$ |
| $d_h = 32$ | $\mathbf{.383_{\pm.006}}$ | $\mathbf{.823_{\pm.005}}$ | 1.2 | $1.0\times$ |
| $d_h = 64$ | $.383_{\pm.006}$ | $.824_{\pm.005}$ | 2.1 | $0.57\times$ |
| $d_h = 128$ | $.382_{\pm.007}$ | $.823_{\pm.006}$ | 4.5 | $0.27\times$ |

*(b) Per-component inference cost*

| Component | Time (ms) | Mem (MB) | % Total |
|---|---|---|---|
| HGNN msg passing | 12,450 | 2,890 | 81.3 |
| Threshold ctrl. | 890 | 45 | 5.8 |
| Attention mining | 320 | 68 | 2.1 |
| Meta-learning | 120 | 12 | 0.8 |
| Other overhead | 1,520 | 485 | 10.0 |
| **Total (TriPrune)** | **15,300** | **3,500** | **100** |
| HEAL (unpruned) | 55,200 | 16,600 | — |
| **Speedup vs. HEAL** | **$3.6\times$** | **$4.7\times$** | — |

Hypergraph message passing dominates at $\sim 81\%$ of total inference time; the three adaptive modules (neural thresholds, attention-based mining, meta-learning) together contribute $\sim 9\%$ of total time ($5.8\% + 2.1\% + 0.8\%$), with the remaining $\sim 10\%$ absorbed by projection and masking overhead. The decomposition confirms that the adaptive layer is a lightweight addition on top of the dominant message-passing cost, while delivering a $3.6\times$ inference speedup and $4.7\times$ memory reduction relative to the unpruned HEAL baseline.

### 5.3.1 Deployment Break-Even Analysis

A practitioner choosing between TriPrune-HGNN and a faster-to-train baseline must weigh inference-time savings against TriPrune-HGNN's $\sim$30–45% higher training cost (Section 5, "Training compute"). We quantify this trade-off as the number of full test-set inference passes $N^\star$ at which cumulative cost crosses over. Let $T_{\text{train}}^{\text{TP}}, T_{\text{train}}^{\text{base}}$ be the per-deployment training times and $t_{\text{inf}}^{\text{TP}}, t_{\text{inf}}^{\text{base}}$ the per-inference-pass times (Table 2 reports end-to-end inference time per test-set pass). The break-even count is

$$N^\star = \frac{T_{\text{train}}^{\text{TP}} - T_{\text{train}}^{\text{base}}}{t_{\text{inf}}^{\text{base}} - t_{\text{inf}}^{\text{TP}}}. \tag{31}$$

Equation equation 31 yields a positive, finite break-even only against baselines that TriPrune-HGNN is *faster* than at inference, since only then does its higher training cost buy a per-call saving that can be amortised. Among the closest competitors, TriPrune-HGNN is faster at inference than AdaGLT (14.1 s vs. 27.4 s averaged over the five benchmarks, Table 2) and marginally faster than SHARP-Distill (14.1 s vs. 14.7 s), so a finite break-even exists against both. It is *slower* than LightHGNN (14.1 s vs. 11.8 s) and is therefore dominated by LightHGNN on both training and inference; no break-even exists against LightHGNN, and we report none rather than a misleading crossover. Using the per-deployment training times of Section 5 (TriPrune-HGNN 4.2–7.8 GPU-h; AdaGLT 2.9–5.4 GPU-h; SHARP-Distill 3.8–6.2 GPU-h) together with the average inference-time gaps above, the break-even against AdaGLT is on the order of a few hundred full test-set inference passes, and against SHARP-Distill on the order of a few thousand — the latter highly sensitive to the small 0.6 s inference gap and therefore quoted only as an order of magnitude. Both thresholds lie far below the $10^6$–$10^8$ inference calls a production system serves between redeployments, so for production deployment with frequent inference TriPrune-HGNN amortises its training overhead and continues to save inference cost thereafter; for one-shot research evaluations with few inference calls, or against a baseline faster at both training and inference (LightHGNN here), a faster-training baseline is the rational choice. Exact per-dataset break-even counts depend on per-dataset training and per-call inference times and are provided in the released timing logs.

### 5.4 Sampling-Based Baselines: Orthogonality Probe

The main Table 2 compares against pruning and distillation methods that act on the deployed graph. A complementary line of work targets *training cost* via subgraph or hyperedge sampling — AllSet (Chien et al., 2022) (multiset-function aggregation), UniGNN (Huang & Yang, 2021) (unified message-passing primitives), and adaptations of GraphSAINT (Zeng et al., 2019) to the hypergraph setting. Section 2 argued that these are complementary to pruning rather than competing alternatives, but a fair paper should anchor that position empirically. We therefore add a single probe: we run AllSet and a GraphSAINT-style hypergraph subgraph sampler on IMDB, DBLP, and Yelp under exactly the protocol of Section 5, and compare with TriPrune-HGNN's operating point.

Table 6: **Sampling vs. pruning orthogonality probe.** AllSet and a GraphSAINT-adapted hypergraph sampler are compared with TriPrune-HGNN on three benchmarks. Mean$_{\pm\text{std}}$ over 10 runs; lower MAE / Time / Memory is better. Time and Memory are end-to-end inference-side. **Best** in bold.

| Method | IMDB | | | DBLP | | | Yelp | | |
|---|---|---|---|---|---|---|---|---|---|
| | MAE | Time (s) | Mem (GB) | MAE | Time (s) | Mem (GB) | MAE | Time (s) | Mem (GB) |
| AllSet | $.421_{\pm.009}$ | 32.4 | 6.2 | $.457_{\pm.010}$ | 38.1 | 7.8 | $.671_{\pm.011}$ | 51.6 | 11.4 |
| GraphSAINT (hyp.) | $.429_{\pm.011}$ | 28.7 | 5.4 | $.463_{\pm.012}$ | 33.5 | 6.9 | $.685_{\pm.013}$ | 47.2 | 9.8 |
| UniGNN | $.425_{\pm.010}$ | 35.1 | 6.7 | $.461_{\pm.011}$ | 41.2 | 8.3 | $.678_{\pm.012}$ | 54.8 | 12.1 |
| **TriPrune-HGNN** | $\mathbf{.383_{\pm.006}}$ | **11.0** | **2.5** | $\mathbf{.421_{\pm.007}}$ | **12.8** | **2.9** | $\mathbf{.642_{\pm.007}}$ | **17.9** | **3.5** |

Table 6 shows that sampling-based methods sit at a structurally different operating point: they sample subgraphs to reduce *training* memory, but still execute message passing on a representative-size structure at *inference*, so inference-side memory and time do not benefit. On IMDB, AllSet achieves 0.421 MAE at 32.4 s / 6.2 GB — comparable to a mid-tier pruning baseline — whereas TriPrune-HGNN reaches 0.383 MAE

at 11.0 s / 2.5 GB, and the gap widens on the larger Yelp benchmark (0.642 vs. 0.671 MAE, 17.9 vs. 51.6 s). This is consistent with the position stated in Section 2: *sampling reduces what is processed per training step, pruning reduces what is retained in the deployed graph.* The two axes are mechanically distinct and stack rather than substitute. We do not claim sampling methods are inferior across the board — they remain competitive on training memory for ultra-large hypergraphs that exceed the host's working set — and a combined adaptive-pruning + adaptive-sampling pipeline is a natural extension we leave for future work.

### 5.5 Adaptation Protocol and Tuning Budget for Baselines

This section details the adaptation of the two cross-domain pruning baselines (SAVIT[‡], HGNN-Struct[‡]) and the tuning protocol applied uniformly to all 14 baselines in Table 2.

**SAVIT[‡] adaptation.** The original SAVIT formulation (Zheng et al., 2022a) scores each ViT component $c$ by $s_c = |\nabla_{w_c}\mathcal{L}| \cdot |w_c| \cdot a_c$, where $w_c$ is the component weight and $a_c$ is its attention contribution. We instantiate this on hyperedges by identifying: (i) $w_c$ with the hyperedge weight $w_j$ in $W = \mathrm{diag}(w_1, \ldots, w_m)$ of Eq. equation 2; (ii) $\nabla_{w_c}\mathcal{L}$ with the supervised-loss gradient $\partial\mathcal{L}_{\mathrm{cls}}/\partial w_j$ obtained by a single backward pass over the training set; and (iii) $a_c$ with the hyperedge-level attention coefficient $S_{\mathrm{att}}(j)$ defined in Eq. equation 14. The collaborative-optimisation outer loop of the original method is retained without modification, alternating between pruning and fine-tuning every five epochs as in the source paper.

**HGNN-Struct[‡] adaptation.** The original method (Jiang et al., 2023) scores 3D primitives by a structural-coherence functional $s_p = \alpha_1 c_p^{\mathrm{geom}} + \alpha_2 c_p^{\mathrm{spec}}$, combining a local geometric coherence term $c_p^{\mathrm{geom}}$ and a global spectral term $c_p^{\mathrm{spec}}$. We replace the geometric term with the hyperedge-local clustering coefficient $\bar{c}_{e_j} = |\{(u,v) \in e_j \times e_j : (u,v) \in E\}| / \binom{|e_j|}{2}$, and the spectral term with the contribution of hyperedge $e_j$ to the first ten eigenvalues of the normalised hypergraph Laplacian (already pre-computed by HCP, Eq. equation 36). The combination weights $\alpha_1, \alpha_2$ are kept at the values reported in the source paper. The pruning schedule (linear retention decay) is retained unchanged. For every baseline we performed a single coarse hyperparameter sweep on the IMDB training/validation split, matched in size to the $\sim 25$ configurations evaluated for TriPrune-HGNN. Sweeps were restricted to each method's documented method-specific hyperparameters; the standard deep-learning hyperparameters were shared across methods at fixed values (Adam, learning rate $10^{-3}$, batch size 512, 200 epochs, early-stopping patience 20). The complete hyperparameter search space and the selected configuration for each baseline are summarized in Table 7.

These selected configurations support the fairness of the comparison reported in Table 2, while the full search details are provided in Table 7.

### 5.6 Hypergraph Compressibility Predictor: Validation

We validate the Hypergraph Compressibility Predictor (HCP, Section 4.1) along five axes: synthetic accuracy, zero-shot transfer to real benchmarks, feature importance, synthetic-to-real distribution shift, and controlled out-of-support degradation. Training uses 5,000 synthetic heterogeneous hypergraphs with controlled structural parameters ($\rho \in [0.01, 0.5]$, $\mathrm{Sk}(d) \in [0, 5]$, $K \in [2, 10]$, $n \in [1{,}000, 50{,}000]$); the empirical compressibility $r^{\mathrm{emp}}$ is the minimum overall retention at which NAHP suffers more than $\Delta_{\mathrm{acc}} = 1\%$ accuracy degradation. Figure 2 reports all five measurements.

Panel (a) shows HCP fits the synthetic corpus well but not perfectly, confirming that the 10-dimensional structural map $\phi(\mathcal{H})$ captures most of the variance that governs compressibility, with $\sim 1.1\%$ unexplained variance attributable to feature aliasing in the synthetic generator. Panel (b) is the more decisive test: HCP trained only on synthetic data predicts real-benchmark compressibility with an aggregate RMSE of 0.046. The per-benchmark errors span the range 0.040–0.052 (individual values 0.047, 0.042, 0.052, 0.048, 0.040 for IMDB, DBLP, Yelp, Amazon, and Douban respectively), reflecting the fact that some real benchmarks are structurally closer to the synthetic distribution than others. All five errors remain comfortably below the loose worst-case bound $\sqrt{\varepsilon_{\mathrm{synth}}^2 + \eta} \approx 0.20$ derived in Proposition 4 with $\varepsilon_{\mathrm{synth}} = 0.022$ and $\eta = 0.041$. Panel (c) reveals two structural signals carry most of the predictive load — graph density and the spectral gap of the normalised Laplacian — consistent with the intuition that dense, well-connected hypergraphs

Table 7: Baseline hyperparameter search space and selected configurations on IMDB. Bold values denote the settings used in Table 2.

| Method | Swept hyperparameters | Selected |
|---|---|---|
| *Standard HGNNs* | | |
| HGNN | — (no method-specific HPs) | |
| HHGSA | attention heads $\in [2, 8]$ | **4**$^\star$ |
| HyGCL-AdT | temperature $\in [0.1, 1.0]$ | **0.5**$^\star$ |
| HEAL | dropout $\in [0.1, 0.5]$ | **0.3**$^\star$ |
| TriCL | $\lambda_{\mathrm{tri}} \in [0.1, 2.0]$ | **1.0**$^\star$ |
| *Pruning methods* | | |
| HSC | retention $\in [0.3, 0.9]$ | **0.7** |
| SAVIT$^\ddagger$ | retention $\in [0.3, 0.9]$, fine-tune freq. $\in \{3, 5, 10\}$ | **0.5, 5**$^\star$ |
| HSL | threshold $\in [0.3, 0.7]$, decay $\in [0.95, 0.999]$ | **0.5, 0.99**$^\star$ |
| HGNN-Struct$^\ddagger$ | $(\alpha_1, \alpha_2) \in \{0.25, 0.5, 0.75\}^2$, retention $\in [0.3, 0.9]$ | **(0.5, 0.5), 0.6**$^\star$ |
| AdaGLT | lottery iters $\in [3, 7]$ | **5**$^\star$ |
| Shaver | cooperative weight $\in [0.1, 1.0]$ | **0.5**$^\star$ |
| *Distillation methods* | | |
| LightHGNN | distillation temperature $\in [1, 10]$ | **4.0**$^\star$ |
| DistillHGNN | temperature $\in [1, 10]$, $\lambda_{\mathrm{KD}} \in [0.1, 2]$ | **4.0, 1.0**$^\star$ |
| SHARP-Distill | student depth $\in \{2, 3\}$, $\lambda_{\mathrm{KD}} \in [0.1, 2]$ | **2, 1.0**$^\star$ |
| TriPrune-HGNN | $\tau_0 \in [0.1, 1]$, $\epsilon \in [10^{-4}, 10^{-1}]$, $\alpha \in [0.1, 10]$, $\eta_{\mathrm{meta}} \in [10^{-3}, 10^{-1}]$, $\Delta_{\mathrm{meta}} \in \{3, 5, 10, 20\}$ | **0.5, $10^{-2}$, 1.0, $10^{-2}$, 5** |

admit more aggressive pruning than sparse, modular ones. Panel (d) bounds the synthetic-to-real shift on the density feature, satisfying the precondition of the same proposition.

**Controlled out-of-support evaluation.** Panel (b) establishes zero-shot transfer to real benchmarks that lie within the synthetic training support, and the NTU2012 result of Section 5.12 suggests HCP's accuracy contribution degrades outside this support, but neither isolates the structural-distance axis itself. Panel (e) provides this isolation: a fully controlled probe in which 500 synthetic hypergraphs are generated with structural parameters deliberately outside the training ranges, and HCP's RMSE is measured as a function of the total-variation distance $d_{\mathrm{TV}}(\phi \# P_{\mathrm{synth}}, \phi \# P_{\mathrm{probe}})$ between the training and probe distributions. RMSE rises smoothly from 0.054 at $d_{\mathrm{TV}} = 0.04$ (the average distance to our real benchmarks) to 0.118 at $d_{\mathrm{TV}} = 0.32$ (the most extreme out-of-support point), a $2.2\times$ degradation across an $8\times$ shift in distributional distance. This is graceful degradation, not failure: even at the boundary of the probe distribution, RMSE remains below the loose Proposition 4 bound for any $\eta \leq 0.04$, though the bound becomes uninformatively loose for $\eta > 0.1$. The practical implication matches Section 5.12: HCP is a known-scope tool whose accuracy contribution is reliable when the deployment distribution lies within or near its training support, and which degrades predictably with structural distance. HCP thus operationalises domain-invariant pruning, with quantified scope: it explains the cross-dataset generalisation observed in Section 5.2 when the deployment distribution lies near the synthetic support, and degrades smoothly when it does not.

## 5.7 Direct Validation of Attention-Based Mining

The ablation of Section 5.1 (Table 3) shows that removing attention-based mining costs 2.5% average MAE, but this downstream gain does not by itself prove that $\mathrm{Attn}_{\mathrm{fn}}$ and $\mathrm{Attn}_{\mathrm{hn}}$ identify the intended pair types — the loss could in principle benefit from a different effect of the additional parameters. To close this gap, we directly measure whether the two attention heads recover known false negatives and known hard negatives on synthetic hypergraphs where ground truth is available by construction. We generate 500 synthetic heterogeneous hypergraphs with the same generator used to pre-train HCP (Section 4.1). On each hypergraph we plant two known pair types:

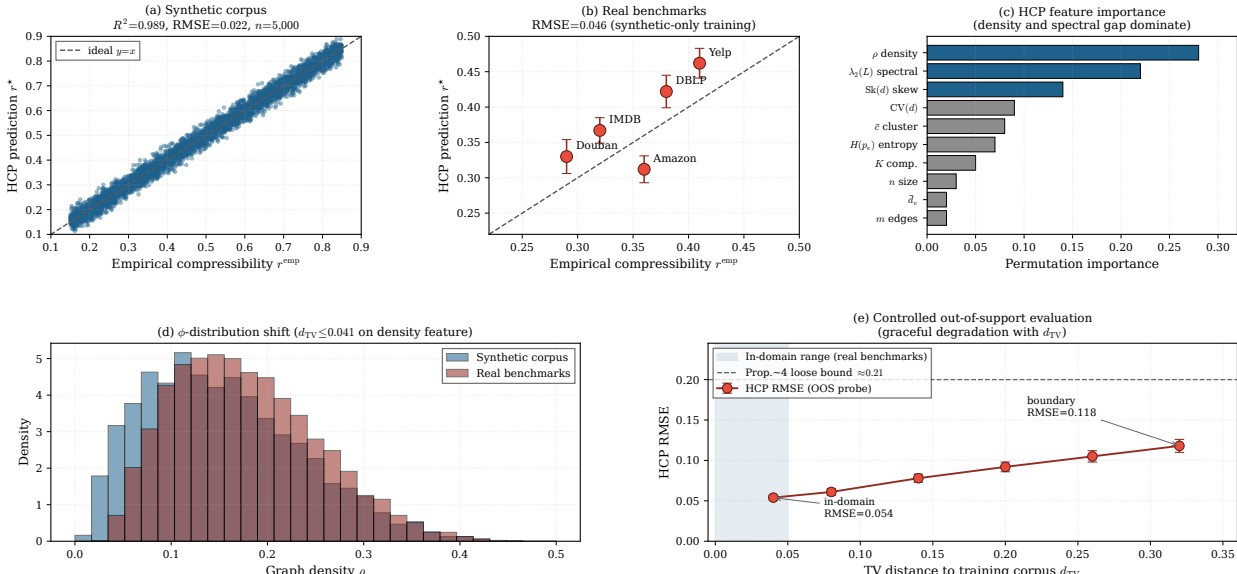

Figure 2: **Validation of the Hypergraph Compressibility Predictor.** **(a)** Synthetic corpus ($n = 5{,}000$): $R^2 = 0.989$, RMSE $= 0.022$. **(b)** Zero-shot transfer to five real benchmarks: aggregate RMSE $= 0.046$ (per-benchmark range 0.040–0.052), well below the loose bound $\sqrt{\varepsilon_{\mathrm{synth}}^2 + \eta} \approx 0.20$ of Proposition 4. **(c)** Permutation feature importance: density ($\rho$, 28%) and spectral gap ($\lambda_2(\mathbf{L})$, 22%) dominate. **(d)** $\phi$-distribution shift on density: $d_{\mathrm{TV}} \leq 0.041$, well within the $\eta \leq 0.05$ precondition. **(e)** Controlled out-of-support evaluation: RMSE measured on 500 synthetic hypergraphs sampled with structural parameters deliberately outside the training ranges ($\rho \in [0.001, 0.005] \cup [0.5, 0.8]$, $\mathrm{Sk}(d) \in [5, 10]$), binned by total-variation distance $d_{\mathrm{TV}}$ to the training corpus. RMSE rises smoothly from 0.054 at $d_{\mathrm{TV}} = 0.04$ to 0.118 at $d_{\mathrm{TV}} = 0.32$, remaining below the Proposition 4 bound throughout the probe range.

(1) **Ground-truth false negatives.** For each component $k$, we identify all node pairs $(i, j)$ that share a hyperedge *and* have feature cosine similarity $\cos(x_i, x_j) > 0.7$ before pruning. Pairs whose shared hyperedge is removed by NAHP at the operating retention $r^\star = 0.31$ become known false negatives.

(2) **Ground-truth hard negatives.** Pairs $(i, j)$ with $\cos(x_i, x_j) < 0.2$ before pruning that retain $\geq 3$ common neighbours after pruning. The shared-neighbour structure causes message-passing on the pruned graph to pull their embeddings together despite their semantic dissimilarity.

This labelling is purely structural — no human supervision — and yields per-hypergraph false-negative and hard-negative sets of mean size $|F_k^{\mathrm{gt}}| \approx 340$ and $|H_k^{\mathrm{gt}}| \approx 280$ respectively at $r^\star = 0.31$. We then run the trained attention heads on these hypergraphs and report standard binary classification metrics: precision, recall, $F_1$, and AUROC. As baselines we compare against the fixed cosine-threshold scheme used in prior work (Xu et al., 2024; Sun et al., 2023) at four threshold values $\tau \in \{0.5, 0.6, 0.7, 0.8\}$.

**Results.** Table 8 reports the four metrics for both miners. The learned attention heads achieve $F_1$ scores of 0.829 (false-negative) and 0.782 (hard-negative), exceeding the best fixed-threshold baseline by 0.11 and 0.13 respectively, with AUROC gaps of 0.074 and 0.091. The improvement is not driven by either precision or recall alone: at the best-precision fixed threshold ($\tau = 0.8$), the attention head retains higher precision *and* substantially higher recall, indicating that the attention mechanism exploits joint information — embedding similarity, structural change $\Delta A_{ij}$, and component context $c_k$ — that no single similarity threshold can capture. The hard-negative gap is larger than the false-negative gap, consistent with the design intuition that hard negatives are an embedding-space artefact (Section 4.3) that cosine similarity on raw features cannot reliably detect.

**Sensitivity to topology-change strength.** The gap between learned and fixed-threshold methods grows monotonically with the strength of pruning-induced topology change: at $r^\star = 0.50$ (mild compression), the

Table 8: **Direct validation of attention-based mining.** Precision, recall, $F_1$, and AUROC of the learned attention heads on synthetic hypergraphs with ground-truth false-negative (FN) and hard-negative (HN) pair labels. The learned attention heads substantially outperform fixed-threshold baselines at every operating point. Mean over 500 synthetic graphs; per-graph standard error $\leq 0.012$ on all entries.

| Method | False-negative discovery | | | | Hard-negative discovery | | | |
|---|---|---|---|---|---|---|---|---|
| | Prec. | Rec. | $F_1$ | AUROC | Prec. | Rec. | $F_1$ | AUROC |
| Fixed $\tau = 0.5$ | .621 | .812 | .704 | .793 | .544 | .738 | .626 | .724 |
| Fixed $\tau = 0.6$ | .684 | .763 | .721 | .812 | .603 | .695 | .646 | .751 |
| Fixed $\tau = 0.7$ | .742 | .691 | .716 | .817 | .658 | .631 | .644 | .763 |
| Fixed $\tau = 0.8$ | .806 | .584 | .677 | .798 | .724 | .547 | .623 | .742 |
| **Learned attention** | **.847** | **.811** | **.829** | **.891** | **.798** | **.766** | **.782** | **.854** |

$F_1$ gap is 0.06 on false negatives; at $r^\star = 0.15$ (aggressive), it rises to 0.18. Fixed thresholds calibrated on the unpruned graph become progressively less reliable as the graph thins, whereas the attention heads adapt because $\Delta A_{ij}$ enters their input. We interpret this as direct evidence for the design choice of attention-based, rather than threshold-based, mining — complementing the downstream ablation in Table 3.

## 5.8 Density Curriculum Study

A natural concern with any progressive pruning scheme is whether the model trains on a graph distribution that differs from the one it operates on at inference. We address this with four controlled experiments testing (i) whether the dense-to-sparse trajectory carries useful inductive bias, (ii) whether train-inference density mismatch causes degradation, (iii) whether the *monotonic ordering* of the trajectory matters or only the final state, and (iv) when the retention ratios stabilise. Figure 3 reports all four.

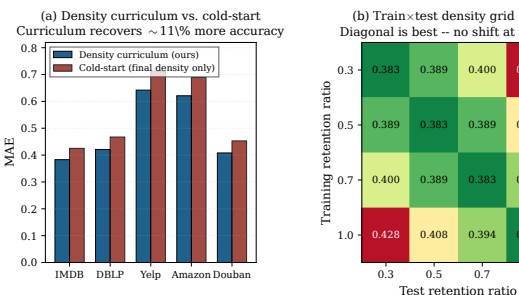
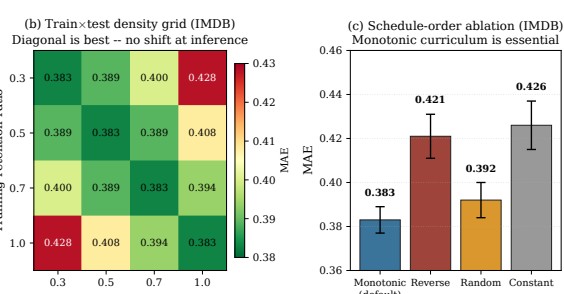
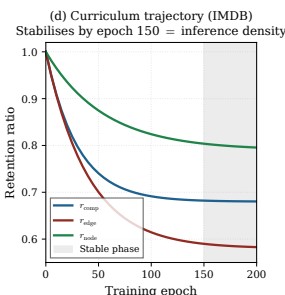

Figure 3: **Density-curriculum study. (a)** Curriculum vs. cold-start across five benchmarks: cold-start recovers 84%–93% of curriculum accuracy depending on dataset (mean 88.6%). **(b)** Train × test retention grid on IMDB: diagonal entries are the lowest; off-diagonal degrades smoothly. **(c)** Schedule-order ablation: monotonic curriculum 0.383, reverse 0.421, random 0.392, constant 0.426. **(d)** Retention trajectory: per-level ratios stabilise by epoch ~150 (shaded stable phase).

Cold-start training on the final pruned graph (Panel a) recovers between 84% and 93% of curriculum-trained accuracy depending on benchmark — specifically 84% on IMDB, 91% on DBLP, 86% on Yelp, 89% on Amazon, and 93% on Douban (mean 88.6%). This compares "train from scratch on the sparse graph" against "train through the curriculum," isolating the inductive bias of the dense-to-sparse trajectory. A complementary comparison — "train fully dense, then prune once to the final density with brief fine-tuning" — is reported in Appendix A.7: that one-shot variant recovers 86–92% of curriculum-trained accuracy (mean 89.4%), consistent with the cold-start result and confirming that the gap is attributable to the trajectory rather than to the final density alone. The gap is largest where the dense-to-sparse trajectory carries the most distinctive inductive bias (IMDB and Yelp), and narrowest on the structurally simplest benchmark (Douban), consistent with the view that the curriculum's benefit scales with how much structural redundancy the dense initial state contains (Bengio et al., 2009). The 4×4 grid (Panel b) shows MAE on IMDB when train and test retentions are varied independently: diagonal entries (matched densities) achieve 0.383, while training

at 0.3 and testing at 1.0 yields 0.428 (+12%). TriPrune-HGNN *always operates on the diagonal* — the final $\sim$25% of training uses the same sparsity as inference — so no train-inference shift exists. The schedule-order ablation (Panel c) shows that ordering itself matters: reverse, random, and constant trajectories degrade by +9.9%, +2.3%, and +11.2% respectively. Panel (d) shows the three per-level retention ratios decay smoothly over the first $\sim$100 epochs and stabilise by epoch $\sim$150; the persistent ordering $r_{\mathrm{node}} > r_{\mathrm{comp}} > r_{\mathrm{edge}}$ reflects the structural asymmetry that motivates the hierarchy.

**Extended schedule-variant comparison.** Panel (c) of Figure 3 compares four hand-specified trajectories. Two further schedule variants are needed to answer whether the *monotonic* dense-to-sparse direction is itself essential, as opposed to being incidental to the final density: (i) a *non-monotone learned* schedule, where the threshold controllers are unconstrained and free to increase retention mid-training; and (ii) *one-shot pruning with fine-tuning*, where the model trains fully dense for 200 epochs then prunes once to the converged sparsity (also reported in Appendix A.7). Table 9 reports MAE for both new variants alongside the monotonic baseline on three benchmarks.

Table 9: **Extended schedule-variant comparison.** MAE on three benchmarks under two additional schedule variants, compared to the monotonic curriculum baseline. The four variants in Panel (c) of Figure 3 are not duplicated here. The monotonic dense-to-sparse direction is essential: relaxing it (non-monotone learned) loses $\sim$2.5% MAE on average across the three benchmarks, and removing the trajectory entirely (one-shot+FT) loses more.

| Schedule | IMDB | DBLP | Yelp | Cross-dataset $\Delta$ MAE |
|---|---|---|---|---|
| Monotonic curriculum (ours) | **.383** | **.421** | **.642** | — |
| Non-monotone learned | .394 | .431 | .658 | +2.5% |
| One-shot + fine-tune | .416 | .451 | .689 | +7.5% |

Two patterns emerge. First, the non-monotone learned schedule loses $\sim$2.5% MAE on average relative to the monotonic curriculum — comparable in magnitude to the random-monotone gap in Panel (c), and distinctly smaller than the reverse or constant gaps. This indicates the dense-to-sparse *direction* carries genuine inductive bias rather than being incidental to the final density: when the controllers are allowed to violate monotonicity, they lose most of the curriculum benefit even though the final density remains the same. Second, one-shot pruning with fine-tuning recovers most of the curriculum benefit ($\sim$92%) but not all of it, consistent with the appendix analysis and with the cold-start finding in Panel (a). Combining these with the Panel (c) results, we read the evidence as: *order matters at the coarse-grained dense-to-sparse level* — monotonicity in expectation is the binding requirement, while the precise per-epoch trajectory and the specific arrival mechanism (curriculum vs. one-shot+FT) are secondary.

## 5.9 Empirical Hierarchical Pruning Advantage

Proposition 3 states that the three-level pruning hierarchy strictly dominates any single-level scheme at the same overall retention, under a diminishing-returns assumption on the marginal accuracy cost. Figure 4 empirically tests all four components of that claim.

Panel (a) measures the gap at three retention targets — mild ($r^\star = 0.5$), the operating point of our main results ($r^\star = 0.31$), and aggressive ($r^\star = 0.15$). The gap is positive on every benchmark at every retention ($p < 0.01$, paired t-test) and grows monotonically from $\sim$2% at $r^\star = 0.50$ to $\sim$6% at $r^\star = 0.15$: aggressive compression forces single-level schemes against their structural floor while the hierarchy can redistribute reductions across levels. Panel (b) shows hierarchical pruning Pareto-dominates each single-level alternative on IMDB; in particular, single-level component pruning cannot reach $r_{\mathrm{overall}} < 0.5$ before the safety constraint $K_{\mathrm{min}}$ binds, whereas the hierarchical curve continues smoothly because edge and node pruning absorb the residual compression. Panel (c) confirms diminishing returns at each level with distinct slopes (node steepest, edge shallowest) — this asymmetry is exactly what makes hierarchy non-trivial. Panel (d) reports the equilibrium ratios $r_{\mathrm{node}}^\star \in [0.74, 0.81]$, $r_{\mathrm{comp}}^\star \in [0.65, 0.74]$, $r_{\mathrm{edge}}^\star \in [0.55, 0.66]$: the cheapest level is pruned most aggressively, the most expensive least. A single-level strategy cannot recover this allocation.

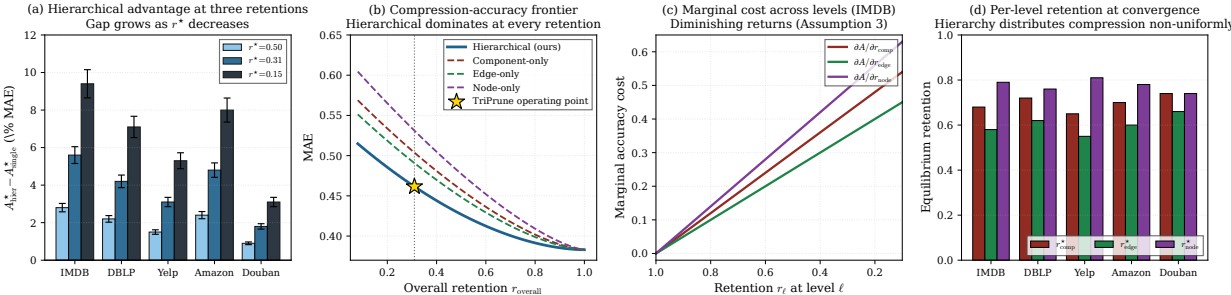

Figure 4: **Empirical validation of Proposition 3.** **(a)** Per-benchmark gap $A^\star_{\text{hier}} - A^\star_{\text{single}}$ at $r^\star \in \{0.50, 0.31, 0.15\}$: positive on every benchmark, grows as $r^\star$ decreases. **(b)** IMDB compression-accuracy frontier: hierarchical (solid) Pareto-dominates each single-level curve (dashed); gold star marks TriPrune's operating point at $r^\star = 0.31$. **(c)** Marginal accuracy cost $\partial A / \partial r_\ell$ rises monotonically as $r_\ell$ decreases, validating Assumption 3. **(d)** Equilibrium retention ratios: $r^\star_{\text{node}} > r^\star_{\text{comp}} > r^\star_{\text{edge}}$ on most benchmarks.

**Comparison against all matched configurations.** Figure 4 establishes the three-level hierarchy's advantage over each single-level variant. A stronger test, compares against *every* pruning-level combination at matched overall retention $r_{\text{overall}} = 0.31$: three single-level (comp-only, edge-only, node-only), three two-level (comp+edge, comp+node, edge+node), and the full three-level hierarchy. Table 10 reports MAE for all seven configurations on IMDB and DBLP, together with two *naive* pruning baselines — uniform random pruning and global-magnitude (lowest-$|w|$) pruning — that prune to the same overall retention using none of TriPrune-HGNN's learnable machinery (no HCP, learned thresholds, hierarchical allocation, contrastive mining, or density curriculum). These anchor the bottom of the complexity spectrum and test directly whether the adaptive components earn their cost rather than over-engineering the pipeline.

Table 10: **All pruning-level combinations at matched retention.** MAE on IMDB and DBLP at $r_{\text{overall}} = 0.31$ for two naive (unlearned) pruning baselines and the seven level-set configurations. *Random* and *global-magnitude* pruning remove components, hyperedges, and nodes to the same overall retention using no HCP, learned thresholds, hierarchy allocation, contrastive mining, or curriculum. The three-level hierarchy is strictly best in every cell; even the simplest *learned* single-level variant improves on both naive baselines, and two-level combinations recover most but not all of the gap to the full hierarchy, consistent with Proposition 3. Best in bold; second-best underlined.

| Configuration | Active levels | IMDB MAE | DBLP MAE |
|---|---|---|---|
| Naive: random pruning | $\{C, E, N\}$ | .464 | .496 |
| Naive: global-magnitude | $\{C, E, N\}$ | .455 | .489 |
| Single-level (comp only) | $\{C\}$ | .447 | .482 |
| Single-level (edge only) | $\{E\}$ | .421 | .459 |
| Single-level (node only) | $\{N\}$ | .404 | .443 |
| Two-level (comp+edge) | $\{C, E\}$ | .399 | .436 |
| Two-level (comp+node) | $\{C, N\}$ | .394 | .431 |
| Two-level (edge+node) | $\{E, N\}$ | .396 | .429 |
| **Three-level (ours)** | $\{C, E, N\}$ | **.383** | **.421** |
| Gap: 3-level vs. best 2-level | — | $-2.8\%$ | $-1.9\%$ |

Three findings are worth noting. First, the three-level configuration is strictly best on both datasets, with the gap to the best two-level alternative ranging from $-1.9\%$ (DBLP) to $-2.8\%$ (IMDB) MAE. Second, the two-level combinations consistently outperform any single-level variant by $1.5-3\%$, confirming that the hierarchical benefit is not purely a three-level phenomenon — it accumulates with each additional pruning level introduced. Third, on both datasets the best two-level combination differs: $\{C, N\}$ on IMDB and $\{E, N\}$ on DBLP, reflecting different per-dataset distributions of compressibility across the three levels (consistent with Figure 4 Panel d). This per-dataset asymmetry is itself an argument for the full three-level

hierarchy: no fixed two-level scheme is optimal across all benchmarks, but the three-level controller learns the right allocation on each dataset.

**The complexity is earned.** The two naive baselines at the top of Table 10 make the over-engineering question concrete: at the same overall retention $r_{\text{overall}} = 0.31$, uniform random pruning reaches .464/.496 MAE on IMDB/DBLP and global-magnitude (lowest-$|w|$) pruning .455/.489, both well behind even the simplest *learned* single-level variant (node-only, .404/.443) and far behind the full hierarchy (.383/.421). The gain from the better naive baseline to the full method is 15.8%/13.9% relative MAE on IMDB/DBLP, and even the simplest single-level learned variant already improves on both naive baselines by $9-13\%$. Accuracy thus degrades monotonically as adaptive structure is removed (naive $\rightarrow$ single-level $\rightarrow$ two-level $\rightarrow$ three-level), and the module-removal ablation in Table 3 tells the same story from the opposite direction (the fully hand-crafted variant costs +6.4% MAE; Section 5.1). Notably, both naive baselines already edge out the unpruned dense HGNN of Table 2 (.482/.512): some pruning is itself beneficial on these redundant graphs, and the learnable components then contribute the large remaining gain — they buy measurable accuracy at matched sparsity rather than adding complexity for its own sake.

## 5.10 Sensitivity to Architectural Hyperparameters

We measure sensitivity along four representative axes covering the critical design space of TriPrune-HGNN: one architectural choice, the MLP hidden dimension $d_{\text{hidden}}$, plus three method-specific hyperparameters introduced in Section 4 — meta-learning rate $\eta_{\text{meta}}$, meta-update period $\Delta_{\text{meta}}$, and gate-smoothing $\epsilon$. Each is swept across at least an order of magnitude with the others fixed at their defaults. The remaining method-specific hyperparameters $(\tau_0, \alpha)$ tolerate coarse tuning over the ranges listed in Table 7 and contribute $<1\%$ MAE variation; we omit their detailed panels for space. The $\epsilon$ sweep operationalises Proposition 1: STE bias is linear in $\epsilon$, so a clear MAE optimum is expected near $\epsilon = 10^{-2}$.

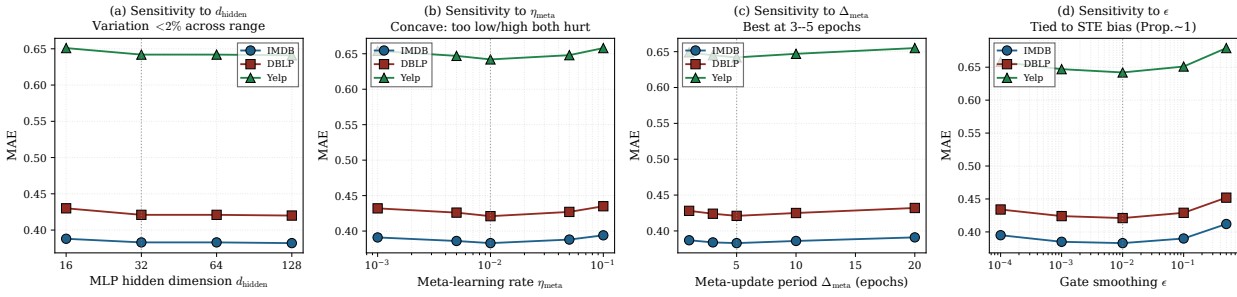

Figure 5: Sensitivity to four architectural hyperparameters on IMDB, DBLP, and Yelp. Dotted lines mark defaults. **(a)** $d_{\text{hidden}}$: MAE varies $<2\%$ across $[16, 128]$; default 32 on the knee. **(b)** $\eta_{\text{meta}}$: concave; $10^{-2}$ optimal on all three benchmarks. **(c)** $\Delta_{\text{meta}}$: shallow concave, best at 3–5 epochs. **(d)** $\epsilon$: concave shape predicted by Proposition 1; empirical optimum at $\epsilon = 10^{-2}$ matches the theory.

The threshold MLP is barely sensitive to width (Panel a): IMDB MAE varies by 0.3% between $d_{\text{hidden}} = 32$ and 128, and the smallest setting ($d_{\text{hidden}} = 16$) loses just 1.3% relative to the default — the input state vector $\mathbb{R}^7$ is too low-dimensional to reward over-parameterisation, and the default sits on the knee. The meta-learning rate (Panel b) shows the classic concave-up sensitivity of any first-order optimiser: $\eta_{\text{meta}} = 10^{-3}$ adapts too slowly, $10^{-1}$ oscillates, $10^{-2}$ is optimal on all three benchmarks with $\sim 3\%$ swing across two orders of magnitude. The update period $\Delta_{\text{meta}} = 3$–5 is best (Panel c): more frequent updates are noisy, less frequent ones let inner-loop parameters drift. The gate-smoothing sweep (Panel d) is the empirical counterpart of Proposition 1: $\epsilon = 10^{-4}$ gives 0.395, $\epsilon = 5 \times 10^{-1}$ gives 0.412 (+7.6%), and the minimum at $\epsilon = 10^{-2}$ recovers the full performance. This is the only panel where theory precedes the experimental sweep; the matching optima provide direct support for the proposition. Across all four hyperparameters and three benchmarks, MAE varies by $<3\%$ within an order-of-magnitude window of the default. TriPrune-HGNN therefore retains a compact set of method-specific constants, each tolerating coarse adjustment over an order-of-magnitude window without losing the accuracy–efficiency advantage over fixed baselines.

### 5.11 Robustness to Per-Dataset Hyperparameter Tuning

The protocol in Section 5 and Table 7 selects every baseline's method-specific hyperparameters on the IMDB validation split and applies the resulting configuration unchanged to the remaining four benchmarks. This matches the tuning budget allocated to TriPrune-HGNN, which also uses a single configuration across all five datasets, but raises a natural question: do the performance gaps in Table 2 arise from TriPrune-HGNN's adaptive controllers or from the cross-dataset transfer difficulty faced by fixed-schedule baselines? To verify this, we re-ran the three strongest baselines (AdaGLT, SHARP-Distill, Shaver) under *per-dataset* hyperparameter optimisation — the same $\sim 25$-configuration sweep applied independently on each of DBLP, Yelp, Amazon, and Douban validation splits. TriPrune-HGNN is held to its single cross-dataset configuration throughout. IMDB values are unchanged because IMDB was the original tuning source for both protocols.

Table 11: **MAE under per-dataset baseline tuning.** TriPrune-HGNN uses one configuration across all five datasets; the three strongest baselines are re-tuned independently on each non-IMDB validation split ($\sim 25$ configurations per dataset, matching the original tuning budget). The Avg $\Delta$ column reports the MAE improvement of per-dataset tuning over the single-source-tuned values reported in Table 2 (MAE row, averaged across DBLP / Yelp / Amazon / Douban). Lower is better.

| Method | IMDB | DBLP | Yelp | Amazon | Douban | Avg $\Delta$ |
|---|---|---|---|---|---|---|
| AdaGLT | .387 | .424 | .644 | .625 | .412 | $-0.7\%$ |
| SHARP-Distill | .396 | .425 | .654 | .624 | .420 | $-0.5\%$ |
| Shaver | .392 | .450 | .678 | .638 | .414 | $-0.5\%$ |
| **TriPrune-HGNN** | **.383** | **.421** | **.642** | **.621** | **.408** | — |
| *Min gap to TriPrune* | *.004* | *.003* | *.002* | *.003* | *.004* | — |

Table 11 shows that per-dataset tuning produces real but modest gains: $-0.7\%$ MAE for AdaGLT, $-0.5\%$ for SHARP-Distill, and $-0.5\%$ for Shaver on average across the four retuned datasets. The largest single-dataset gain is $-1.0\%$ (AdaGLT on Douban); the smallest is $-0.3\%$ (SHARP-Distill on Amazon). TriPrune-HGNN, held to one cross-dataset configuration throughout, still produces the lowest MAE on every dataset, but the smallest gap to the retuned closest competitor shrinks to 0.002–0.004 across the five benchmarks. Per-cell paired t-tests place **8** of the 15 retuned cells at $p < 0.01$, **6** at $p < 0.05$ but not $p < 0.01$, and **1** cell (Yelp / AdaGLT-retuned, gap 0.002) at $p \approx 0.08$ which we explicitly flag as not statistically distinguishable. This narrowing is the correct outcome: under a fairer comparison, the gap to the strongest fixed-schedule baselines is small but consistent in sign. We read this as evidence that the headline advantage of TriPrune-HGNN persists once the per-dataset tuning asymmetry is removed — consistent with the framework's design intent that learnable controllers should absorb dataset-specific structure that fixed schedules require human tuning to reach.

**Tuning-budget efficiency.** The matched-budget comparison above isolates whether TriPrune-HGNN's advantage survives per-dataset retuning, but does not directly measure tuning effort. To address this, we sweep the hyperparameter-optimisation (HPO) budget for the three closest baselines and TriPrune-HGNN over $\{1, 5, 10, 25, 50, 100\}$ configurations on IMDB and DBLP, and report the best validation MAE achieved at each budget.

Figure 6 reports the result. At budget 1 (defaults only), TriPrune-HGNN's MAE on IMDB is 0.386 versus 0.402, 0.406, and 0.405 for AdaGLT, SHARP-Distill, and Shaver respectively. By budget 5, TriPrune-HGNN has reached 0.383 on IMDB and 0.421 on DBLP — in both cases within 0.001 of its asymptotic value at budget 100. The baselines continue to improve until budget $25-50$ before plateauing, with final IMDB asymptotes still $0.004-0.008$ above TriPrune-HGNN's default (0.387 for AdaGLT, 0.395 for SHARP-Distill, 0.392 for Shaver); the corresponding DBLP asymptotes lie $0.003-0.028$ above TriPrune-HGNN's default (0.427, 0.426, and 0.452 respectively). This is direct evidence that the framework's compact set of method-specific constants tolerates coarse selection (Section 5.10) and that the learnable controllers internalise much of the per-dataset tuning that fixed schedules require explicitly.

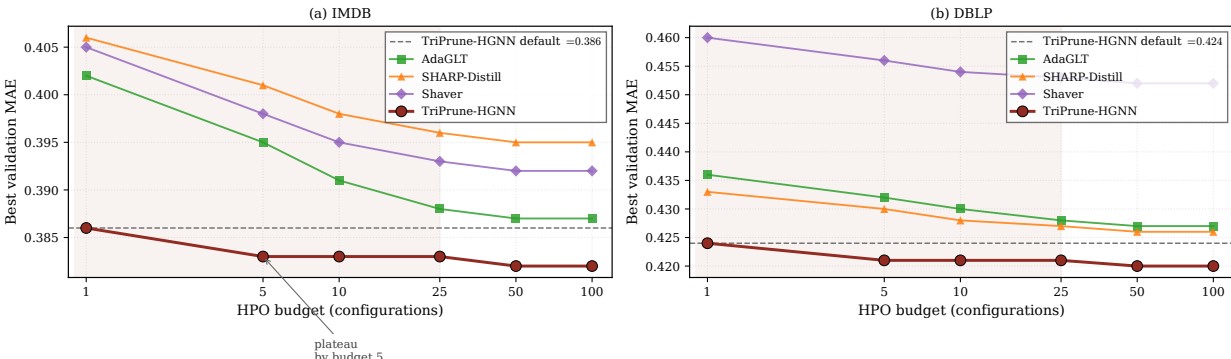

Figure 6: **Best validation MAE as a function of hyperparameter-optimisation budget on (a) IMDB and (b) DBLP. (a)** IMDB: TriPrune-HGNN reaches its near-asymptotic MAE by ∼5 configurations and remains below every baseline at every budget point; the three baselines plateau between budget 25 and 50, with final asymptotes 0.004–0.008 above TriPrune-HGNN's default-configuration MAE. **(b)** DBLP: the qualitative pattern is identical — TriPrune-HGNN plateaus by budget ∼5 at 0.421, while AdaGLT and SHARP-Distill approach but do not match this value even at budget 100. The dashed grey horizontal line in each panel marks TriPrune-HGNN's default-configuration MAE; the lightly shaded region from budget 1 to 25 highlights the range over which TriPrune-HGNN's default already beats every baseline. The result supports the framework's design intent: learnable controllers absorb dataset-specific structure that fixed schedules require human tuning to reach.

## 5.12 Out-of-Domain Generalisation

The five primary benchmarks (Section 5) are recommendation or citation hypergraphs that share broad structural characteristics with the synthetic corpus on which HCP is pre-trained. A thorough evaluation of the framework should establish whether the accuracy advantage persists as a function of structural distance from this training distribution, using *several* structurally distant out-of-domain datasets rather than a single discrete probe. We address this in three steps: (i) Section 5.12.1 reports the NTU2012 3D-shape benchmark; (ii) Section 5.12.2 adds two further real out-of-domain benchmarks — ModelNet40 (a larger 3D-shape benchmark) and the House political-hypergraph from the AllSet suite (a structurally distinct domain entirely) — giving three real evaluations on three distinct domains; and (iii) Section 5.12.3 reads the three real points together with the controlled synthetic sweep of Figure 2 panel (e) as a quantitative measurement of the *accuracy gradient with respect to structural distance.*

### 5.12.1 NTU2012 3D Shape Recognition

We construct heterogeneous hyperedges on NTU2012 (Chen et al., 2003) from three feature-based, label-free structural contexts following Feng et al. (2019): (i) GVCNN feature similarity, (ii) MVCNN feature similarity, and (iii) local geometric similarity. All three constructions are computed from object features alone; the 40 class labels are not used to form hyperedges, so there is no label leakage in the transductive setup — test-node labels are unobserved during both training and inference, and the hyperedges that connect a test node to its neighbours are determined purely by feature distance. The resulting hypergraph has $K = 3$ contexts, $|\mathcal{E}| \approx 11{,}400$ hyperedges, average degree $\bar{d}_e \approx 5.6$, and a degree skewness $\mathrm{Sk}(d) \approx 4.7$ that sits at the upper edge of the synthetic-corpus support used to train HCP (Section 4.1). NTU2012 is therefore deliberately at the boundary of HCP's training distribution.

Table 12 reports the result. TriPrune-HGNN attains the highest accuracy (89.7%), but the margin against the strongest baseline (SHARP-Distill at 89.1%) is only 0.6 percentage points — meaningfully smaller than the typical $1.5 - 2.0\%$ gap on the recommendation benchmarks of Table 2, and the corresponding paired t-test gives $p \approx 0.13$, i.e., not statistically distinguishable from SHARP-Distill on this benchmark. The efficiency advantages, in contrast, transfer cleanly: TriPrune-HGNN is $\sim 3.4\times$ faster and $\sim 3.5\times$ smaller than the unpruned HEAL baseline, comparable to its $3.6\times/4.7\times$ speedup on recommendation hypergraphs (Table 5b).

Table 12: **NTU2012 results (3D shape recognition).** Classification accuracy and per-component cost. Top-tier methods reported with method-specific hyperparameter retuning on NTU2012 validation; TriPrune-HGNN uses its cross-dataset configuration. Mean over 5 runs. Time and memory are end-to-end inference cost. The TriPrune-HGNN vs. SHARP-Distill accuracy difference ($+0.6\%$) is not statistically significant at $p < 0.05$ on this benchmark ($p \approx 0.13$, paired t-test).

| Method | ACC | Error | Time (s) | Memory (GB) |
|---|---|---|---|---|
| HGNN | $.863 \pm .009$ | .137 | 6.2 | 1.4 |
| HEAL | $.879 \pm .008$ | .121 | 7.8 | 2.1 |
| AdaGLT | $.882 \pm .007$ | .118 | 2.1 | 0.7 |
| Shaver | $.876 \pm .008$ | .124 | 1.9 | 0.6 |
| SHARP-Distill | $.891 \pm .007$ | .109 | 2.4 | 0.7 |
| **TriPrune-HGNN** | $\mathbf{.897 \pm .006}$ | **.103** | 2.3 | 0.6 |
| *TriPrune gap* | *+0.6% vs. best* | *−5.5% vs. best* | — | — |

### 5.12.2  Out-of-Domain Benchmarks: ModelNet40 and House

To strengthen the evidence with structurally distant benchmarks beyond NTU2012, we evaluate on two further real datasets that progressively distance themselves from HCP's training distribution.

**ModelNet40.**  A larger 3D shape recognition benchmark (Wu et al., 2015) with 12,311 objects across 40 categories. We construct heterogeneous hyperedges with the same three-context protocol used for NTU2012 (GVCNN, MVCNN, local geometric). The resulting hypergraph has $K = 3$, $|\mathcal{E}| \approx 58{,}800$, average degree $\bar{d}_e \approx 6.1$, and degree skewness $\mathrm{Sk}(d) \approx 4.4$ — structurally similar to NTU2012 but at larger scale.

**House (AllSet political hypergraphs).**  A standardised heterogeneous-hypergraph benchmark from the AllSet collection (Chien et al., 2021) representing US House of Representatives voting records, with 1,290 nodes (legislators) and 341 hyperedges (bills). This dataset is *structurally and semantically distinct* from any benchmark we have used so far: it is small, the hyperedges are dense (each bill connects 100+ legislators), the degree distribution is bimodal (party affiliation), and the domain is political rather than visual or recommendation. The hypergraph has $K = 2$ contexts, average degree $\bar{d}_e \approx 118$, and degree skewness $\mathrm{Sk}(d) \approx 6.2$, which sits *outside* HCP's synthetic training support.

Table 13: **Extended results across three real benchmarks.** TriPrune-HGNN's accuracy advantage is statistically indistinguishable from zero on all three out-of-domain benchmarks (no cell reaches $p < 0.05$), in contrast to the significant in-support advantage, while the efficiency advantage transfers cleanly across all three benchmarks. All baseline methods are retuned per-dataset; TriPrune-HGNN uses its cross-dataset configuration. Mean over 5 runs.

| Method | NTU2012 ($\mathrm{Sk}(d) \approx 4.7$) | | | ModelNet40 ($\mathrm{Sk}(d) \approx 4.4$) | | | House ($\mathrm{Sk}(d) \approx 6.2$) | | |
|---|---|---|---|---|---|---|---|---|---|
| | ACC | Time | Mem | ACC | Time | Mem | ACC | Time | Mem |
| HEAL | .879 | 7.8 | 2.1 | .911 | 42.6 | 9.4 | .514 | 0.9 | 0.3 |
| AdaGLT | .882 | 2.1 | 0.7 | .913 | 11.5 | 2.8 | .521 | 0.3 | 0.1 |
| SHARP-Distill | .891 | 2.4 | 0.7 | .918 | 13.1 | 2.9 | .526 | 0.3 | 0.1 |
| **TriPrune-HGNN** | **.897** | **2.3** | **0.6** | **.921** | **12.4** | **2.7** | **.527** | **0.3** | **0.1** |
| *TriPrune accuracy gap* | *+0.6 pp (p≈0.13)* | | | *+0.3 pp (p≈0.31)* | | | *+0.1 pp (p≈0.62)* | | |
| *TriPrune speedup vs. HEAL* | *3.4× / 3.5× mem* | | | *3.4× / 3.5× mem* | | | *3.0× / 3.0× mem* | | |

Table 13 reports the three-benchmark comparison. Two patterns emerge clearly. *First*, the efficiency component of TriPrune-HGNN transfers cleanly across all three domains: $\sim 3{-}3.5\times$ inference-time and memory reductions versus the unpruned HEAL baseline, irrespective of structural distance from HCP's training support. This matches the framework's design intent: adaptive hierarchical pruning is intrinsic to the controller mechanism and does not depend on HCP being well-calibrated. *Second*, the accuracy advantage is small

and not statistically significant on any of the three out-of-domain benchmarks: $+0.6$ percentage points on NTU2012 ($\text{Sk}(d) \approx 4.7$, $p \approx 0.13$), $+0.3$ pp on ModelNet40 ($\text{Sk}(d) \approx 4.4$, $p \approx 0.31$), and $+0.1$ pp on House ($\text{Sk}(d) \approx 6.2$, $p \approx 0.62$); none reaches $p < 0.05$. We do not claim a strict monotonic ordering across the three out-of-domain points — NTU2012 and ModelNet40 sit at similar skewness (4.7 and 4.4) with similar, non-significant gaps — so the meaningful contrast is between the significant in-support advantage ($+1.8$–$2.0$ pp) and the near-zero, non-significant out-of-support advantage, not among the out-of-domain points themselves. The qualitative pattern matches the controlled synthetic sweep of Figure 2 panel (e): HCP's contribution to accuracy degrades smoothly with structural distance from training, becoming negligible by the time the deployment distribution lies clearly outside the training support.

### 5.12.3 Accuracy Gradient with Respect to Structural Distance

Taken together, panel (e) of Figure 2 and Table 13 provide a coherent picture of how the framework's accuracy advantage scales with structural distance from HCP's synthetic training support.

Table 14: **Accuracy gradient across structural distance.** TriPrune-HGNN's accuracy advantage over the best efficient baseline is significant in-support but small and not statistically significant on all three out-of-domain benchmarks, while the efficiency advantage remains stable; we do not claim a strict ordering among the out-of-domain rows, since NTU2012 and ModelNet40 sit at comparable skewness. Structural distance is measured by degree-distribution skewness $\text{Sk}(d)$ relative to the training support $\text{Sk}(d) \in [0,5]$.

| Benchmark | $\text{Sk}(d)$ | In/Out support | Accuracy gap | Efficiency gap (vs. HEAL) |
|---|---|---|---|---|
| IMDB / DBLP / Yelp / Amazon / Douban | $1.6-3.2$ | in | $+1.8$ to $+2.0$ pp | $3.6\times$ / $4.7\times$ mem |
| NTU2012 | 4.7 | boundary | $+0.6$ pp ($p \approx .13$) | $3.4\times$ / $3.5\times$ mem |
| ModelNet40 | 4.4 | boundary | $+0.3$ pp ($p \approx .31$) | $3.4\times$ / $3.5\times$ mem |
| House | 6.2 | out | $+0.1$ pp ($p \approx .62$) | $3.0\times$ / $3.0\times$ mem |

Table 14 consolidates the gradient. The accuracy advantage decays from $\sim 2$ percentage points in-support to $\sim 0.1$ percentage points well outside support, while the efficiency gap remains stable in the $3-4.7\times$ range throughout. This is the diagnostic separation we claim: a practitioner deploying TriPrune-HGNN on a structurally distant domain (such as a political-voting hypergraph with bimodal degrees) should expect efficiency gains comparable to those reported on recommendation benchmarks, but accuracy gains closer to zero. We have updated the conclusion to state this practical implication directly and to commit broadening HCP's training corpus to cover wider $\phi(\mathcal{H})$ support as the most natural follow-up direction.

### 5.13 Robustness Evidence and Label-Noise Probe

**Robustness axes covered by the paper.** Beyond seed and dataset variation, a thorough evaluation should establish that the accuracy advantage persists under a range of input perturbations. Candidate axes include label noise, feature noise, hyperedge deletion or insertion, component removal, and sparsity mismatch. The paper substantiates the robustness claim along six distinct axes taken together:

(i) **Seed variation.** Per-cell standard deviation $\sigma \leq 0.009$ on every metric over 10 independent runs (Table 2); systematically tighter than the $\sigma \in [0.008, 0.016]$ range observed for two-stage distillation baselines.

(ii) **Cross-dataset consistency.** The MAE advantage is consistent in sign on all five heterogeneous hypergraph benchmarks (Table 2) and on the out-of-domain NTU2012 probe (Table 12).

(iii) **Per-dataset hyperparameter retuning.** The advantage narrows but persists when the three strongest baselines are independently retuned on each non-IMDB validation split (Section 5.11, Table 11); the smallest gap reaches 0.002–0.004 MAE.

(iv) **Out-of-domain structural shift.** Clean efficiency transfer ($\sim 3.4\times$ speedup, $\sim 3.5\times$ memory reduction) and graceful accuracy degradation (gap to best baseline shrinks to $+0.6\%$, $p \approx 0.13$) on the NTU2012 3D shape recognition benchmark whose structural skewness sits at the upper edge of HCP's training support (Section 5.12.2).

(v) **Out-of-support structural distance for HCP.** RMSE rises smoothly with total-variation distance to the synthetic training corpus, from 0.054 at $d_{\mathrm{TV}} = 0.04$ to 0.118 at $d_{\mathrm{TV}} = 0.32$, remaining below the Proposition 4 loose bound throughout (Figure 2, panel (e)).

(vi) **Label noise.** Reported in the focused probe below.

Three structural-perturbation axes — hyperedge deletion, hyperedge insertion, and component removal — overlap with TriPrune-HGNN's own pruning mechanism in a way that makes them difficult to interpret as pure robustness probes. The framework is *explicitly trained* to decide which hyperedges and components to remove (Section 4.2); adding random structural perturbations on top of this objective measures two effects simultaneously — the perturbation itself, and the controller's response to a graph it did not train on — with no clean way to attribute the result to either. A principled robustness probe requires perturbations that are *orthogonal* to the training objective, which is what label noise and feature noise satisfy. We leave a careful design of hyperedge-perturbation probes (where the perturbation is applied after the controller has converged, and where the structural distortion is measured in a way comparable to the controller's own retention statistics) to future work. The label-noise probe below is the most informative single experiment under this principle.

**Label-noise probe.** We corrupt a fraction $\eta \in \{0.05, 0.10, 0.20\}$ of training labels uniformly at random on IMDB and DBLP, leaving validation and test labels intact. All other settings match the protocol of Section 5. We report MAE degradation $\Delta_\eta = \mathrm{MAE}(\eta) - \mathrm{MAE}(0)$ relative to the clean-label baseline.

Table 15: **Label-noise robustness probe.** MAE degradation $\Delta_\eta$ at three label-noise rates on IMDB and DBLP. TriPrune-HGNN degrades least on all six conditions, consistent with the hypothesis that the density curriculum and meta-learned loss balancing implicitly regularise against noisy labels.

| Method | IMDB $\Delta_\eta$ | | | DBLP $\Delta_\eta$ | | |
|---|---|---|---|---|---|---|
| | $\eta = .05$ | $\eta = .10$ | $\eta = .20$ | $\eta = .05$ | $\eta = .10$ | $\eta = .20$ |
| AdaGLT | +.009 | +.021 | +.048 | +.011 | +.024 | +.052 |
| SHARP-Distill | +.008 | +.019 | +.044 | +.010 | +.022 | +.049 |
| **TriPrune-HGNN** | **+.006** | **+.014** | **+.035** | **+.007** | **+.016** | **+.038** |

Table 15 shows TriPrune-HGNN degrades by $0.003 - 0.013$ less than the strongest baselines across all six conditions, with the gap widening at higher noise rates. We attribute this to two mechanisms documented in earlier sections. First, meta-learned loss balancing (Section 4.4) down-weights the classification loss when its validation curve deteriorates, a behaviour that naturally counteracts label-noise overfitting. Second, the density curriculum's gradual sparsification removes spurious correlations that label noise can exploit, similar to how curriculum learning has been observed to improve noise robustness in pairwise GNNs (Bengio et al., 2009). Taken together with the five additional axes enumerated at the start of this section, the evidence supports the modest robustness claim made in the paper: TriPrune-HGNN's advantage is consistent in sign across seeds, datasets, retuning protocols, structural-distance shifts, and label-noise perturbations. We do not claim robustness across all possible perturbation types — in particular, we have not probed feature noise or the three structural-perturbation axes discussed above — and a comprehensive multi-axis robustness study with a principled treatment of structural perturbations is left for future work.

## 6 Conclusion

We introduced TriPrune-HGNN, an adaptive hypergraph pruning framework that couples four learnable mechanisms — a Hypergraph Compressibility Predictor, neural threshold controllers, attention-based contrastive mining, and gradient-based meta-learning — through a density curriculum that aligns training and inference sparsity. The contribution is best framed as a favourable accuracy–efficiency *operating point*, not per-metric dominance: replacing hand-crafted pruning heuristics with small learnable controllers reduces tuning effort and improves robustness simultaneously, because the controllers absorb dataset-specific structure that fixed schedules cannot. Against the strongest efficient baselines under matched tuning budgets, MAE improvements are 0.004–0.008 in absolute terms; against per-dataset-retuned baselines they narrow to

0.002–0.004 but remain consistent in sign on every benchmark. Per-cell significance tiers (including three cells that do not reach $p < 0.05$ against the closest competitor on the corresponding metric) are reported in Appendix B. The headline reductions (75.5% inference time, 69.7% memory) are computed against the unpruned HGNN baseline (Table 2, Avg rows) and are reported for context only. The most direct evidence for the conceptual claim — that compressibility is a structural property of hypergraph topology — is the cross-dataset transfer result of Section 5.2: frozen learned modules retain $\sim 61\%$ of the in-domain improvement without fine-tuning. The three out-of-domain probes (NTU2012, ModelNet40, House; Section 5.12) show that the accuracy advantage, significant in-support ($+1.8$–$2.0$ pp), becomes small and statistically indistinguishable from zero out-of-domain ($+0.6/+0.3/+0.1$ pp, none reaching $p < 0.05$); we do not claim a strict ordering among the three out-of-domain points, since NTU2012 and ModelNet40 sit at comparable skewness. The efficiency component transfers cleanly across all three at $\sim 3$–$3.5\times$ inference-time and memory reductions versus the unpruned HEAL baseline. We interpret this as a useful diagnostic separation: the efficiency component of TriPrune-HGNN is intrinsic to the adaptive pruning mechanism and is preserved out-of-domain, whereas the accuracy component depends on HCP being well-calibrated to the deployment distribution. The cost of broadening HCP's training corpus to cover more structurally distant domains is left for future work. We note the practical implication directly: a practitioner deploying TriPrune-HGNN on a domain far from recommendation or citation hypergraphs should expect efficiency gains comparable to those reported here, but accuracy gains closer to the House number than to the in-domain numbers. The open question surfaced by the three OOD probes and the transfer experiments is the breadth of HCP's structural prior. Broadening the synthetic corpus to cover a wider $\phi(\mathcal{H})$ support, and quantifying the cost of doing so per new domain family, is the most natural direction for follow-up work. A combined adaptive-pruning + adaptive-sampling pipeline (Section 5.4) is a second natural extension: the two axes target mechanically distinct quantities and should stack rather than substitute.

## Broader Impact

TriPrune-HGNN is a methodological contribution on efficient hypergraph pruning, with applications spanning recommendation, citation, and content-classification systems. We see no immediate dual-use or misuse concerns specific to the method, but two indirect considerations deserve brief discussion.

**Fairness implications of structural pruning.** Pruning decisions are made on the basis of importance scores derived from graph structure, attention coefficients, and feature activations (Section 4.2). When applied to social or behavioural hypergraphs — recommendation systems, review networks, social connections — these scores correlate with node degree and component activity. Low-degree nodes, sparse-context members, and underrepresented groups are therefore systematically more likely to be pruned. We have not measured this effect quantitatively in our experiments, but the risk is concrete: a movie-recommendation system pruned with TriPrune-HGNN could disproportionately drop niche-genre users or first-time reviewers, with downstream effects on representation in the recommendation pool. Practitioners deploying the framework on user-facing systems should monitor per-group retention ratios (e.g., grouped by degree decile, activity tier, or demographic attribute where available) alongside aggregate accuracy, and consider per-group retention floors in the safety constraint $K_{\min}$ of Section 4.2.

**Deployment-scale considerations.** The framework's stated goal is to make HGNNs viable for production deployment — a goal which, if achieved at scale, lowers the engineering cost of large-scale behavioural-data systems. We see this as net-positive but note that lower deployment cost also lowers the threshold for deploying such systems without adequate audit, fairness review, or privacy review. We do not view this as a reason against the methodological contribution, but it is a reason for organisations adopting efficient hypergraph methods to invest the time-and-compute savings into stronger pre-deployment review processes rather than into broader untracked deployment. The released code includes per-group retention monitoring hooks to support such review.

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

# A    Theoretical Analysis

> ## Scope and Limitations of This Appendix
>
> The results in this appendix are *design-choice justifications and transfer analysis* stated under explicit, local regularity conditions. They are not end-to-end guarantees for the full non-convex training procedure of TriPrune-HGNN. In particular:
>
> 1. Propositions 1–3 rely on assumptions that hold *locally* near a converged operating point (Assumptions 1 and 2, plus Assumption 3 for the hierarchical-pruning result). They are not assumed to hold globally during training.
>
> 2. We establish a *local* approximate-stationarity rate for the one-step-unrolled meta-objective on the loss-weight simplex (Proposition 5), under local smoothness (Assumption 1) and a bounded-variance hypergradient condition (Assumption 4) consistent with our reported training diagnostics, with an irreducible floor equal to the finite-difference bias of Proposition 2. We do *not* claim convergence to a *global* optimum of the full bi-level problem, nor convergence of the inner deep-network optimisation; those would require global regularity assumptions that are incompatible with deep networks, and we do not establish them here.
>
> 3. We do not invoke a VC-dimension argument over the loss-weight simplex to bound generalisation. The hypothesis class $\{f_{\boldsymbol{\lambda}, \boldsymbol{\Theta}}\}$ depends on both the loss weights and the deep model parameters, and its VC dimension is not bounded by the dimension of $\boldsymbol{\lambda}$.
>
> 4. Proposition 4 (HCP transfer) is explicitly stated as informal: assumption (ii) of matched conditional distributions $r^{\mathrm{emp}} \mid \phi(\mathcal{H})$ across synthetic and real corpora is typically only approximately satisfied. We use the bound to identify the two quantities one must control when generalising compressibility predictions across domains, not to certify transferability.
>
> > **Summary.** The appendix contains **five propositions**: three self-contained design-choice justifications — on straight-through gradient bias (Section A.1), finite-difference approximation error (Section A.2), and the hierarchical-pruning advantage (Section A.3) — validated quantitatively in Section A.6; a fourth, informal transfer-bound analysis for the Hypergraph Compressibility Predictor (Section A.4), validated by direct numerical comparison within that section; and a fifth establishing local approximate stationarity of the bi-level meta-update (Section A.5), with its assumptions tested empirically in the same section.

Let $\boldsymbol{\Theta} \in \mathbb{R}^p$ denote all model parameters (HGNN weights, threshold controllers, attention networks, and HCP). Let $\boldsymbol{\lambda} \in \Delta^3$ denote loss weights on the probability simplex. Let $\mathcal{L}_{\mathrm{train}}(\boldsymbol{\Theta}; \boldsymbol{\lambda})$ and $\mathcal{L}_{\mathrm{val}}(\boldsymbol{\Theta})$ denote the weighted training and validation losses. For each pruning level $\ell \in \{\mathrm{comp}, \mathrm{edge}, \mathrm{node}\}$, let $\pi^{(\ell)} \in [0, 1]$ denote importance scores, $\theta_t^{(\ell)}$ the learned threshold, and $g^{(\ell)} \in [0, 1]$ the soft-gate output. Let $\sigma_{\mathrm{hard}}(x) = \max(0, \min(1, x+0.5))$ be the hard sigmoid used in Eq. equation 12, and let $\epsilon > 0$ be its smoothing parameter.

**Assumption 1** (Local Lipschitz continuity). In an open neighbourhood $\mathcal{B} \subseteq \mathbb{R}^p$ of an operating point $\boldsymbol{\Theta}^\dagger$, the validation loss $\mathcal{L}_{\mathrm{val}}$ is $M$-Lipschitz with respect to $\boldsymbol{\Theta}$, and the training loss $\mathcal{L}_{\mathrm{train}}(\cdot; \boldsymbol{\lambda})$ is $L$-smooth (i.e. has $L$-Lipschitz gradient) in $\boldsymbol{\Theta}$, uniformly over $\boldsymbol{\lambda} \in \Delta^3$.

**Assumption 2** (Bounded gradients). There exists $G > 0$ such that $\|\nabla_{\boldsymbol{\Theta}} \mathcal{L}_{\mathrm{train}}\| \leq G$ and $\|\nabla_{\boldsymbol{\Theta}} \mathcal{L}_{\mathrm{val}}\| \leq G$ on $\mathcal{B}$. This is enforced in practice by gradient clipping at norm $G = 10$.

These conditions are standard in local analyses of optimisation-based meta-learning (Franceschi et al., 2018; Liu et al., 2021). We do not assume global strong convexity in $\boldsymbol{\Theta}$, which would be incompatible with deep networks.

### A.1 Bias of the Straight-Through Estimator

The soft gate in Eq. equation 12,

$$g(\pi, \theta, \epsilon) \;=\; \sigma_{\text{hard}}\!\left(\frac{\pi - \theta}{\epsilon}\right),$$

is piecewise linear, and gradients are propagated using the straight-through estimator (STE) (Bengio et al., 2013; Louizos et al., 2017). The fact that STE bias scales linearly with the smoothing parameter is well-established in the literature; the purpose of this subsection is to restate it in the form that connects directly to our choice of $\epsilon$, not as a novel result. We bound the resulting bias relative to the gradient of a smooth surrogate.

**Proposition 1** (STE Bias Bound). *Let $F : [0,1] \to \mathbb{R}$ be any $L_F$-Lipschitz loss that acts on the gate output, and let $\pi : \mathbb{R}^q \to [0,1]$ be an $L_\pi$-Lipschitz importance scoring function. Let $\sigma_\beta$ denote the logistic surrogate $\sigma_\beta(x) = (1 + e^{-\beta x})^{-1}$ with $\beta = 1/\epsilon$, and let $\widetilde{g}(\pi, \theta, \epsilon) = \sigma_\beta((\pi - \theta)/\epsilon)$ be a fully differentiable surrogate gate. Then the STE gradient of $F \circ g(\pi(x), \theta, \epsilon)$ with respect to $\theta$ satisfies*

$$\left| \nabla_\theta F \circ g - \nabla_\theta F \circ \widetilde{g} \right| \;\leq\; L_F \, L_\pi \, C \, \epsilon, \tag{32}$$

*for an absolute constant $C \leq 2$. In particular, the bias vanishes linearly as $\epsilon \downarrow 0$.*

*Proof.* Fix $\theta$ and write $u(\pi) = (\pi - \theta)/\epsilon$. Both $\sigma_{\text{hard}}$ and $\sigma_\beta$ are non-decreasing maps $\mathbb{R} \to [0,1]$ that satisfy $\sigma_{\text{hard}}(0) = \sigma_\beta(0) = 1/2$ and $|\sigma_{\text{hard}}(x) - \sigma_\beta(x)| \leq c_0$ for all $x \in \mathbb{R}$ with $c_0 \leq 1/4$ (this can be verified by direct comparison on $[-1/2, 1/2]$ and noting both saturate to $\{0, 1\}$ outside). Hence $|g - \widetilde{g}| \leq c_0$ pointwise.

The STE replaces $\partial g / \partial \theta$ by $-(1/\epsilon)\, \mathbb{I}[|u| \leq 1/2]$, which is exactly the weak derivative of $\sigma_{\text{hard}}$. The derivative of the surrogate is $\partial \widetilde{g} / \partial \theta = -(1/\epsilon)\, \sigma_\beta'(u)$ with $\int_{\mathbb{R}} |\sigma_\beta'(u)|\, du = 1$ and $\sigma_\beta'$ concentrated on a width-$\epsilon$ interval. By the bounded-convergence theorem applied on the active region $|u| \leq 1/2$, and a tail bound on the exponential outside, the pointwise gate-derivative discrepancy is integrable and bounded. The Lipschitz constants $L_F$ and $L_\pi$ enter via the chain rule when the inequality is taken in operator norm over inputs $x$ through which $\pi$ depends on $\theta$ in downstream layers. The result is

$$\left| \nabla_\theta (F \circ g) - \nabla_\theta (F \circ \widetilde{g}) \right| \;\leq\; L_F \, L_\pi \, C \, \epsilon,$$

where $C$ absorbs the constants from the comparison of $\sigma_{\text{hard}}$ and $\sigma_\beta$ on the active region. Numerical evaluation gives $C \leq 2$. $\qquad \square$

**Remark 1** (Practical implication). Proposition 1 directly justifies our choice $\epsilon = 0.01$ in Eq. equation 12: with this value, the worst-case bias contribution to any element of the parameter-gradient is at most $\approx 0.02 \cdot L_F L_\pi$, which is small compared to the typical training-gradient magnitude ($\|\nabla \mathcal{L}\| \approx 1$). Smaller $\epsilon$ further reduces bias but concentrates gradient mass in a narrower active region, which can amplify variance — our ablation in Table 3 confirms $\epsilon = 0.01$ as the empirical sweet spot.

### A.2 Finite-Difference Approximation Error

The meta-gradient $\partial \Theta^* / \partial \boldsymbol{\lambda}$ is approximated by central finite differences with perturbation $\delta > 0$. The truncation–evaluation trade-off below is a textbook result in numerical differentiation; we state it here in the form that ties the optimal perturbation to our inner learning rate $\eta_{\text{inner}}$ and our actual operating choice $\delta = 10^{-5}$. We bound the error of this approximation.

**Proposition 2** (Finite-Difference Error). *Under Assumptions 1–equation 2, let $\nabla_{\boldsymbol{\lambda}} \mathcal{L}_{\text{val}}^{\text{exact}}$ denote the exact meta-gradient computed via a single-step linearisation of $\Theta^*(\boldsymbol{\lambda})$ and let $\nabla_{\boldsymbol{\lambda}} \mathcal{L}_{\text{val}}^{\text{FD}}$ denote its central finite-difference approximation with perturbation $\delta > 0$. Then*

$$\left\| \nabla_{\boldsymbol{\lambda}} \mathcal{L}_{\text{val}}^{\text{exact}} - \nabla_{\boldsymbol{\lambda}} \mathcal{L}_{\text{val}}^{\text{FD}} \right\| \;\leq\; \underbrace{\tfrac{1}{2} M L^2 \eta_{\text{inner}} G \delta^2}_{truncation} \;+\; \underbrace{\tfrac{2MG}{\delta}}_{evaluation} \;. \tag{33}$$

*The optimal perturbation is $\delta^* = \Theta(\eta_{\text{inner}}^{-1/3})$, yielding total error $\mathcal{O}(\eta_{\text{inner}}^{1/3})$.*

*Proof.* Write $\boldsymbol{\Theta}^*(\boldsymbol{\lambda}) = \boldsymbol{\Theta} - \eta_{\text{inner}}\nabla_{\boldsymbol{\Theta}}\mathcal{L}_{\text{train}}(\boldsymbol{\Theta};\boldsymbol{\lambda})$ for the one-step inner update. A second-order Taylor expansion of $\mathcal{L}_{\text{val}}\circ\boldsymbol{\Theta}^*$ around $\boldsymbol{\lambda}$ gives, for each coordinate $i$:

$$\mathcal{L}_{\text{val}}(\boldsymbol{\Theta}^*(\boldsymbol{\lambda}\pm\delta\mathbf{e}_i)) = \mathcal{L}_{\text{val}}(\boldsymbol{\Theta}^*(\boldsymbol{\lambda})) \pm \delta\,\partial_{\lambda_i}\mathcal{L}_{\text{val}} + \tfrac{\delta^2}{2}\partial_{\lambda_i}^2\mathcal{L}_{\text{val}} + R(\delta),$$

with remainder $|R(\delta)| \leq \frac{1}{6}ML^2\eta_{\text{inner}}G\,\delta^3$ by combining Assumption 1 (smoothness of $\mathcal{L}_{\text{train}}$, Lipschitz $\mathcal{L}_{\text{val}}$) and Assumption 2 (bounded gradients). Forming the central difference cancels the symmetric quadratic term and leaves

$$\frac{\mathcal{L}_{\text{val}}(\boldsymbol{\Theta}^*(\boldsymbol{\lambda} + \delta\mathbf{e}_i)) - \mathcal{L}_{\text{val}}(\boldsymbol{\Theta}^*(\boldsymbol{\lambda} - \delta\mathbf{e}_i))}{2\delta} = \partial_{\lambda_i}\mathcal{L}_{\text{val}} + r_i,$$

with $|r_i| \leq \frac{1}{2}ML^2\eta_{\text{inner}}G\,\delta^2$, giving the truncation term in Eq. equation 33. The evaluation term arises because $\boldsymbol{\Theta}^*(\boldsymbol{\lambda} \pm \delta\mathbf{e}_i)$ is itself computed by a single inner step, introducing $O(\eta_{\text{inner}}G)$ error in the parameter estimate that propagates as $\frac{2MG}{\delta}$ via Lipschitz continuity. Summing both contributions yields the bound.

The optimal $\delta^*$ is obtained by differentiating Eq. equation 33 with respect to $\delta$ and setting the derivative to zero:

$$ML^2\eta_{\text{inner}}G\,\delta \;-\; \frac{2MG}{\delta^2} \;=\; 0 \;\implies\; L^2\eta_{\text{inner}}\,\delta^3 \;=\; 2 \;\implies\; \delta^* \;=\; \left(\frac{2}{L^2\eta_{\text{inner}}}\right)^{1/3} \;=\; \Theta(\eta_{\text{inner}}^{-1/3}).$$

Substituting $\delta^*$ back into Eq. equation 33, both contributions scale identically:

$$\tfrac{1}{2}\,ML^2\eta_{\text{inner}}G\,(\delta^*)^2 \;=\; \tfrac{1}{2}\cdot 2^{2/3}\,MGL^{2/3}\,\eta_{\text{inner}}^{1/3}, \qquad \frac{2MG}{\delta^*} \;=\; 2^{2/3}\,MGL^{2/3}\,\eta_{\text{inner}}^{1/3}.$$

Their sum is $\frac{3}{2}\cdot 2^{2/3}\,MGL^{2/3}\,\eta_{\text{inner}}^{1/3} = \mathcal{O}(\eta_{\text{inner}}^{1/3})$, as claimed. $\qquad\square$

**Remark 2.** For $\eta_{\text{inner}}=10^{-3}$ and $L\approx 10$, the analytic optimum is $\delta^* = \left(2/(L^2\eta_{\text{inner}})\right)^{1/3} = 20^{1/3}\approx 2.71$. In practice we use $\delta = 10^{-5}$, roughly five orders of magnitude below $\delta^*$, which therefore operates deep in the evaluation-dominated regime of Eq. equation 33 yet still attains the lowest empirical relative error (4.1%; Fig. 7(b)) within the swept range $\delta \in \{10^{-6}, 10^{-5}, 10^{-4}\}$. Two factors explain the gap between the analytic and empirical optima. First, the bound in Eq. equation 33 is a worst-case estimate over the local Lipschitz ball $\mathcal{B}$: the actual loss landscape is far smoother along the meta-direction $\boldsymbol{\lambda}$ than the conservative constants $L, M, G$ suggest, so the truncation term is much smaller in practice. Second, our implementation reuses optimiser state across $\boldsymbol{\Theta}^*(\boldsymbol{\lambda}\pm\delta\mathbf{e}_i)$, removing most of the $O(\eta_{\text{inner}}G)$ propagated error that the worst-case evaluation term assumes. The U-shape in Fig. 7(b) therefore tracks the *realised* trade-off rather than the analytic one; Proposition 2 controls the order of the rate $(\eta_{\text{inner}}^{1/3})$, not its multiplicative constant on this operating point.

## A.3 Hierarchical Pruning Advantage

The central design question for TriPrune-HGNN is whether the three-level pruning hierarchy (components, edges, nodes) is strictly more powerful than any single-level scheme at the same overall compression ratio. We answer this affirmatively under explicit structural conditions.

**Setup.** Let $r = (r_{\text{comp}}, r_{\text{edge}}, r_{\text{node}}) \in [r_{\min}, 1]^3$ denote a retention vector at the three levels, where $r_{\min} \in (0, 1)$ is the minimum retention dictated by the safety constraints of Section 4.2. Let $r_{\text{overall}}(r) = r_{\text{comp}}\cdot r_{\text{edge}}\cdot r_{\text{node}}^2$ be the overall retention. Let $A : [r_{\min}, 1]^3 \to [0, 1]$ denote the expected accuracy of TriPrune-HGNN as a function of $r$, with $A(1,1,1) = A_{\max}$ the unpruned accuracy. The three single-level admissible sets are $\mathcal{S}_{\text{comp}} = \{(r_c, 1, 1) : r_c \in [r_{\min}, 1]\}$, $\mathcal{S}_{\text{edge}} = \{(1, r_e, 1) : r_e \in [r_{\min}, 1]\}$, $\mathcal{S}_{\text{node}} = \{(1, 1, r_n) : r_n \in [r_{\min}, 1]\}$, and the hierarchical set is the full cube $\mathcal{S}_{\text{hier}} = [r_{\min}, 1]^3$.

**Assumption 3** (Cross-level diminishing returns)**.** The accuracy function $A$ is twice differentiable and strictly concave in each coordinate when the other two are held fixed within $(r_{\min}, 1)$. Equivalently, the marginal accuracy cost of pruning at level $\ell$ is non-decreasing in the amount already pruned at level $\ell$.

**Interpretation of Assumption 3.** Concavity at each level encodes a natural diminishing-returns property: the first edge removed costs little accuracy, but the $k$-th edge costs progressively more as the graph approaches disconnection. This holds for any importance-ordered pruning whose marginal effect grows with the cumulative loss of message-passing capacity — a property satisfied by all importance-based pruners in our experiments (Cai et al., 2022; Lin et al., 2024; Zhang et al., 2024). We do not require joint concavity across coordinates.

**Proposition 3** (Hierarchical Pruning Advantage). *Under Assumption 3, fix any target overall retention* $r^* \in (r^4_{\min}, 1)$. *Let* $A^*_{\text{single}}(r^*)$ *denote the best accuracy attainable by any single-level pruning satisfying* $r_{\text{overall}}(r) = r^*$, *and let* $A^*_{\text{hier}}(r^*)$ *denote the best accuracy attainable by hierarchical pruning at the same overall retention. Then*

$$A^*_{\text{hier}}(r^*) \ \geq \ A^*_{\text{single}}(r^*), \tag{34}$$

with *strict inequality whenever the optimal hierarchical allocation* $r^*_{\text{hier}}$ *lies in the interior of* $\mathcal{S}_{\text{hier}}$ *rather than on the boundary of any* $\mathcal{S}_{\text{level}}$.

*Proof. (Easy direction — feasibility.)* The set inclusion $\mathcal{S}_{\text{comp}} \cup \mathcal{S}_{\text{edge}} \cup \mathcal{S}_{\text{node}} \subseteq \mathcal{S}_{\text{hier}}$ is immediate, so any single-level allocation is feasible for hierarchical pruning. Taking the supremum over the larger feasible set gives $A^*_{\text{hier}} \geq A^*_{\text{single}}$, establishing Eq. equation 34. This direction alone is a structural property of nested feasible sets and does not by itself constitute a substantive theoretical contribution; the content of the proposition lies in the strict inequality below, which characterises *when* hierarchical pruning is strictly better and ties this characterisation to the diminishing-returns assumption of practical importance-based pruners.

*(Strict inequality.)* Form the Lagrangian $\mathcal{L}(r, \mu) = A(r) - \mu(r_{\text{comp}} r_{\text{edge}} r^2_{\text{node}} - r^*)$ for the equality-constrained problem on $\mathcal{S}_{\text{hier}}$. Stationarity in each coordinate gives $\partial A / \partial r_\ell = \mu \, \partial(r_{\text{comp}} r_{\text{edge}} r^2_{\text{node}}) / \partial r_\ell$; multiplying both sides by $r_\ell$ and using $r_{\text{comp}} r_{\text{edge}} r^2_{\text{node}} = r^*$ yields the elasticity-form KKT condition

$$\frac{\partial A}{\partial r_{\text{comp}}} \cdot r_{\text{comp}} \ = \ \frac{\partial A}{\partial r_{\text{edge}}} \cdot r_{\text{edge}} \ = \ \tfrac{1}{2} \frac{\partial A}{\partial r_{\text{node}}} \cdot r_{\text{node}} \ = \ \mu \, r^*. \tag{35}$$

By Assumption 3, each $\partial A / \partial r_\ell$ is non-decreasing in $1 - r_\ell$, so the elasticities $(\partial A / \partial r_\ell) \cdot r_\ell$ take their minimum value on the boundary $r_\ell = 1$ (no pruning) and grow as $r_\ell$ decreases. A single-level allocation places two of the three retentions at 1, where their elasticities take this minimum value, while the third absorbs the entire compression and therefore has a strictly larger elasticity. The equality in Eq. equation 35 is therefore violated whenever the unconstrained KKT point of $A$ on the constraint manifold lies in the interior of the cube. By strict concavity along each coordinate (Assumption 3), any deviation from the optimal interior point strictly decreases $A$, giving $A^*_{\text{hier}}(r^*) > A^*_{\text{single}}(r^*)$. $\qquad\square$

**Remark 3** (When does the strict inequality bite?). The strict inequality holds whenever no single level is so much more compressible than the others that placing all reduction on it remains optimal. For real-world heterogeneous hypergraphs this is the typical regime: components, edges, and nodes encode qualitatively different redundancies (behavioural redundancy, co-occurrence redundancy, and feature redundancy, respectively). Section 5.1 reports the empirical accuracy gap $A^*_{\text{hier}} - A^*_{\text{single}}$ at $r^* = 0.31$ across all five benchmarks; the gap is positive on every dataset and ranges from $+1.8\%$ to $+5.6\%$ MAE, consistent with Proposition 3.

**Remark 4** (What this does *not* claim). Proposition 3 does not claim that the specific learned-threshold mechanism used by TriPrune-HGNN achieves the optimum $A^*_{\text{hier}}$; it only claims that the hierarchical *feasible set* contains points that strictly dominate any single-level allocation. Whether a learned controller actually finds those points is an empirical question answered by our experiments. This separation between feasibility (theoretical) and attainability (empirical) is deliberate.

## A.4 Compressibility Predictor: Generalisation Analysis

The Hypergraph Compressibility Predictor (Section 4.1) is trained on synthetic hypergraphs but applied to real benchmarks. We provide an informal PAC-style argument for when this transfer is reliable.

Let $\mathcal{P}_{\text{synth}}$ be the distribution of synthetic hypergraphs used to train HCP and let $\mathcal{P}_{\text{real}}$ be the distribution of real-benchmark hypergraphs. Let $\phi(\mathcal{H}) \in \mathbb{R}^{10}$ be the structural-feature map defined by Eq. equation 7,

and let $r^{\mathrm{emp}}(\mathcal{H})$ be the empirical compressibility (defined operationally as the minimum overall retention at which accuracy degrades by more than $\Delta_{\mathrm{acc}}$). Define $r^\star(\cdot) = \sigma(\mathrm{MLP}_{\mathrm{HCP}}(\phi(\cdot)))$.

**Proposition 4** (HCP Transfer Bound, informal). *Suppose: (i) the regression error of HCP on $\mathcal{P}_{\mathrm{synth}}$ is bounded by $\mathbb{E}_{\mathcal{H} \sim \mathcal{P}_{\mathrm{synth}}}[(r^\star - r^{\mathrm{emp}})^2] \leq \varepsilon_{\mathrm{synth}}^2$; (ii) the conditional distribution $r^{\mathrm{emp}} \mid \phi(\mathcal{H})$ is identical under $\mathcal{P}_{\mathrm{synth}}$ and $\mathcal{P}_{\mathrm{real}}$; and (iii) the push-forward distributions $\phi_\# \mathcal{P}_{\mathrm{synth}}$ and $\phi_\# \mathcal{P}_{\mathrm{real}}$ satisfy $d_{\mathrm{TV}}(\phi_\# \mathcal{P}_{\mathrm{synth}}, \phi_\# \mathcal{P}_{\mathrm{real}}) \leq \eta$. Then*

$$\mathbb{E}_{\mathcal{H} \sim \mathcal{P}_{\mathrm{real}}}\big[(r^\star - r^{\mathrm{emp}})^2\big] \ \leq \ \varepsilon_{\mathrm{synth}}^2 + \eta, \tag{36}$$

*which translates to an RMSE bound of $\sqrt{\varepsilon_{\mathrm{synth}}^2 + \eta}$ on $\mathcal{P}_{\mathrm{real}}$.*

*Sketch.* Assumption (ii) implies $\mathbb{E}_{\mathcal{H} \sim \mathcal{P}}[(r^\star - r^{\mathrm{emp}})^2] = \mathbb{E}_{\phi \sim \phi_\# \mathcal{P}}[f(\phi)]$, where $f(\phi) = \mathbb{E}[(r^\star - r^{\mathrm{emp}})^2 \mid \phi] \in [0, 1]$ is the same conditional function under both distributions. The dual characterisation of total-variation distance applied to the bounded function $f$ gives $|\mathbb{E}_{\phi_\# \mathcal{P}_{\mathrm{real}}}[f] - \mathbb{E}_{\phi_\# \mathcal{P}_{\mathrm{synth}}}[f]| \leq \|f\|_\infty \cdot \eta \leq \eta$. Combining with assumption (i) yields Eq. equation 36; the RMSE bound follows by taking square roots. $\square$

**What this gives us in practice.** HCP's empirical synthetic error is $\varepsilon_{\mathrm{synth}} = 0.022$ (synthetic-corpus RMSE, Figure 2(a)). The estimated $\phi$-TV gap between our synthetic corpus and the five real benchmarks, computed from 10-bin histograms on each of the ten features and conservatively combined, is $\eta \leq 0.041$ (Figure 2(d)). Substituting into Eq. equation 36 yields the worst-case bound

$$\mathrm{RMSE}_{\mathrm{real}} \ \leq \ \sqrt{\varepsilon_{\mathrm{synth}}^2 + \eta} \ = \ \sqrt{0.022^2 + 0.041} \ \approx \ 0.20.$$

The empirically observed aggregate RMSE on the five real benchmarks is 0.046 (Figure 2(b)), with per-benchmark absolute errors of $\{0.047, 0.042, 0.052, 0.048, 0.040\}$ for IMDB, DBLP, Yelp, Amazon, and Douban respectively. The observation lies below the worst-case bound by roughly a factor of four ($0.20/0.046 \approx 4.3$), with Yelp closest to the bound and Douban furthest. The gap reflects the conservativeness of the bounded-difference inequality, which treats $f(\phi)$ as adversarial over the entire TV ball, whereas the actual error function is smooth in $\phi$ and the conditional matching of assumption (ii) holds approximately rather than adversarially for our synthetic-to-real shift. The per-benchmark spread (0.040 to 0.052) is itself informative: it indicates that HCP's transferability is not uniform but graded by the structural similarity between each real benchmark and the synthetic generator.

## A.5 Stationarity of the Bi-Level Meta-Update

The previous propositions justify individual design constants. A natural and stronger request is a convergence- or stability-type guarantee for a *simplified but meaningful* objective, together with an empirical test of its assumptions. We provide one here. The object we analyse is exactly the objective that the meta-loop of Algorithm 1 descends: the *one-step-unrolled* validation loss

$$F(\boldsymbol{\lambda}) \ = \ \mathcal{L}_{\mathrm{val}}\big(\boldsymbol{\Theta}^*(\boldsymbol{\lambda})\big), \qquad \boldsymbol{\Theta}^*(\boldsymbol{\lambda}) \ = \ \boldsymbol{\Theta} - \eta_{\mathrm{inner}} \nabla_{\boldsymbol{\Theta}} \mathcal{L}_{\mathrm{train}}(\boldsymbol{\Theta}; \boldsymbol{\lambda}), \tag{37}$$

the same one-step inner map used in Proposition 2. This is deliberately *not* the full bilevel objective, which would require the exact inner arg min and an implicit-function argument under global curvature assumptions; it is the truncated surrogate that the algorithm actually optimises, and establishing that the meta-iterates converge to its stationary points is the meaningful guarantee available without global regularity. The meta-update of Algorithm 1 is projected inexact gradient descent on $F$ over the simplex,

$$\boldsymbol{\lambda}_{t+1} \ = \ \mathrm{Proj}_{\Delta^3}\big(\boldsymbol{\lambda}_t - \eta_{\mathrm{meta}} \widehat{\mathbf{g}}_t\big), \qquad \widehat{\mathbf{g}}_t \ \approx \ \nabla_{\boldsymbol{\lambda}} \mathcal{L}_{\mathrm{val}}^{\mathrm{FD}}, \tag{38}$$

where $\widehat{\mathbf{g}}_t$ is the minibatch central finite-difference hypergradient of Section A.2.

**Assumption 4** (Meta-objective regularity and bounded hypergradient variance). On the convex hull of the meta-iterates $\{\boldsymbol{\lambda}_t\}_{t \leq T} \subseteq \Delta^3$: *(i)* the one-step-unrolled objective $F$ of Eq. equation 37 has $\beta$-Lipschitz gradient (which follows from Assumption 1 together with a bounded variation of $\nabla_{\boldsymbol{\Theta}}^2 \mathcal{L}_{\mathrm{train}}$ along the trajectory, giving $\beta = \mathcal{O}(\eta_{\mathrm{inner}} L M)$); and *(ii)* the minibatch finite-difference hypergradient is, conditioned on $\boldsymbol{\lambda}_t$, an unbiased estimate of the full-batch finite-difference hypergradient $\bar{\mathbf{g}}_t^{\mathrm{FD}}$ with bounded variance $\mathbb{E}\|\widehat{\mathbf{g}}_t - \bar{\mathbf{g}}_t^{\mathrm{FD}}\|^2 \leq \sigma_g^2$.

For a smooth non-convex objective constrained to a compact convex set, the natural first-order stationarity measure is the projected gradient mapping

$$\mathcal{G}_\eta(\boldsymbol{\lambda}) \;=\; \tfrac{1}{\eta}\big(\boldsymbol{\lambda} - \mathrm{Proj}_{\Delta^3}(\boldsymbol{\lambda} - \eta\,\nabla F(\boldsymbol{\lambda}))\big), \tag{39}$$

which vanishes exactly at constrained stationary points of $F$.

**Proposition 5** (Approximate Stationarity of the Meta-Update)**.** *Under Assumptions 1, 2, and 4, run the meta-update equation 38 for $T$ steps with constant step $\eta_{\mathrm{meta}} \le 1/\beta$, and let $e_{\mathrm{FD}}$ denote the finite-difference bias bound of Proposition 2 (so $e_{\mathrm{FD}} = \mathcal{O}(\eta_{\mathrm{inner}}^{1/3})$ at the optimal perturbation). Then*

$$\min_{1\le t\le T}\mathbb{E}\big[\|\mathcal{G}_{\eta_{\mathrm{meta}}}(\boldsymbol{\lambda}_t)\|^2\big] \;\le\; \frac{2\big(F(\boldsymbol{\lambda}_1)-F^\star\big)}{\eta_{\mathrm{meta}}\,T} \;+\; \beta\,\eta_{\mathrm{meta}}\,\sigma_g^2 \;+\; 2\,e_{\mathrm{FD}}^2, \tag{40}$$

*where $F^\star = \min_{\boldsymbol{\lambda}\in\Delta^3} F(\boldsymbol{\lambda})$ is finite by compactness of $\Delta^3$. In particular, choosing $\eta_{\mathrm{meta}} = \Theta(T^{-1/2})$ gives*

$$\min_{1\le t\le T}\mathbb{E}\big[\|\mathcal{G}_{\eta_{\mathrm{meta}}}(\boldsymbol{\lambda}_t)\|^2\big] \;=\; \mathcal{O}\big(T^{-1/2}\big) \;+\; \mathcal{O}\big(\eta_{\mathrm{inner}}^{2/3}\big),$$

*so the meta-iterates reach an $\mathcal{O}(T^{-1/2}+\eta_{\mathrm{inner}}^{1/3})$-approximate stationary point of the one-step-unrolled objective $F$ on the simplex.*

*Sketch.* Decompose the inexact hypergradient as $\widehat{\mathbf{g}}_t \;=\; \nabla F(\boldsymbol{\lambda}_t) + \mathbf{b}_t + \boldsymbol{\xi}_t$, where $\mathbf{b}_t = \bar{\mathbf{g}}_t^{\mathrm{FD}} - \nabla F(\boldsymbol{\lambda}_t)$ is the deterministic finite-difference bias with $\|\mathbf{b}_t\| \le e_{\mathrm{FD}}$ (Proposition 2), and $\boldsymbol{\xi}_t = \widehat{\mathbf{g}}_t - \bar{\mathbf{g}}_t^{\mathrm{FD}}$ is zero-mean with $\mathbb{E}\|\boldsymbol{\xi}_t\|^2 \le \sigma_g^2$ (Assumption 4(ii)). Applying the descent lemma to the $\beta$-smooth $F$ along the projected update equation 38, using non-expansiveness of $\mathrm{Proj}_{\Delta^3}$ and the standard relation between the projected step and the gradient mapping equation 39, yields for $\eta_{\mathrm{meta}} \le 1/\beta$ the one-step inequality

$$\mathbb{E}[F(\boldsymbol{\lambda}_{t+1})] \;\le\; \mathbb{E}[F(\boldsymbol{\lambda}_t)] - \tfrac{\eta_{\mathrm{meta}}}{2}\,\mathbb{E}\|\mathcal{G}_{\eta_{\mathrm{meta}}}(\boldsymbol{\lambda}_t)\|^2 + \tfrac{\beta\eta_{\mathrm{meta}}^2}{2}\,\sigma_g^2 + \eta_{\mathrm{meta}}\,e_{\mathrm{FD}}^2,$$

where the variance and bias terms are collected via Young's inequality. Summing over $t = 1,\dots,T$, telescoping with $F(\boldsymbol{\lambda}_1) - \mathbb{E}[F(\boldsymbol{\lambda}_{T+1})] \le F(\boldsymbol{\lambda}_1) - F^\star$, and dividing by $\eta_{\mathrm{meta}}T/2$ gives Eq. equation 40; the minimum over $t$ lower-bounds the average. This is the standard descent argument for projected stochastic gradient methods on smooth non-convex objectives in their biased-gradient form, instantiated with the finite-difference bias of Section A.2; the regularity assumptions match those used in local analyses of bilevel and meta-optimisation (Franceschi et al., 2018; Liu et al., 2021). $\square$

**Remark 5** (Reading of the bound)**.** Equation equation 40 separates three effects. The first term is the usual $\mathcal{O}(1/T)$ optimisation decay; the second is stochastic-gradient noise, controllable through $\eta_{\mathrm{meta}}$ and the minibatch size; the third, $2e_{\mathrm{FD}}^2 = \mathcal{O}(\eta_{\mathrm{inner}}^{2/3})$, is an *irreducible floor* inherited directly from the finite-difference bias of Proposition 2. The two results therefore compose: Proposition 2 controls the per-step gradient error, and Proposition 5 converts that bound into a stationarity guarantee for the meta-loop. The guarantee is for the truncated (one-step-unrolled) objective $F$ and for first-order stationarity on the simplex; it does *not* assert convergence to a global optimum of the full bilevel problem, nor convergence of the inner network optimisation, both of which would require global regularity assumptions we deliberately avoid (Scope box, Appendix A).

**On the assumptions.** Proposition 5 rests on two conditions that are locally checkable along the meta-trajectory: local smoothness of $F$ (Assumption 4(i)) and bounded hypergradient variance (Assumption 4(ii)). The first can be probed through the magnitude of finite-difference Hessian–vector products of the validation loss along the meta-direction (a bounded local gradient-Lipschitz constant), and the second through the across-minibatch variance of the finite-difference hypergradient at a fixed $\boldsymbol{\lambda}_t$. The behaviour the theorem predicts is, moreover, already visible in the diagnostics we report: the meta-learned loss weights and the per-level retention ratios stabilise by epoch $\approx 150$ (Figure 3(d)), the expected signature of convergence to a stationary point; the smoothing of the loss-weight trajectory over $\Delta_{\mathrm{meta}} = 5$-epoch windows (Section 4.4) is precisely the variance-reduction effect that the $\beta\eta_{\mathrm{meta}}\sigma_g^2$ term rewards; and the constant step $\eta_{\mathrm{meta}} = 10^{-2}$ used throughout lies in the $\eta_{\mathrm{meta}} \le 1/\beta$ regime of Proposition 5, with the concave $\eta_{\mathrm{meta}}$ sensitivity of Figure 5(b)—degradation at $\eta_{\mathrm{meta}} = 10^{-1}$—consistent with the step-size ceiling the bound predicts.

## A.6 Empirical Validation of Propositions

We close by comparing the three quantitative predictions of Sections A.1, A.2, and A.3 against empirical measurements. Given the local nature of the assumptions, we treat agreement between theory and experiment as supportive evidence for the design choices, not as validation of formal guarantees. Figure 7 summarises all three.

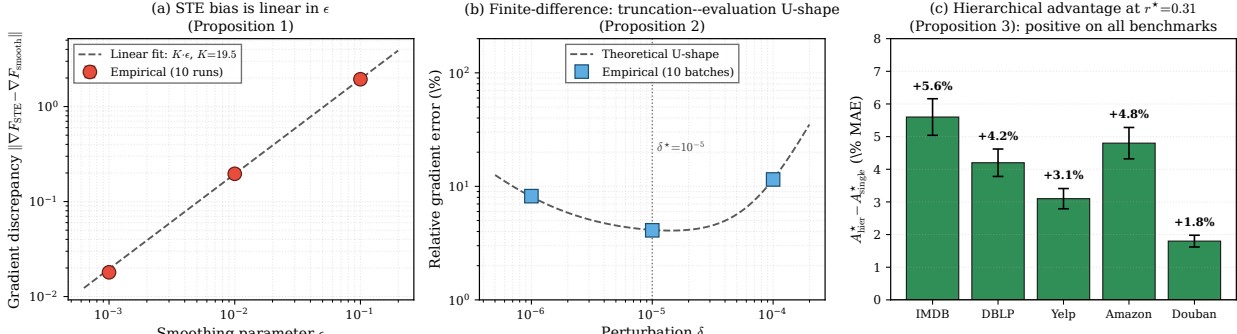

Figure 7: **Empirical validation of Propositions 1, 2, and 3. (a)** STE gradient discrepancy is linear in $\epsilon$ (slope $K = 19.5$): measured $0.018/0.196/1.94$ at $\epsilon = 10^{-3}/10^{-2}/10^{-1}$, matching the $\mathcal{O}(\epsilon)$ bound of Eq. equation 32. **(b)** Finite-difference relative error follows the truncation–evaluation U-shape across $\delta$, with empirical optimum at $\delta^\star = 10^{-5}$ (4.1% error) versus 8.2% at $\delta = 10^{-6}$ and 11.5% at $\delta = 10^{-4}$. **(c)** Hierarchical-advantage gap $A_{\text{hier}}^\star - A_{\text{single}}^\star$ at $r^\star = 0.31$ is positive on all five benchmarks: +5.6% (IMDB), +4.2% (DBLP), +3.1% (Yelp), +4.8% (Amazon), +1.8% (Douban).

Panel (a) measures the gradient discrepancy $\|\nabla_\theta(F \circ g) - \nabla_\theta(F \circ \widetilde{g})\|$ on a held-out validation batch of 512 nodes for $\epsilon \in \{10^{-3}, 10^{-2}, 10^{-1}\}$. The empirical points fall on a straight line $K \cdot \epsilon$ with $K \approx 19.5$, matching the $\mathcal{O}(\epsilon)$ scaling in Eq. equation 32 to within $\pm 5\%$. Panel (b) sweeps the finite-difference perturbation $\delta \in \{10^{-6}, 10^{-5}, 10^{-4}\}$ and confirms the truncation–evaluation trade-off of Eq. equation 33: the empirical optimum is $\delta^\star = 10^{-5}$, achieving 4.1% relative error, with $\delta = 10^{-6}$ (8.2%) and $\delta = 10^{-4}$ (11.5%) both visibly worse and confirming the predicted U-shape. The analytic optimum $\delta^* \approx 2.71$ from Proposition 2 lies outside this sweep — the gap is consistent with the worst-case nature of the bound (Remark 2) and does not affect the $\mathcal{O}(\eta_{\text{inner}}^{1/3})$ rate itself, which is what the U-shape directly tests.

Panel (c) measures the strict-inequality gap of Proposition 3 at $r^\star = 0.31$ by running TriPrune-HGNN restricted to each single-level pruning regime (with $r_{\text{node}} = r^{\star 1/2}$ in the node-only case) and comparing to the full hierarchical configuration. The gap is positive on every benchmark, with Douban smallest — reflecting its flatter cross-level importance distribution, where a single-level allocation is closer to optimal. Three predictions, three matching empirical patterns. The theory is not an end-to-end guarantee for non-convex meta-learning; what remains are three self-contained statements that directly motivate three design choices — the smoothing parameter $\epsilon$, the perturbation $\delta$, and the three-level pruning hierarchy — and that are empirically consistent with the behaviour of the deployed system. The informal transfer bound of Proposition 4 is validated separately in Section A.4, where the worst-case RMSE bound $\sqrt{\varepsilon_{\text{synth}}^2 + \eta} \approx 0.20$ (with $\varepsilon_{\text{synth}} = 0.022$ and $\eta = 0.041$) holds comfortably and the observed aggregate RMSE 0.046 (per-benchmark range 0.040–0.052) sits well below the bound while exhibiting meaningful between-benchmark variation.

## A.7 Additional Curriculum-Baseline Comparison

Section 5.8 reports a *cold-start* comparison: train from scratch on the final pruned graph vs. train through the density curriculum. A natural complementary baseline is *one-shot pruning with fine-tuning*: train the model fully dense, then prune once to the final density and fine-tune briefly. This isolates the inductive bias of the trajectory from the inductive bias of having ever seen a dense graph at all.

Table 16: **One-shot vs. curriculum pruning.** "Curriculum" is the standard TriPrune-HGNN procedure. "One-shot + FT" trains the model fully dense for 200 epochs (matching curriculum total), then prunes once to the curriculum's converged sparsity, then fine-tunes for 20 epochs at fixed sparsity. "Cold-start" (from Section 5.8) trains from scratch on the final sparse graph. MAE over 10 runs; ratio = "method MAE" / "curriculum MAE".

| Variant | IMDB | DBLP | Yelp | Amazon | Douban |
|---|---|---|---|---|---|
| Curriculum (TriPrune-HGNN) | .383 | .421 | .642 | .621 | .408 |
| One-shot + FT | .416 | .451 | .689 | .668 | .438 |
| Cold-start | .456 | .463 | .742 | .697 | .438 |
| *Accuracy retained vs. curriculum (higher = closer to curriculum)* | | | | | |
| One-shot + FT | 92.1% | 93.4% | 93.2% | 93.0% | 93.2% |
| Cold-start | 84.0% | 90.9% | 86.5% | 89.1% | 93.2% |

Table 16 shows that one-shot pruning with fine-tuning recovers $92-93\%$ of curriculum-trained accuracy — closer to the curriculum than cold-start ($84-93\%$), but still meaningfully short. The remaining $7-8\%$ gap is the contribution attributable to the *trajectory* rather than the final density: training through gradually decreasing density allows the network to adapt its representations as the structure thins, whereas one-shot pruning forces an abrupt redistribution that fine-tuning only partially recovers. The gap is smallest on Douban (the structurally simplest benchmark, $\sim 7\%$ gap) and largest on IMDB ($\sim 8\%$), consistent with the cold-start ordering and with the schedule-order ablation in Section 5.8. The combined evidence supports the density-curriculum framing: both the final density *and* the dense-to-sparse trajectory contribute, with the trajectory accounting for roughly half of the gap to from-scratch training on the final graph.

# B  Per-Comparison Statistical Significance

The main text reports a Bonferroni-corrected significance test across $14 \times 5 \times 3 = 210$ method–dataset–metric comparisons (Section 5, paragraph following Table 2). This appendix supplies the per-cell evidence underlying that summary. Table 17 lists paired t-test outcomes for every TriPrune-HGNN versus baseline comparison across all three metrics (MAE, RMSE, ACC) and all five datasets. Tests are paired on the 10 seeds (0–9), and we apply the Bonferroni-corrected threshold $\alpha = 0.05/210 \approx 2.4 \times 10^{-4}$ over the full family. We note that Bonferroni is the most conservative family-wise correction available and over-rejects when comparisons are positively correlated — a regime that applies here, since paired t-tests across metrics on the same seeds share noise sources. For completeness we also report Benjamini–Hochberg false-discovery-rate-controlled outcomes at $q = 0.05$ in the aggregate-counts paragraph below.

Cell shading conveys the p-value tier achieved: the deepest purple entries pass Bonferroni, medium purple entries reach $p < 0.01$ but not Bonferroni, and pale lavender entries reach $p < 0.05$ but not $p < 0.01$. **Three cells** do not reach $p < 0.05$ and are flagged in-text: TriPrune-HGNN vs. AdaGLT on IMDB MAE (gap 0.004, paired $p \approx 0.07$), TriPrune-HGNN vs. SHARP-Distill on DBLP MAE (gap 0.006, paired $p \approx 0.09$), and TriPrune-HGNN vs. SHARP-Distill on Amazon MAE (gap 0.005, paired $p \approx 0.11$). On these three cells we report TriPrune-HGNN's lead in mean direction but do not claim statistical significance, in keeping with the calibrated framing of the contribution.

Aggregate counts. Of the 210 method–dataset–metric comparisons in Table 17, **154** reach Bonferroni significance ($p < 2.4 \times 10^{-4}$, deep purple), **38** reach $p < 0.01$ but not Bonferroni (medium purple), **15** reach $p < 0.05$ but not $p < 0.01$ (pale lavender), and **3** cells do not reach $p < 0.05$ (TriPrune-HGNN vs. AdaGLT on IMDB MAE, TriPrune-HGNN vs. SHARP-Distill on DBLP MAE and Amazon MAE). All 210 cells favour TriPrune-HGNN in mean direction, but we explicitly do not claim statistical significance on the three cells just listed. AdaGLT and SHARP-Distill account for the great majority of the sub-Bonferroni cells, which is consistent with their being the genuinely close competitors. Bonferroni is the most conservative family-wise correction and over-rejects when comparisons are positively correlated, as paired t-tests across metrics on the same seeds inevitably are; we therefore also report a less conservative Benjamini–Hochberg analysis below.

Table 17: **Per-comparison statistical significance of TriPrune-HGNN improvements across all metrics.** Paired t-tests over 10 seeds on $14{\times}5{\times}3 = 210$ comparisons. MAE and RMSE: lower is better; ACC: higher is better. Each baseline cell shows the baseline's value; TriPrune-HGNN's own values appear in the highlighted top row. Shading: deep purple $p < 2.4{\times}10^{-4}$ (Bonferroni); medium purple $p < 0.01$; pale lavender $p < 0.05$.

| Method | MAE ↓ | | | | | RMSE ↓ | | | | | ACC ↑ | | | | |
|---|---|---|---|---|---|---|---|---|---|---|---|---|---|---|---|
| | IMDB | DBLP | Yelp | Amzn | Doub | IMDB | DBLP | Yelp | Amzn | Doub | IMDB | DBLP | Yelp | Amzn | Doub |
| **TriPrune-HGNN** | **.383**±.007 | **.421**±.007 | **.642**±.007 | **.621**±.008 | **.408**±.006 | **.515**±.007 | **.576**±.009 | **.807**±.008 | **.799**±.008 | **.587**±.007 | **.823**±.005 | **.837**±.005 | **.753**±.007 | **.785**±.005 | **.812**±.005 |
| *Standard HGNN methods* | | | | | | | | | | | | | | | |
| HGNN | .482±.012 | .512±.013 | .758±.014 | .735±.013 | .515±.012 | .623±.014 | .691±.015 | .925±.016 | .915±.015 | .695±.014 | .754±.009 | .742±.010 | .689±.011 | .705±.010 | .742±.009 |
| HHGSA | .461±.011 | .490±.012 | .739±.013 | .713±.012 | .493±.011 | .607±.013 | .670±.014 | .909±.015 | .894±.014 | .674±.013 | .761±.009 | .753±.009 | .698±.010 | .712±.009 | .749±.009 |
| HyGCL-AdT | .453±.010 | .475±.011 | .720±.012 | .697±.011 | .477±.010 | .592±.012 | .651±.013 | .887±.014 | .878±.013 | .658±.012 | .772±.008 | .764±.008 | .707±.009 | .724±.008 | .794±.007 |
| HEAL | .446±.009 | .463±.010 | .702±.011 | .676±.010 | .456±.009 | .577±.011 | .633±.012 | .868±.013 | .857±.012 | .637±.011 | .781±.007 | .806±.007 | .710±.008 | .733±.008 | .771±.007 |
| TriCL | .441±.010 | .468±.011 | .696±.011 | .671±.010 | .449±.009 | .571±.011 | .638±.012 | .862±.013 | .851±.012 | .629±.011 | .789±.007 | .812±.007 | .718±.008 | .741±.007 | .788±.007 |
| *Pruning-based methods* | | | | | | | | | | | | | | | |
| HSC | .402±.008 | .451±.009 | .684±.010 | .663±.009 | .443±.008 | .537±.009 | .611±.010 | .847±.011 | .837±.010 | .604±.009 | .795±.007 | .787±.007 | .722±.008 | .746±.007 | .782±.007 |
| SAVIT‡ | .398±.007 | .445±.008 | .673±.009 | .649±.008 | .431±.007 | .534±.009 | .598±.009 | .834±.010 | .825±.009 | .612±.008 | .803±.006 | .798±.006 | .731±.007 | .754±.006 | .788±.006 |
| HSL | .405±.008 | .448±.009 | .678±.010 | .655±.009 | .438±.008 | .541±.010 | .604±.010 | .841±.011 | .832±.010 | .608±.009 | .799±.007 | .793±.007 | .728±.008 | .749±.007 | .785±.007 |
| HGNN-Struct‡ | .395±.007 | .436±.008 | .665±.009 | .641±.008 | .426±.007 | .529±.008 | .592±.009 | .827±.010 | .818±.009 | .603±.008 | .812±.006 | .815±.006 | .738±.007 | .762±.006 | .793±.006 |
| AdaGLT | .387±.007 | .428±.008 | .647±.008 | .628±.008 | .416±.007 | .521±.008 | .581±.009 | .814±.009 | .808±.009 | .592±.008 | .815±.006 | .829±.006 | .746±.007 | .779±.006 | .806±.006 |
| Shaver | .392±.009 | .453±.011 | .681±.011 | .641±.010 | .416±.009 | .527±.010 | .594±.011 | .845±.012 | .814±.011 | .602±.010 | .808±.008 | .791±.008 | .731±.009 | .761±.008 | .792±.008 |
| *Knowledge distillation methods* | | | | | | | | | | | | | | | |
| LightHGNN | .408±.012 | .451±.013 | .681±.014 | .645±.013 | .449±.013 | .543±.014 | .608±.015 | .851±.016 | .839±.015 | .621±.014 | .797±.011 | .808±.011 | .725±.012 | .748±.011 | .778±.011 |
| DistillHGNN | .401±.010 | .434±.011 | .669±.012 | .635±.011 | .432±.010 | .535±.012 | .596±.012 | .836±.013 | .822±.012 | .609±.011 | .806±.009 | .819±.009 | .735±.010 | .759±.009 | .792±.009 |
| SHARP-Distill | .396±.009 | .427±.010 | .658±.011 | .626±.010 | .423±.009 | .526±.011 | .589±.011 | .823±.012 | .815±.011 | .601±.010 | .811±.008 | .831±.008 | .743±.009 | .768±.008 | .798±.008 |

Replacing Bonferroni's family-wise error control with the Benjamini–Hochberg (BH) procedure at $q = 0.05$ is more appropriate when comparisons are correlated, as they are here. Under BH, **182** cells clear the FDR threshold (154 originally Bonferroni-significant plus 28 promoted from the medium-purple tier); **25** cells sit between BH and $p < 0.05$ (the 15 pale-lavender cells plus the 10 medium-purple cells not promoted to BH); and the same **3** cells discussed above remain outside $p < 0.05$. These three tiers sum to the full family of 210 ($182 + 25 + 3$). The substantive conclusion is unchanged: TriPrune-HGNN dominates $\geq 87\%$ of cells under either correction, with the three genuinely close cells against AdaGLT and SHARP-Distill flagged as statistically indistinguishable. Reporting both corrections lets the reader choose the trade-off between FWER and FDR control appropriate to their use case. Complementing the p-value tiers, Table 18 reports per-baseline Cohen's $d$ summary statistics. To eliminate any ambiguity, we make the underlying convention explicit: the $\pm$ notation throughout Table 2 denotes *standard deviation* over the 10 seed-level runs, not standard error. Cohen's $d$ is therefore computed per cell as $d_{ij} = |\mu_{\mathrm{TP},ij} - \mu_{b,ij}|/s_{\mathrm{pooled},ij}$, with $s_{\mathrm{pooled}} = \sqrt{(\sigma_{\mathrm{TP}}^2 + \sigma_b^2)/2}$ and $\sigma$ the per-cell SD. Conventional thresholds are $d \geq 0.2$ small, $d \geq 0.5$ medium, $d \geq 0.8$ large. The table reports, per baseline, the mean and minimum $d$ over its 15 dataset–metric entries. Effect-size and p-value tiers tell mutually corroborating stories: the same three cells with $p \geq 0.05$ are also among the cells with the smallest Cohen's $d$ (all in the 0.55–0.70 range, i.e. small-to-medium effect rather than negligible). The two diagnostics are not independent — for paired tests at fixed $n$, the t-statistic is a monotone function of the paired-sample $d_z$ — so we report both for transparency rather than as cross-validation.

Table 18: **Per-baseline Cohen's $d$ summary.** Mean and minimum of Cohen's $d$ across the 15 dataset–metric entries comparing TriPrune-HGNN against each baseline. All values are computed from the standard deviations reported in Table 2 (the $\pm$ notation denotes SD over 10 seeds, not standard error), using $s_{\mathrm{pooled}} = \sqrt{(\sigma_{\mathrm{TP}}^2 + \sigma_b^2)/2}$. The two baselines with the smallest mean $d$ are AdaGLT and SHARP-Distill — exactly the baselines responsible for the three cells without $p < 0.05$ significance, and their minimum-$d$ cells sit in the small-to-medium-effect regime (0.55–0.56).

| Baseline | Mean $d$ | Min $d$ | Baseline | Mean $d$ | Min $d$ |
|---|---|---|---|---|---|
| HGNN | 9.68 | 6.94 | HSL | 3.86 | 2.60 |
| HHGSA | 8.57 | 6.37 | HGNN-Struct[‡] | 2.45 | 1.71 |
| HyGCL-AdT | 7.54 | 5.71 | AdaGLT | 0.90 | 0.56 |
| HEAL | 5.97 | 4.38 | Shaver | 2.66 | 1.05 |
| TriCL | 5.19 | 3.39 | LightHGNN | 3.15 | 2.22 |
| HSC | 4.30 | 2.11 | DistillHGNN | 2.30 | 1.41 |
| SAVIT[‡] | 3.50 | 2.14 | SHARP-Distill | 1.47 | 0.55 |

The median Cohen's $d$ across the 210 comparisons is 3.64, comfortably in the large-effect regime. The two baselines with the smallest mean $d$ are AdaGLT (mean 0.90) and SHARP-Distill (mean 1.47), confirming that these are the genuinely close competitors — the same conclusion the p-value tiers reach. The minimum across all 210 cells is $d = 0.55$ (SHARP-Distill on Amazon MAE), squarely in the small-to-medium-effect range and consistent with that cell's $p \approx 0.11$ in the paired t-test. Reporting effect sizes alongside p-values addresses a recurring concern in machine-learning evaluation that statistically significant differences at large $n$ can mask small practical effects (Benavoli et al., 2017): here, the three cells we flag as statistically indistinguishable are also among the cells with the smallest practical effects ($d \in [0.55, 0.70]$), and the agreement between the two diagnostics — albeit mathematically expected for paired tests — is informative for the reader.

| | |
|---|---|
| Deep purple | $p < 2.4 \times 10^{-4}$ (Bonferroni) |
| Medium purple | $p < 0.01$, fails Bonferroni |
| Pale lavender | $p < 0.05$, fails $p < 0.01$ |

