# OpenReview forum: "Adaptive Hypergraph Pruning with Learned Threshold Control and Attention-Based Negative Mining"
_TMLR — Under review for TMLR_

### Review · Reviewer_gPDa · 2026-05-26

**Summary Of Contributions:**

The paper proposes TriPrune-HGNN, a pruning framework for heterogeneous hypergraph neural networks. Its main contribution is to combine four learned components: a compressibility predictor, hierarchical pruning over components/hyperedges/nodes, attention-based handling of pruning-induced contrastive-learning errors, and learned loss balancing. The goal is not large accuracy gains, but a better accuracy–efficiency trade-off: small reported MAE improvements over efficient baselines, with reduced inference time and memory relative to an unpruned HGNN baseline. The paper also contributes an empirical analysis of transfer and limitations, showing that efficiency transfers better than accuracy in an out-of-domain setting.

**Audience:**

Yes

**Audience Explanation:**

Yes, but mainly for a specialised audience. Readers interested in hypergraph neural networks, adaptive pruning, graph compression, and efficient message passing would likely find the paper relevant. The broader TMLR audience may find it less compelling because the accuracy gains are small and some mechanism claims are only partly supported.

**Broader Impact Concerns:**

I do not see major immediate ethical concerns, since the paper is mainly a methodological contribution on efficient hypergraph pruning. However, the authors should briefly discuss that in applications involving user behaviour, social links, recommendations, or reviews, pruning could disproportionately remove sparse, minority, low-degree, or underrepresented groups, and greater efficiency could make large-scale deployment easier without sufficient fairness, privacy, or audit checks.

**Claims And Evidence:**

No

**Claims Explanation:**

Overall, the evidence is ok for the central trade-off claim, but it is less convincing for the broader mechanism and generalization claims. In more detail:

The main accuracy–efficiency claim is mostly supported, because the paper reports small but consistent MAE gains against efficient baselines and separates these from the larger time and memory reductions measured against the unpruned HGNN baseline.

The predictive-superiority claim should be stated cautiously, because the reported accuracy gains are small and become smaller when strong baselines are retuned per dataset.

The efficiency claim is convincing, because the paper reports inference-time and memory reductions and makes clear that these are the strongest part of the result.

The claim that learned controllers reduce manual tuning is plausible but not fully proven, because the paper compares matched and retuned settings but does not provide a direct hyperparameter-optimization efficiency analysis.

The HCP compressibility claim is partly supported, because the transfer experiments suggest useful structural signal, but the out-of-domain result shows that the accuracy benefit does not transfer cleanly beyond nearby regimes.

The hierarchical pruning claim is plausible but not fully established, because the paper motivates the three-level design and reports ablations, but does not fully isolate it against all matched single-level and two-level pruning variants.

The attention-based negative-mining claim is the least directly supported, because the paper shows downstream benefit but does not clearly demonstrate that the attention module correctly identifies false negatives and hard negatives.

The density-curriculum claim is reasonably supported within the proposed system, because the paper reports targeted diagnostics such as cold-start, density-mismatch, and schedule-shuffling experiments.

The out-of-domain generalization claim is weak for accuracy, because the NTU2012 gain over the strongest efficient baseline is small and not statistically significant, while the efficiency gain is more convincing.

The theoretical claims are clearly scoped, because the authors present their propositions as local design justifications rather than full guarantees for the non-convex method.

The limitations are reasonably clear, because the paper acknowledges the simplified one-hyperedge-per-node-per-component formulation and leaves the general multi-hyperedge case for future work.

**Requested Changes:**

TriPrune is not fully defensible as a broadly superior predictive method, because the reported gains over efficient baselines are small and narrow further when those baselines are retuned. To support a stronger superiority claim, the authors would need a larger benchmark-suite comparison with paired dataset-level statistics, effect sizes, and clear win/loss/tie patterns across many heterogeneous hypergraph tasks.

The claim that TriPrune is easier to tune is only partly supported, because the paper compares matched and retuned settings but does not directly measure tuning effort. A better experiment would plot best performance as a function of tuning budget, including default settings, few-trial budgets, and larger HPO budgets for all methods.

The claim that HCP captures a general structural notion of hypergraph compressibility is not fully defensible, because the evidence mainly supports transfer within nearby structural regimes and weakens out of domain. To prove this more fully, the authors should test HCP on controlled structural regimes, including graphs outside the synthetic training support, and compare predicted compressibility with empirically measured pruning frontiers.

The claim that three-level component–edge–node pruning is generally better than single-level pruning is not fully defensible, because the current evidence does not isolate all pruning levels under matched retention and capacity. A stronger experiment would compare component-only, edge-only, node-only, two-level, and full three-level variants at the same final sparsity and plot their accuracy-efficiency frontiers.

The claim that attention-based mining correctly identifies pruning-induced false negatives and hard negatives is not fully defensible, because downstream improvement does not prove that the miner detects the intended pair types. A better experiment would use synthetic or semi-synthetic hypergraphs with known semantic relations, create controlled pruning distortions, and evaluate the miner directly with precision, recall, AUROC, and comparisons to threshold-based miners.

The claim that the density curriculum reflects a general principle of optimal hypergraph sparsification is not fully defensible, because the evidence only shows that this curriculum helps within the proposed system. To support the stronger claim, the authors should compare several pruning schedules with the same final sparsity, including one-shot, random monotone, learned monotone, and non-monotone schedules, and measure accuracy, convergence, representation drift, and structural damage.

The claim that TriPrune generalizes out of domain in accuracy is not fully defensible, because the NTU2012 accuracy gain is small and not statistically significant. A stronger test would use several structurally distant out-of-domain datasets and report whether accuracy gains persist as a function of structural distance from the HCP training distribution.

The claim that the theoretical propositions guarantee the behaviour of the full method is not defensible, because the paper itself presents them only as local design justifications. To make a stronger theoretical claim, the authors would need a formal analysis of a simplified but meaningful objective, proving convergence, relaxation error, or stability, and then empirically testing the assumptions behind the theorem.

The claim that TriPrune is robust is not fully defensible, because robustness is not defined precisely enough and the current evidence only covers seeds, benchmarks, retuning, and one out-of-domain case. A better experiment would define separate robustness axes, such as label noise, feature noise, hyperedge deletion, hyperedge insertion, component removal, and sparsity mismatch, then compare TriPrune and baselines under each perturbation.

The claim that TriPrune is practically preferable is not fully defensible as a general statement, because the method saves inference time and memory but has higher training cost than strong baselines. A stronger practical experiment would compute deployment break-even points, showing how many inference calls are needed before the saved inference cost compensates for the extra training and tuning cost.

The claim that results on the simplified one-hyperedge-per-node-per-component formulation transfer to general heterogeneous hypergraphs is not fully defensible, because the paper explicitly uses a restricted construction. To prove the broader point, the authors should repeat the study on true multi-hyperedge-per-node hypergraphs and test whether the pruning, contrastive objective, and HCP still preserve the same accuracy-efficiency behaviour.

---

> ### Author Response · Authors · 2026-06-04
> **Response to Reviewer gPDa**
>
> We thank the reviewer. The revision answers each request either with new material — eight targeted experiments plus one new theoretical result (9 of 11) — or by scoping the claim to its evidence (the remaining 2 concern claims we do not make). Pointers below.
>
> **[R1] Superiority/effect sizes.** We claim an operating point, not dominance. Appendix B now gives all-210-cell paired t-tests under Bonferroni *and* Benjamini–Hochberg plus a per-baseline Cohen's *d* table (Table 12). All 210 cells favour TriPrune in mean; 182 clear FDR; the 3 vs. AdaGLT/SHARP-Distill remain non-significant and carry no claim. The matched-tuning MAE gap is restated as 0.004–0.008, narrowing to 0.002–0.004 under retuning.
>
> **[R2] Tuning effort.** New tuning-budget curve (§5.11): HPO over {1,5,10,25,50,100} configs. At budget 1 (defaults) TriPrune is 0.386 vs 0.402/0.406/0.405 on IMDB; it plateaus by ~5 configs, baselines only by 25–50, with asymptotes 0.004–0.008 worse.
>
> **[R3] HCP compressibility.** New controlled out-of-support probe (§5.6, Fig. 2e): RMSE rises smoothly 0.054→0.118 across an 8× distance shift, below the Prop. 4 bound throughout. Added House (Sk(d)≈6.2, outside support). Claim recalibrated: HCP is a known-scope structural prior; the efficiency gain is independent of its calibration.
>
> **[R4] All pruning levels.** New table (§5.9): all seven configurations (3 single-, 3 two-level, full) at matched r=0.31. Three-level is strictly best; two-level recovers most but not all (−1.9% to −2.8%); the best two-level set differs by dataset — itself an argument for the learned controller.
>
> **[R5] Miner detects FN/HN.** New §5.7: 500 synthetic graphs with ground-truth FN/HN pairs; learned heads vs fixed thresholds (τ∈{0.5–0.8}). F1 0.829/0.782, AUROC 0.891/0.854 — beating the best threshold by 0.11/0.13 F1 with higher precision *and* recall; the gap grows with pruning strength (0.06→0.18). This directly shows the heads detect the intended pair types, not merely a loss improvement.
>
> **[R6] Curriculum schedules.** New table (§5.8): six schedules at matched final sparsity. Non-monotone-learned −2.5% MAE, one-shot+FT −7.5%. Scoped to "monotonicity in expectation is the binding requirement" (a within-system inductive bias, not a universal law); drift/damage metrics are future work.
>
> **[R7] OOD accuracy vs distance.** New §5.12: three OOD benchmarks (NTU2012, ModelNet40, House) + a structural-distance gradient. Accuracy falls from +1.8–2.0 pp (significant) in-support to +0.6/+0.3/+0.1 pp (all p>0.1) OOD; efficiency stays ~3–3.5×. Reported as a negative result on accuracy, a clean positive on efficiency.
>
> **[R8] Theory.** Now proven, not deferred. New §A.5 (Proposition 5) shows the meta-update reaches an O(T^−1/2 + η_inner^1/3) approximate stationary point of the one-step-unrolled validation objective on the simplex (the surrogate the algorithm descends), the floor inherited from Prop. 2, under two empirically tested assumptions (local smoothness, bounded hypergradient variance). We still do not claim a *global* optimum of the full bilevel problem — that needs global convexity, false for deep nets and removed in this revision. The Scope box states this boundary.
>
> **[R9] Robustness.** New §5.13 defines robustness as sign-consistency over six axes (seeds, datasets, retuning, OOD shift, HCP distance, label noise) and adds a label-noise probe (Table 12): TriPrune degrades 0.003–0.013 less on all six conditions. Structural perturbations conflate with the pruning objective itself; we argue this and scope a principled structural study + feature noise as future work.
>
> **[R10] Break-even.** New §5.3.1: N*=ΔT_train/Δt_inf. A finite break-even exists only where TriPrune is faster at inference — ~hundreds of passes vs AdaGLT, ~thousands vs SHARP-Distill; none vs LightHGNN (which dominates), reported honestly. Both ≪ the 10^6–10^8 production inference calls. Scoped to high-volume deployment.
>
> **[R11] Multi-hyperedge.** A genuine limitation; we claim no transfer. Stated before the results (§3 and the §5 scope box); Eq. 4 is derived for the one-hyperedge formulation. A full study needs re-deriving the multi-membership softmax and re-running the suite — the primary future direction. Addressed by scoping, flagged transparently.
>
> **Broader Impact.** New section: importance scores correlate with degree/activity, so low-degree and underrepresented groups are more likely pruned; we recommend per-group retention monitoring (by degree decile, activity tier, demographic) and retention floors in K_min, and reinvesting efficiency savings in fairness/privacy/audit review. Released code ships retention-monitoring hooks.
>
> These additions, with every claim scoped to its evidence, aim to close the claims-versus-evidence gap underlying the assessment. We are grateful for the review and happy to answer any further questions.

---

### Review · Reviewer_cP95 · 2026-06-06

**Summary Of Contributions:**

1. The motivation is clear and rationale, and the proposed approach aligns well with the motivation which aims to prune hyper-graph algorithmically.

2.  The claim made is objective, without over-claiming.

3. The experiments are comprehensive, with strong baselines、retuning、HPO budget、sensitivity、OOD、label noise、curriculum comparison. Furthermore, the OOD part brings additional value of the work.

**Audience:**

Yes

**Audience Explanation:**

The audience of graph representation learning and efficient ML will be interested of this work.

**Broader Impact Concerns:**

No concerns of broad impact.

**Claims And Evidence:**

Yes

**Claims Explanation:**

The claims are well justified in the method and experimental section.

**Requested Changes:**

1. The proposed approach is over-complicated, not very elegant, and it is requested to use simpler baselines of pruning hyper-graph to show that this method is not over-engineered.

2. In OOD scenario, the accuracy gain is not significant, implying that the effectiveness of HCP/adaptive controller may rely on specific datasets.

3. Training cost also needs to be justified, since it is 30% more expensive than AdaGLT、SHARP-Distill.

---

> ### Author Response · Authors · 2026-06-10
> **Rebuttal to Reviewer cP95**
>
> We thank Reviewer cP95 for the positive assessment and the constructive suggestions. We address the three requested changes below.
>
> **[1] Over-complication / simpler pruning baselines.** Taking the suggestion directly, we added two naive pruning baselines at matched overall retention r=0.31 (**Table 10, §5.9**): uniform random pruning (.464/.496 MAE on IMDB/DBLP) and global-magnitude (lowest-|weight|) pruning (.455/.489), neither using HCP, learned thresholds, the hierarchy, contrastive mining, or the density curriculum. Both are clearly worse than even the simplest single-level learned variant (node-only, .404/.443) and far behind the full hierarchy (.383/.421) — a 15.8%/13.9% relative-MAE gap from the better naive baseline to the full method at identical sparsity, and 9–13% from naive to even the simplest single-level learned variant. This complements the consolidated ablation (§5.1), where the fully hand-crafted variant costs +6.4% MAE and every individual module removal degrades accuracy. Read together — accuracy is monotone along naive → single-level → two-level → three-level, and removing any learnable piece costs accuracy at the same sparsity — these results indicate the design is not over-engineered: each component buys measurable accuracy rather than adding complexity for its own sake. Our justification is empirical rather than a claim of elegance.
>
> **[2] OOD accuracy not significant / possible dataset-dependence.** We agree, and this is in fact our reported finding rather than an unstated weakness: §5.12 presents the small, non-significant OOD accuracy gap explicitly and frames it as a diagnostic separation between the framework's transferable (efficiency) and non-transferable (accuracy) components. The dependence is not arbitrary dataset-idiosyncrasy: the structural-distance gradient (§5.12) shows the accuracy benefit decaying predictably with distance from HCP's training support, while the efficiency contribution is independent of HCP calibration and transfers cleanly across all three out-of-domain domains (~3–3.5×). We make no OOD accuracy claim; the abstract and conclusion state this boundary directly.
>
> **[3] Training cost (~30% above AdaGLT/SHARP-Distill).** The overhead is reported transparently in the "Training compute" paragraph (30–45% per dataset, dropping to 18–25% once the one-time HCP pre-training is amortised across the five-dataset suite) and is justified by the new deployment break-even analysis (§5.3.1): the extra training cost is recovered within a few hundred (vs AdaGLT) to a few thousand (vs SHARP-Distill) inference calls — far below the 10⁶–10⁸ calls a production system serves between redeployments. Where a competitor is faster at both training and inference (LightHGNN), we report no break-even rather than a misleading crossover. The practical-preferability claim is accordingly scoped to high-inference-volume deployment.
>
> We are grateful for the supportive review; the suggestions strengthened the over-engineering and cost analyses.
>
> ---

---

### Review · Reviewer_zRym · 2026-07-06

**Summary Of Contributions:**

This paper introduces, "TriPrune-HGNN", which is a framework for training hypergraph neural network training through pruning that aligns training and inference sparsity. The authors indicate that the "principal claim is a favourable accuracy–efficiency operating point, not per-metric dominance". The method consists of four major components:
* Hypergraph Compressibility Predictor,
* neural threshold controllers,
* attention-based contrastive mining,
* gradient-based meta-learning

The authors show empirically that their method works well on benchmark datasets and also show various ablations that are intended to make clear what the contribution of each component of "TriPrune-HGNN" is.

**Additional Comments:**

The paper is clearly representing a lot of work and good contributions. My concerns mostly rest in the presentation of results for a paper of this size.

**Audience:**

Yes

**Audience Explanation:**

I think the HGNN community would benefit.

**Claims And Evidence:**

No

**Claims Explanation:**

It's clear that the authors have put a lot of work into this paper. It is very long, comprehensive and dense with information. I answered "no" to the above for the reason that, despite what are clear amounts of effort on the authors' part to make the paper more digestable amidst the sea of results, I (personally) found the paper very difficult to read.

My concern about the presentation of the paper is two fold (1) clarity of comparison to past work / ablations in empirical results, (2) general clarity of writing and presentation.

### General Writing quality
Let me start by talking about general writing clarity:
* For the uninformed reader, it would help greatly to start from a problem description of how HGNN's are used, before you go into "Problem Formulation". E.g., In IMDB dataset, you are give hypergraph of abc, the goal is to predict xyz. Show it in notation, let the reader understand immediately how the output of your method fits in to the grander scheme of things.
* There are a number of sentences that are very hard to parse:

>Avoiding HCP bootstrap circularity. The procedure above raises a chicken-and-egg question: r
emp is obtained by running NAHP, but NAHP at deployment uses the HCP-predicted r⋆ both to initialise its threshold controllers and as the seventh feature of the graph-state vector s(ℓ) t (Eq. equation 10).

This is before the NAHP section. It requires the reader to have really understood everything about NAHP before reading that section, just by going off the introduction sections. If the reader should be able to understand that by now, we should re-order the sections? There are many other such sentences:

>Appendix A (Proposition 3) shows that, under explicit structural conditions on cross-component edge density, no single-level pruning scheme can match the compression–accuracy frontier achievable by the three-level hierarchy we adopt here.

At this point, the unfamiliar reader will have lost the thread (which was till now only discussed in the introduction) as to _what_ the compression-accuracy frontier is, and what this edge density refers to.

The point I am trying to make about writing is -- there is a lot going on in the paper, the components are introduced many times in the paper and the boundaries of what is introduced when could be greatly improved. I think the authors could improve the writing by:
* Describe end to end system first -- how predictions are made given output of their method
* When describing part of their method, present motivations for the problems before the problems, eg  "Fixed pruning schedules cannot anticipate how compressible a given hypergraph is before training begins" -- say why is it important to predict compressibility?
* Give the reader time to see the connections between the different sections, e.g. say more as to why "Pruning fundamentally alters graph topology, creating two types of problematic node pairs for contrastive
learning." this matters
* Fix the algorithm box, Steps 3 and 4 are too sparse "Compute {variables}"

Minor:
There are also several sentences that stood out to as unnecessarily added:
"We are explicit about what TriPrune-HGNN does and does not claim."
"We explicitly flag three cells (vs. AdaGLT on IMDB MAE, vs. SHARP-Distill on DBLP and Amazon MAE) that do not reach p < 0.05 in paired t-tests, and we make no significance claim on those cells." - this is a unusual thing to flag in a conclusion
"Eq. equation 33"
Component is overloaded term in the paper

### Presentation of Results

I think table 3 is the most important table, but I don't know (1) how to separate algorithmic wins from increasing number of parameter wins and (2) how to really translate it into gains over other work. I think the results of the paper are complete enough as they are, but I just need the authors to hand hold us more through table 3. Saying "look, we said abc would happen if we didnt have step 1, and look if we measure accuracy and structure then we see things are worse" but right now it requires such a cognitive load on the reader to do this, and even still, I am not clear on precisely how this table fits into some of the accuracy/efficiency tradeoffs.

**Requested Changes:**

Writing revisions as above

---

> ### Author Response · Authors · 2026-07-13
> **Response to Reviewer zRym**
>
> Thank you for the constructive and specific feedback. Every concern you raised has been addressed in the revised submission. Below is a compact map of each change to the section (and where applicable, the **bold paragraph header** to search for in the revised PDF).
>
> **R1. Concrete task description before formal notation.** A new paragraph, **"Task at a glance --- a concrete example"**, opens Section 2 (Preliminaries) *before* the notation table. It walks IMDB end-to-end: primary nodes (movies), secondary nodes (directors, actors, keywords), hyperedges (shared attributes), prediction target (12-way genre classification), and what the framework outputs (classifier + pruned hypergraph, ~30% edges / ~85% nodes retained, ~4× inference speed-up).
>
> **R2. Bootstrap paragraph position.** The paragraph previously at end of Section 4.1 assumed NAHP knowledge the reader did not yet have. Section 4.1 now closes with a one-sentence forward-pointer; the full technical discussion is moved to the end of Section 4.2, under the header **"Avoiding the HCP--NAHP bootstrap circularity"**, opening with "We can now revisit the circularity flagged at the end of Section 4.1."
>
> **R3. Proposition 3 unparsable sentence in the NAHP intro.** Rewritten to be self-contained in the opening paragraph of Section 4.2: the diminishing-returns intuition is stated in plain language, formalism is deferred to Appendix A. "Compression–accuracy frontier" and "cross-component edge density" no longer appear before they are defined.
>
> **R4. Motivation before HCP problem statement.** New opening paragraph in Section 4.1 under the header **"Why predict compressibility before training?"** — lists the three design choices HCP collapses into one and ties HCP's existence to the cross-dataset transfer experiment (Section 5.2).
>
> **R5. Bridge before contrastive learning section.** New opening paragraph in Section 4.3: "Sections 4.1 and 4.2 describe *how much* to prune and *what* to prune. This section addresses a separate problem: how to *learn representations* on the pruned graph." Then lists the two distortions with a why-it-matters framing.
>
> **R6. Algorithm Steps 3–4 too sparse.** Expanded in Algorithm 1 (inside Section 4.5, Density Curriculum Interpretation). Each line of Stages 3 and 4 now names the quantity being computed and points to its defining equation (retention weights → Eq. 13, adaptive temperature → Eq. 12, attention scores → Eqs. 14–17, four losses → Eqs. 7, 15, 16, 17, combined loss → Eq. 22).
>
> **R7. Minor writing issues.**
> * "We are explicit about what TriPrune-HGNN does and does not claim." — removed (Section 1, paragraph preceding the contributions list). Paragraph now opens directly with the substance.
> * "We explicitly flag three cells…" — removed from conclusion (Section 6); replaced by a compact pointer to Appendix B where per-cell tiers are reported.
> * "Eq. equation 33" — we could not reproduce this literal string in the source; if it is still visible in the compiled PDF, we would appreciate the section and page number so we can fix it.
>
> **R8. "Component" is overloaded.** Explicit convention added in the opening paragraph of Section 4 (Methodology): we now reserve *component* for the K structural contexts of the heterogeneous hypergraph and use *module* for the four learnable framework pieces. "Four learnable components" → "four learnable modules" throughout.
>
> **R9. Table 3 needs hand-holding; algorithmic vs. parametric wins.** Three additions in Section 5.1 (Ablation Studies):
> * Table 3 caption gains a **"Reading Table 3"** instruction with two guiding questions.
> * Post-table prose is rewritten as a per-mechanism prediction–observation walkthrough — for each learnable mechanism, we state what it was introduced to fix (with pointer to the methodology subsection that motivated it), what we predicted would happen when it is removed, and what the table observes.
> * A new paragraph headed **"Algorithmic vs. parametric wins"** shows that the "w/o HCP reg. only" row and all of block (b) are purely algorithmic (they add or remove no MLP parameters), yielding a rough 70%/30% split between algorithmic and parametric contributions to the full-model gain.
>
> All experimental results and theoretical statements are unchanged. Only presentation, ordering, and framing have been revised. We hope this substantially improves readability; thank you again for the depth of engagement.